# The Persistence of Neural Collapse Despite Low-Rank Bias

**Connall Garrod**
Mathematical Institute
University of Oxford
`connall.garrod@maths.ox.ac.uk`

**Jonathan P. Keating**
Mathematical Institute
University of Oxford

## Abstract

Neural collapse (NC) and its multi-layer variant, deep neural collapse (DNC), describe a structured geometry that occurs in the features and weights of trained deep networks. Recent theoretical work by Sukenik et al. using a deep unconstrained feature model (UFM) suggests that DNC is suboptimal under mean squared error (MSE) loss. They heuristically argue that this is due to low-rank bias induced by L2 regularization. In this work, we extend this result to deep UFMs trained with cross-entropy loss, showing that high-rank structures—including DNC—are not generally optimal. We characterize the associated low-rank bias, proving a fixed bound on the number of non-negligible singular values at global minima as network depth increases. We further analyze the loss surface, demonstrating that DNC is more prevalent in the landscape than other critical configurations, which we argue explains its frequent empirical appearance. Our results are validated through experiments in deep UFMs and deep neural networks.

## 1 Introduction

Many researchers have explored the geometric structures that emerge in well-trained deep neural networks (DNNs) applied to classification tasks [35, 11, 36]. Among these structures is neural collapse (NC) [36], which refers to the tendency of overparameterized neural networks to form a simplex equiangular tight frame (ETF) in their final-layer features and weights. Researchers have sought to explain the ubiquity of NC [24]. A common approach involves the unconstrained feature model (UFM) [32], which treats the last-layer features as optimization variables and optimizes them jointly with the last-layer weights, effectively abstracting away the rest of the network. Within this framework, NC has been shown to be a global optimum in various settings [32, 21], and the associated optimization function has been characterized as a strict saddle function [58, 56, 52, 57].

While NC was originally observed in the last layer, researchers have empirically verified that similar structure emerges in earlier layers of the network [13, 39, 37]—a phenomenon known as deep neural collapse (DNC). Theoretical investigations of DNC have been facilitated by an extension of the UFM, referred to as the deep UFM, which isolates multiple layers from the network's approximated component. While DNC has been shown to be optimal in a variety of restricted contexts [6, 45, 43], a recent study by Sukenik et al. [42] demonstrated that in the ReLU MSE case, DNC is generally not optimal in the deep UFM. They argue that suboptimality arises due to a low-rank bias induced by $L_2$ regularization. This is illustrated by constructing lower-rank solutions that achieve lower loss than any DNC configuration.

While this finding is significant, Sukenik et al. do not analyze whether DNC or their low-rank solutions are local minima. Consequently, the interplay between nonlinearity, the loss function, and low-rank bias remains poorly understood. Their work additionally raises a central question: why does DNC often emerge in practice despite being suboptimal? Sukenik et al. offer some empirical

evidence—DNC appears more often with increased width or reduced regularization—but no theory yet explains its empirical persistence.

To address this question, we examine a deep UFM with cross-entropy (CE) loss and linear layers. This model strikes a balance between analytical tractability and sophistication: the linear layers facilitate straightforward analysis, while the model remains sufficiently rich to admit low-rank solutions that outperform DNC. While Sukenik et al. [42] demonstrate the suboptimality of DNC by presenting an isolated better solution, we provide a complete theoretical characterization of how low-rank bias manifests across the loss landscape and why it leads to the systematic suboptimality of DNC. Specifically:

- We prove that any output matrix with rank higher than necessary to fit the data cannot be optimal when the number of separated layers $L$ is sufficiently large. This provides a global classification of how low-rank bias shapes the loss surface.

- We show that global minima must exhibit approximate low-rankness, with all but a small number of singular values decaying exponentially in $L$. This demonstrates how low-rank bias governs the structure of optimal solutions.

- We establish that DNC, although suboptimal for general $L$, remains a critical point with a positive semi-definite Hessian when the regularization level is small. This explains why local search algorithms can converge to DNC solutions. It also contrasts with the $L = 1$ case [56, 58], where the loss surface is a strict saddle function. This demonstrates that the fundamental geometry of the optimization landscape changes due to low-rank bias.

- We analyze the role of network width $d$ and show that the dimensionality of the solution space associated with DNC grows more rapidly than that of lower-rank solutions. This suggests that width significantly influences the prevalence of DNC in the loss landscape, offering a heuristic explanation for why local optimization methods frequently converges to DNC despite its suboptimality.

- We verify that DNC remains suboptimal in our model even when ReLU activations are used, suggesting that our findings for the linear model likely extend beyond linear layers and offer broader generalizability.

- We provide empirical validation on both deep UFMs and standard deep classification networks, confirming that our theoretical insights extend beyond linear models.

Together, these results provide a comprehensive theoretical characterization of low-rank bias and its implications for the loss surface, and offer the first explanation for the empirical persistence of DNC despite its suboptimality.

## 1.1 Related work

**Neural collapse**: Many researchers have studied the conditions under which NC emerges using UFMs [32, 7], including across various loss functions [57, 32, 21, 12]. Landscape analyses have revealed a favorable optimization structure, where the loss surface contains only global minima and non-degenerate saddles [56, 58, 21]. Extensions of NC research have explored its behavior in settings with a large number of classes [23] and imbalanced class distributions [51, 44, 7, 6, 14]. Generalizations from the hard-label setting have also been considered for the language setting [55] and multi-class regression [1]. The influence of dataset properties on NC has also been examined [15, 25]. Jacot et al. [20] show that NC arises in wide networks with an architecture similar to our own, but they do not address DNC, low-rank bias, or use the UFM. NC has also been linked to various practical aspects of deep learning [41, 8, 30, 18, 50, 54], and the impact of different network architectures has also been explored [29, 49]. For a review of NC, see [24].

To theoretically analyze DNC, researchers have employed deep UFMs, particularly for MSE loss. This includes two-layer networks [45], linear layers [6, 10], binary classification [43], and more general settings [42]. Additionally, DNC has been studied beyond the UFM framework using unique layer-wise training procedures [3]. Zangrando et al. [53] examine how DNC and weight decay lead to a low-rank bias. Our work, which considers solutions with lower rank than DNC, diverges substantially from theirs.

**Low-rank bias**: Low-rank bias has been studied for matrix completion [2, 4], where Schatten quasi-norm minimization [33, 47] is used to recover low-rank matrices from partial observations.

Wang et al. [48] show that in linear networks trained for matrix completion, stochastic gradient descent (SGD) can jump to lower-rank solutions but not back.

In classification settings, low-rank bias has been analyzed in architectures with linear layers. Ongie et al. and Parkinson et al. [34, 38] study networks with a single nonlinearity in the final layer, demonstrating that effective rank tends to decrease as depth increases. A similar perspective across various architectures is explored in [5]. Low-rank bias is proven for separable data and exponential-tailed loss with linear networks in the binary setting by Ji et al. [22]. In nonlinear contexts, Galanti et al. [9] examine low-rank bias in mini-batch SGD, observing that smaller batch sizes, higher learning rates, and stronger weight decay promote lower-rank neural networks. Huh et al. [17] and Le et al. [27] study factors contributing to low-rank behavior. Le et al. [27] use networks with linear final layers, but focus on binary classification and do not use unconstrained features.

## 2  Background

We consider a classification task with $K$ classes and $n$ samples per class. We denote the $i^{\text{th}}$ data point of the $c^{\text{th}}$ class by $x_{ic} \in \mathbb{R}^{d_0}$, with corresponding one-hot encoded labels $y_c \in \mathbb{R}^K$. A deep neural network $f(x) : \mathbb{R}^{d_0} \to \mathbb{R}^K$ is used to model the relationship between training data and class labels. We write the network as $f(x) = W_L \sigma(W_{L-1} \sigma(...\sigma(W_1 h(x))...))$, where $W_L \in \mathbb{R}^{K \times d}$ and $W_{L-1}, \ldots, W_1 \in \mathbb{R}^{d \times d}$ are weight matrices. The function $h : \mathbb{R}^{d_0} \to \mathbb{R}^d$ represents a highly expressive feature map, such as a ResNet. The activation function $\sigma : \mathbb{R} \to \mathbb{R}$ is applied elementwise. Network parameters are trained via a variant of gradient descent on CE loss, along with weight decay.

We denote the image of the data under the map $h$ by $h(x_{ic}) = h_{ic}^{(1)}$, and define the matrix $H_1$ to be the image of the full dataset in class order: $H_1 = [h_{1,1}^{(1)}, ..., h_{n,1}^{(1)}, h_{1,2}^{(1)}, ..., h_{n,K}^{(1)}] \in \mathbb{R}^{d \times Kn}$. Similarly, let $H_2 = W_1 H_1, H_3 = W_2 \sigma(H_2), ..., H_{L-1} = W_{L-1} \sigma(H_{L-1})$ denote the output matrices of each separated layer before the nonlinearity, with columns $h_{ic}^{(l)}$. Let $\tilde{H}_l$ for $l = 2, ..., L$ be the corresponding matrices after the non-linearity, with columns $\tilde{h}_{ic}^{(l)}$. Define the class feature means out of each layer as $\mu_c^{(l)} = \text{Av}_i\{h_{ic}^{(l)}\}$ and $\tilde{\mu}_c^{(l)} = \text{Av}_i\{\tilde{h}_{ic}^{(l)}\}$. Let $M_l$ and $\tilde{M}_l$ be the matrices with class feature means as columns in class order. Define the global means as $\mu_G^{(l)} = \text{Av}_c\{\mu_c^{(l)}\}$ and $\tilde{\mu}_G^{(l)} = \text{Av}_c\{\tilde{\mu}_c^{(l)}\}$. Also, define the simplex ETF matrix as $S = I_K - \frac{1}{K}1_K 1_K^T \in \mathbb{R}^{K \times K}$.

DNC refers to the following observations, which are found to approximately hold in overparameterized neural networks as training continues, with adherence typically increasing as the depth of the considered layer grows. By 'overparameterized,' we mean the network is capable of attaining approximately zero training error.

**Definition 1: Deep neural collapse:** *the last $L$ layers obey DNC if, for $l = 1, ..., L$, they satisfy:*

*DNC1: Feature vectors collapse onto their means:* $H_l = M_l \otimes 1_n^T, \tilde{H}_l = \tilde{M}_l \otimes 1_n^T$.

*DNC2: The feature mean matrices $M_l$ and $\tilde{M}_l$, after global centering, align with a simplex ETF:* $(M_l - \mu_G^{(l)} 1_K^T)^T (M_l - \mu_G^{(l)} 1_K^T) \propto (\tilde{M}_l - \tilde{\mu}_G^{(l)} 1_K^T)^T (\tilde{M}_l - \tilde{\mu}_G^{(l)} 1_K^T) \propto S$.

*DNC3: The rows of the weight matrices $W_l$ are, after global centering, linear combinations of the columns of the matrix $\tilde{M}_l - \tilde{\mu}_G^{(l)} 1_K^T$.*

**Deep unconstrained feature models:** To define the deep UFM, we approximate the feature map $h(x)$ as being capable of mapping the training data to arbitrary points in feature space, treating the feature vectors $h_{ic}^{(1)}$ as freely optimized variables. The loss function becomes:

$$\mathcal{L}(H_1, W_1, ..., W_L) = g(Z) + \sum_{l=1}^{L} \frac{1}{2}\lambda \|W_l\|_F^2 + \frac{1}{2}\lambda \|H_1\|_F^2, \tag{1}$$

where $g$ implements the CE loss

$$g(Z) = -\frac{1}{Kn} \sum_{c=1}^{K} \sum_{i=1}^{n} \log\left( \frac{\exp((z_{ic})_c)}{\sum_{c'=1}^{K} \exp((z_{ic})_{c'})} \right), \tag{2}$$

and $z_{ic} = W_L\sigma(W_{L-1}\sigma(...W_2\sigma(W_1 h_{ic}^{(1)})...))$ are the logit vectors, which make up the columns of the matrix $Z \in \mathbb{R}^{K \times Kn}$, using the same ordering as in the feature matrices. Here, $\lambda > 0$ is a regularization coefficient, and for simplicity, we apply equal regularization to all parameters. Note that, due to the UFM approximation, weight decay is applied to $H_1$.

**Low-rank solutions:** Recent work by Sukenik et al. [42] demonstrates that, when using ReLU layers, DNC is generally not optimal in the deep UFM when using MSE loss. They provide useful intuition for why such solutions exist. Consider the deep UFM defined in Equation (1) and focus on the regularization component. A well-known Schatten quasi-norm result [40] states:

$$\frac{1}{2}L\lambda\|X\|_{S_{2/L}}^{2/L} = \min\left\{\frac{1}{2}\lambda\|H_1\|_F^2 + \frac{1}{2}\sum_{l=1}^{L-1}\lambda\|W_l\|_F^2\right\}, \quad \text{where} \quad \|X\|_{S_p}^p = \sum_{i=1}^{\text{rank}(X)} s_i^p.$$

Here, the minimization is over $W_{L-1}, ..., W_1, H_1$ subject to $X = W_{L-1}...W_1 H_1$, and $\{s_i\}_{i=1}^{\text{rank}(X)}$ are the nonzero singular values of $X$. For large $L$, the Schatten quasi-norm approximates the rank of the matrix $X$, encouraging it to be low-rank. This is traded off against the fit loss, which requires that the network still fit the data. Since the DNC solution has rank $K$ in the MSE case, sufficiently large $K$ allows the existence of lower-rank solutions that outperform DNC.

# 3 The deep unconstrained feature model

We consider the deep UFM, defined in Equation (1), initially using linear layers, $\sigma = \text{id}$, within the separated portion of the network. While linear networks lack the expressiveness of modern deep models, the unconstrained feature assumption effectively approximates an expressive network, mitigating this limitation. This model can be regarded as approximating a standard classification architecture with appended linear layers. We consider the full nonlinear case at the end of the section. All proofs appear in Appendix B.

We now describe the specific DNC structure for this case, informed by previous works [58, 6]. First, we demonstrate a well-known simplicity in the singular value decompositions (SVDs) of the parameter matrices of a linear network at any critical solution:

**Lemma 1:** *Let the network width satisfy $d \geq K$. At any critical point of the deep linear UFM, there exists an SVD of the parameter matrices in the following form:*

$$W_l = U_l\Sigma U_{l-1}^T, \quad \text{for } l = 1, ..., L-1; \quad W_L = U_L\tilde{\Sigma}U_{L-1}^T; \quad H_1 = U_0\bar{\Sigma}V_0^T, \tag{3}$$

*where $U_L \in \mathbb{R}^{K \times K}$, $V_0 \in \mathbb{R}^{Kn \times Kn}$, $U_{L-1}, ..., U_0 \in \mathbb{R}^{d \times d}$ are all orthogonal matrices, and $\Sigma \in \mathbb{R}^{d \times d}$, $\tilde{\Sigma} \in \mathbb{R}^{K \times d}$, $\bar{\Sigma} \in \mathbb{R}^{d \times Kn}$ have their top $K \times K$ block given by $\text{diag}(s_1, ..., s_K)$, where $s_i \geq 0$ for $i = 1, ..., K$, and all other entries are zero.*

This result indicates that at critical points, each parameter matrix has equal singular values, and the singular vectors of adjacent parameter matrices are aligned.

If we construct a solution that satisfies the structure of Lemma 1, we need only specify the logit matrix $Z$, since

$$Z = W_L...W_1 H_1 = U_L\left[\text{diag}(s_1^{L+1}, ..., s_K^{L+1}) \quad 0_{K \times (n-1)K}\right] V_0^T, \tag{4}$$

and the only remaining free parameters are $U_{l-1}, ..., U_0$, which do not affect the loss. Thus, the loss at any point with this structure can be expressed entirely in terms of the logit matrix $Z$. Our definition of DNC structure is consistent with this alignment between matrices in the linear model and naturally leads to the DNC properties.

**Definition 2: DNC in the deep linear UFM:** *A DNC solution in this setting is any set of matrices $W_L, ..., W_1, H_1$ that both satisfy the SVD properties stated in Lemma 1 and yield a logit matrix of the form $Z = \alpha S \otimes 1_n^T$, for some $\alpha > 0$.*

**Proposition 1:** *Let the matrices $W_L, ..., W_1, H_1$ form a DNC solution as in Definition 2. Then these matrices satisfy the DNC definition stated in Definition 1, with $\mu_G^{(l)} = 0$ for $l = 1, ..., L$.*

## 3.1 Low-rank solutions

While the DNC structure is optimal for $L = 1$ [58], it generally ceases to be optimal when $L \geq 2$.

**Theorem 1:** *Consider the deep linear UFM with network width $d \geq K$. If $K \geq 4, L \geq 3$, or $K \geq 6, L = 2$, then no solution with DNC structure can be a global minimum.*

This result is proven by leveraging a low-rank structure observed in experiments. While the DNC structure is given by $Z \propto S \otimes 1_n^T$, the low-rank structure we analyze, when $K$ is even, takes the form

$$Z \propto \begin{bmatrix} X & 0 & ... & 0 \\ 0 & X & ... & 0 \\ ... & ... & ... & ... \\ 0 & 0 & ... & X \end{bmatrix} \otimes 1_n^T, \quad \text{where } X = \begin{bmatrix} 1 & -1 \\ -1 & 1 \end{bmatrix}. \tag{5}$$

We show that, when the scales are set equal, the low-rank solution outperforms the DNC solution.

Theorem 1 highlights that the $L = 1$ case, known as the UFM, fails to capture the full behavior of deep networks, as its optimal structure does not generalize to the last layer in deeper models. While the proof relies on a specific structure, the underlying cause is a low-rank bias, which is further explored in the next two theorems. Furthermore, since DNC and NC are equivalent in our model due to layer alignment, this result also rules out the original NC phenomenon.

We can extend Theorem 1 further: any fixed structure whose rank exceeds the minimal rank required to fit the data is also suboptimal when the number of layers $L$ is sufficiently large. To formalize this, we introduce the following definition.

**Definition 3: Diagonally superior matrix:** *A matrix $M \in \mathbb{R}^{K \times Kn}$, with columns denoted as $m_{ic}$ in class order, is diagonally superior when $(m_{ic})_c > (m_{ic})_{c'}$ for all $c' \neq c$.*

This definition is useful because such matrices can achieve arbitrarily small loss in the fit term of our objective by setting the scale appropriately large. With this, we state the next theorem.

**Theorem 2:** *Let $X, M \in \mathbb{R}^{K \times Kn}$ have ranks $p$ and $q < p$, with $M$ diagonally superior. Consider the deep linear UFM with network width $d \geq K$, where solutions are construct using the structure induced by a critical point, allowing the loss to be written in terms of $Z$. Let $\mathcal{L}_L(Z)$ denote this loss, with $L$ dependence made explicit. Then there exists $L_0 \in \mathbb{N}$ such that for all $L > L_0$,*

$$\min_{\alpha \geq 0}\{\mathcal{L}_L(\alpha X)\} \geq \min_{\beta \geq 0}\{\mathcal{L}_L(\beta M)\},$$

*with equality occurring only if the minimum is at $\alpha = \beta = 0$.*

Theorem 2 asserts that for sufficiently large $L$, no structure can be optimal if its rank exceeds that of the rank-minimal diagonally superior matrix of size $K \times Kn$, except the trivial zero solution when regularization is too large. This underscores the role of low-rank bias: when a matrix can achieve arbitrarily small loss on the fit term, low-rank bias ensures that—once enough layers are separated—it will outperform any higher-rank solution. While this theorem requires large enough $L$, in practice a well-chosen $M$ can outperform high-rank solutions even for small $L$, as shown in Theorem 1.

**The imbalanced case:** Thus far, we have focused on balanced classes, but our arguments extend to imbalanced settings as well. A concise description of the single-layer UFM in the imbalanced case is provided by Thrampoulidis et al. [44]. Using their characterization of emergent structures in imbalanced scenarios, we show that the analogy to DNC in these settings is also globally suboptimal in the deep linear UFM. A detailed discussion and supporting results are given in Appendix A.

## 3.2 Optimal structure

We now aim to characterize how low-rank bias influences the structure at global optima. To do so, we must explicitly account for the scaling of the regularization parameter as $L$ increases. To clarify this dependence, we write $\lambda(L) = \lambda_L$. Additionally, we note that the minimal rank of a diagonally superior matrix in $\mathbb{R}^{K \times Kn}$ is the same as in $\mathbb{R}^{K \times K}$. Therefore, this minimal rank is independent of $n$ and can be denoted by $q_K$. The following theorem provides detailed information about the singular values of a globally optimal solution.

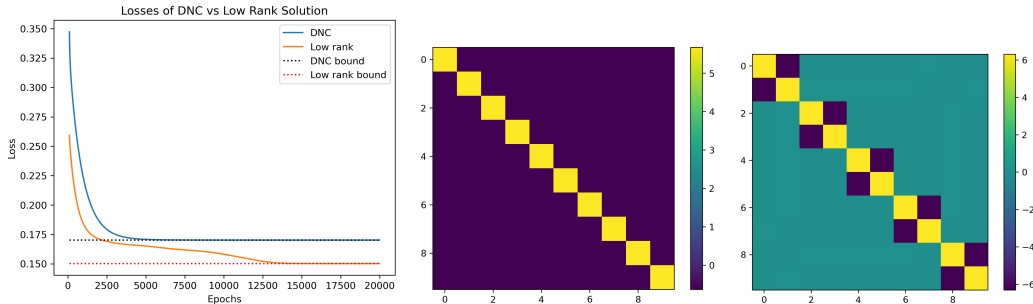

Figure 1: Experiments in the deep linear UFM: **Left**: Loss curves for a solution that converges to DNC versus one that converges to the low-rank structure described in Equation (5). **Middle/Right**: Corresponding mean logit matrices at convergence. Hyperparameters: $L = 2$, $d = 70$, $\lambda = 2^{-10}$, $K = 10$, $n = 5$, learning rate $= 0.5$.

**Theorem 3:** *Consider a sequence of deep linear UFM optimization problems, indexed by the number of layers $L = 1, 2, 3, \ldots$. Assume the regularization parameter satisfies $\lambda_L > 0$ for all $L$, and that the network width is fixed at $d \geq K$. Denote each loss function by $\mathcal{L}_L$. Let $Z_L^*$ be the network output at a global optimum of $\mathcal{L}_L$. Also define the minimal rank of a diagonally superior matrix in $\mathbb{R}^{K \times Kn}$ to be $q_K$. The following holds:*

*(i) If $\lambda_L$ is scaled such that $\lambda_L^{-1} = o(L)$, there exists $L_0$ such that for all $L > L_0$, each singular value of $Z_L^*$ is zero or converges to zero at a rate faster than exponential in $L$.*

*(ii) If $\lambda_L$ is scaled such that $\lambda_L = o(L^{-1})$, there exists an $L_0$ such that for all $L > L_0$, the matrix $Z_L^*$ is diagonally superior, and hence has rank at least $q_K$. Furthermore, at most $q_K$ singular values of the normalized matrix $\hat{Z}_L^*$ are neither zero, nor converge to zero at a rate at least exponential in $L$.*

The second part of this theorem further clarifies the role of low-rank bias. When regularization is sufficiently weak, increasing $L$ enhances the low-rank bias, causing most singular values of the normalized optimal matrix $\hat{Z}_L^*$ to decay exponentially in $L$. As a result, $\hat{Z}_L^*$ resembles a low-rank matrix, with at most an exponentially suppressed perturbation.

This raises the question of how low rank a diagonally superior matrix can be. Consider a matrix $X \in \mathbb{R}^{2 \times K}$, where each column $x_c$ satisfies $|x_c|_2 = 1$ and no column is repeated. Then $X^\top X \in \mathbb{R}^{K \times K}$ is diagonally superior and has rank 2, implying $q_K \leq 2$. This demonstrates the significant gap between the rank of the optimal solution for large $L$ and that of the DNC solution.

### 3.3 The prevalence of deep neural collapse

Although DNC is not globally optimal, numerical experiments show that models often converge to it—especially when the regularization parameter $\lambda$ is small or the width $d$ is large. We also observe DNC in real networks, even where our model predicts suboptimality. To explain this persistence, we show that there are DNC solutions that remain locally attractive despite being suboptimal.

**Theorem 4:** *Consider the deep linear UFM with network width $d \geq K$. When the regularization parameter $\lambda > 0$ is suitably small, the following holds:*

*(i) There exists solutions with DNC structure that are critical points of the model. Moreover, a subset of these solutions have a scale $\alpha$ that diverges as $\lambda \to 0$.*

*(ii) These scale-divergent solutions have Hessian matrices with no negative eigenvalues.*

This result makes it clear that although no DNC solution corresponds to a global minimum, there are DNC solutions that correspond to local minima or degenerate saddle points—both of which are difficult for local optimization methods to escape. Consequently, gradient descent can readily converge to DNC solutions in this model.

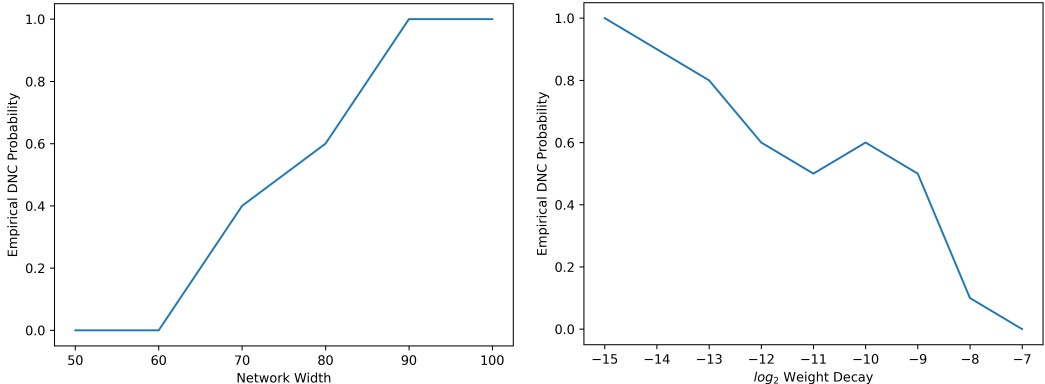

Figure 2: Experiments in the deep linear UFM: **Left**: Empirical probability of DNC versus width $d$. **Right**: Empirical probability of DNC versus regularization $\lambda$. Averaged over 10 runs; same hyperparameters as Figure 1.

While this result is necessary for the persistence of DNC, it is not sufficient. We now turn our attention to the network width parameter $d$, and examine how changes in the loss surface structure may further incentivize DNC. We begin with the following definition:

**Definition 4: Dimension of a critical point in the loss surface:** *Suppose $Z^*$ denotes the network output at a critical point of the deep linear UFM loss. Define the dimension of the space of network parameters $(W_L, ..., W_1, H_1)$ that produce $Z^*$ to be $D_{Z^*}$.*

In the specific case where the solution has DNC structure, denote this dimension by $D_{\text{DNC}}$. We can compare the dimension of a general low-rank solution to that of the DNC solution.

**Theorem 5:** *Consider the deep linear UFM with network width $d \geq K$. Let the regularization parameter $\lambda > 0$ be small enough that there exists a DNC solution that is a critical point of the loss. let $Z^*$ be the network output at another critical point at some fixed scale, with $rank(Z^*) = r \in [2, K-1)$. Define the ratio of the dimensions in the loss surface of $Z_{DNC}$ and $Z^*$ to be*

$$R(d) = \frac{D_{\text{DNC}}(d)}{D_{Z^*}(d)}.$$

*This ratio is a monotonic increasing function on the set of natural numbers $d \geq K$, starting below 1 and tending towards $(K-1)/r > 1$ as $d \to \infty$.*

This theorem shows that, as $d$ increases, the dimension associated with DNC eventually exceeds that of any low-rank solution. Since both $D_{\text{DNC}}(d)$ and $D_{Z^*}(d)$ tend to infinity as $d \to \infty$, the difference $D_{\text{DNC}}(d) - D_{Z^*}(d)$ also diverges. This implies that, in the large-width limit, DNC structure becomes more prevalent in the loss surface relative to low-rank solutions, which helps explain why local optimization methods may converge to DNC more frequently. Although low-rank solutions are still observed even when $R(d) > 1$, suggesting that flatness and basin-of-attraction properties also play important roles, this result supports the growing persistence of DNC as influenced by the parameter $d$.

**Impact of the regularization parameter $\lambda$:** As $\lambda \to 0$, the loss of any diagonally superior structure at its optimal scale tends to zero. This implies that as $\lambda$ becomes small, the DNC loss becomes arbitrarily close to the optimal loss. We can also compare local flatness measures of the DNC solution to those of the low-rank solution described in Equation (5).

**Theorem 6:** *Consider the deep linear UFM with network width $d \geq K$, where the number of classes $K$ is even. Let the regularization parameter $\lambda$ be small enough that there exists both low-rank solutions as described in Equation (5), and DNC solutions, that are critical points of the loss with scales that diverge as $\lambda \to 0$. Define the local flatness of these solutions, measured by the Frobenius, spectral, or trace norm of their respective Hessians, as $F_{\text{DNC}}$ and $F_{\text{LR}}$. Then, as $\lambda \to 0$, we have:*

$$F_{\text{DNC}}, F_{\text{LR}} \to 0, \quad \text{such that} \quad \frac{F_{\text{DNC}}}{F_{\text{LR}}} \to C,$$

*where $C$ is a constant that depends on the choice of flatness measure.*

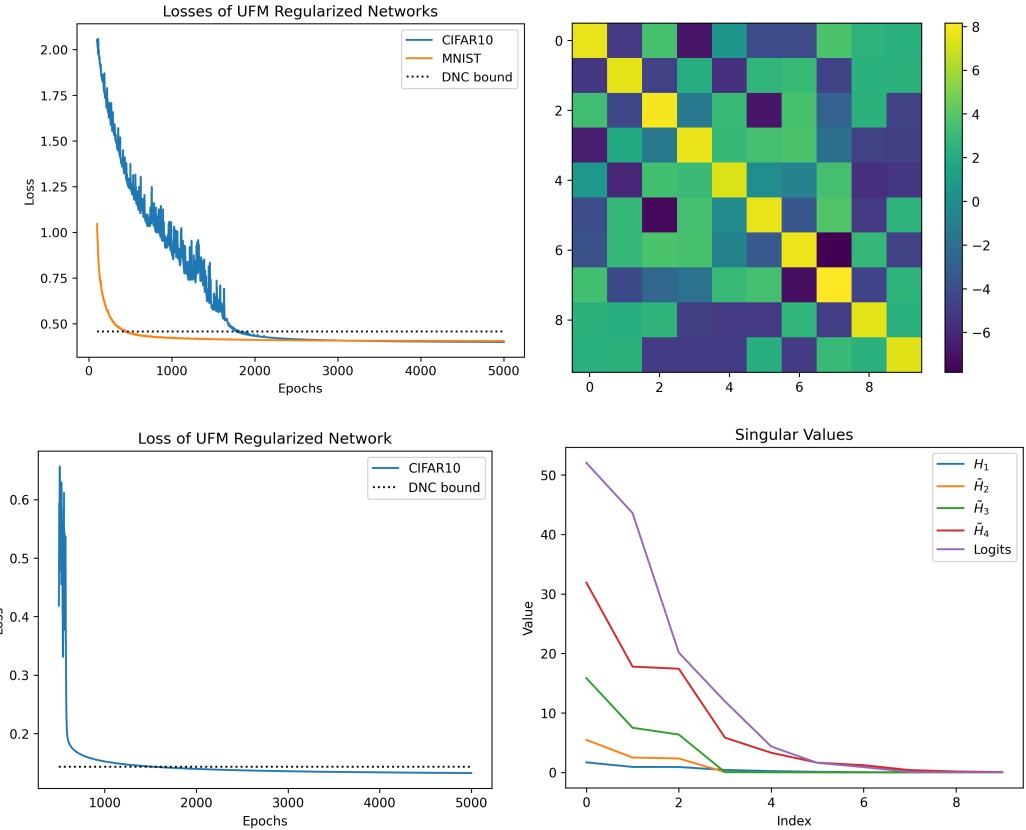

Figure 3: Experiments using UFM-style regularization: **Top**: Losses of low-rank solutions on MNIST and CIFAR-10 using linear layers in the fully connected head, along with mean logit matrix for CIFAR-10. Hyperparameters: $L = 3$, $d = 64$, $\lambda_W = 5 \times 10^{-3}$, $\lambda_H = 10^{-6}$, learning rate $= 0.05$. **Bottom**: Loss and singular values across layers for CIFAR-10 using ReLU in the fully connected head. Same hyperparameters as above, except $L = 4$.

This result shows that both critical points have local flatness metrics tending to zero, with their ratio converging to a constant. This suggests that flatness alone does not account for the preferential selection of DNC. However, when combined with the insight from Theorem 5, it supports a heuristic explanation for DNC's persistence: while all minima are approximately equally flat and perform equally well, DNC structures dominate in number. We provide numerical evidence for this interpretation in the next section and leave a rigorous demonstration of such a connection to future work.

While this analysis sheds light on the geometry of the loss surface, we ultimately believe that fully understanding behavior in the $\lambda \to 0$ limit will require accounting for the implicit biases of gradient descent and the role of initialization scale. In particular, whether initialization causes the network output to initially align with a simplex—and thereby increases the likelihood of falling into the basin of attraction of DNC—appears more amenable to investigation through gradient flow dynamics than through landscape analysis.

### 3.4 The deep ReLU unconstrained feature model

We show that low-rank bias extends to the deep UFM, defined in Equation (1), with ReLU activations. This provides evidence that the phenomena described in previous sections persist in more realistic settings. First, we note how the definition of DNC structure changes.

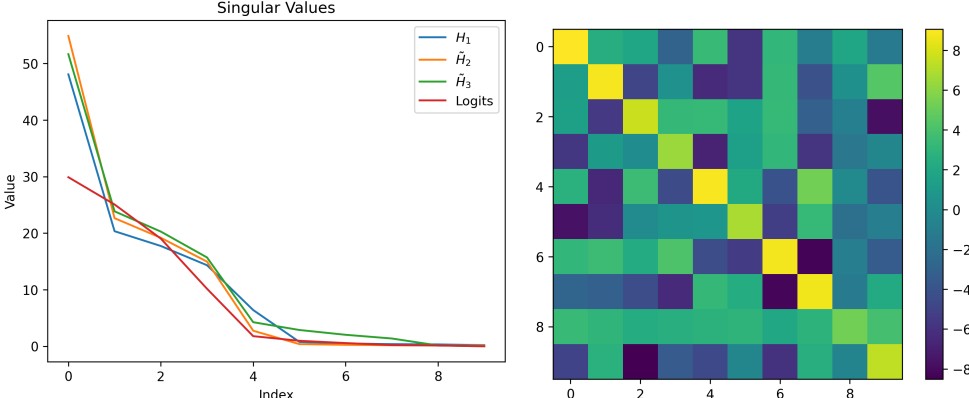

Figure 4: Experiments with standard regularization on CIFAR-10: **Left**: Singular values of each feature matrix in the fully connected head. **Right**: Mean logit matrix. Hyperparameters: $L = 3$, $d = 64$, $\lambda = 10^{-2}$, learning rate = $10^{-3}$.

**Definition 5: DNC in the deep ReLU UFM:** *A solution of the deep ReLU UFM has DNC structure if it satisfies the properties of DNC stated in Definition 1, and additionally, the distinct columns of the logit matrix directly align with a simplex ETF: $Z \propto S \otimes 1_n^T$.*

The additional condition on $Z$ is consistent with experiments, matches the linear case, and is implied by the original NC observation of Papyan. We also make the following technical assumption:

**Assumption 1:** *For a DNC solution in the deep ReLU UFM, the ratio of the norm of the global mean to the norm of the simplex component of $M_l$ is not decreased by the application of ReLU. Specifically, for $2 \le l \le L$ if we define*

$$r_l = \frac{\|\mu_G^{(l)}\|_2}{\|M_l - \mu_G^{(l)} 1_K^T\|_F}, \quad \tilde{r}_l = \frac{\|\tilde{\mu}_G^{(l)}\|_2}{\|\tilde{M}_l - \tilde{\mu}_G^{(l)} 1_K^T\|_F},$$

*then $r_l \le \tilde{r}_l$.*

This technical assumption is used to avoid solutions that begin with a large global mean and gradually reduce it to amplify the simplex ETF component. Such behavior would result in early features approximating a rank-1 matrix with a small perturbation, rather than the typical $K - 1$-dimensional simplex ETF. These cases do not arise in our experiments, and Assumption 1 is found to hold. It is likely that such a structure is prohibited by the properties of ReLU. With this, we present the following:

**Theorem 7:** *Consider the deep ReLU UFM with network width $d \ge K$. If Assumption 1 holds, and $K \ge 10, L \ge 5$, or $K \ge 16, L = 4$, then no DNC solution can be globally optimal.*

This theorem is proven by showing that the DNC loss in the linear model is a lower bound for the loss of a DNC solution in the ReLU model, and then constructing a solution that outperforms this bound using a method similar to that in Theorem 1.

Theorem 7 confirms that the low-rank bias observed in the linear case also occurs in the more realistic ReLU setting. Moreover, the similarity between the proofs for the two cases suggests that the linear model effectively captures key phenomenological aspects of the more complex nonlinear setting.

## 4 Numerical experiments

We empirically validate our theoretical results in the deep UFM, and provide supporting experimental evidence on the MNIST [28] and CIFAR-10 [26] datasets using both UFM-style and standard regularization. Additional experiments and details can be found in Appendix D.

**Deep UFM experiments**: We Examine the deep linear UFM, defined in Equation (1), trained from random initialization using gradient descent. Figure 1 compares the loss achieved by a DNC solution

with that of the low-rank solution described in Equation (5). The final class mean logit matrices at convergence are also shown. As predicted by Theorem 1, the low-rank solution outperforms the DNC solution. Figure 2 shows how the probability of DNC changes as network width $d$ increases or regularization decreases. We observe that DNC is unlikely at small widths but becomes increasingly probable as the network widens, supporting the argument derived from Theorem 5. We also observe that the probability of DNC tends to increase as regularization decreases.

**Full network experiments**: We next consider a ResNet-20 feature map with a fully connected head, applied to the MNIST and CIFAR-10 datasets using UFM-style regularization. The top row of Figure 3 shows the losses achieved by low-rank solutions when using linear layers in the fully connected component. We see that the DNC bound is beaten by such solutions. We also display the logit matrix at output, which clearly is not deep neural collapsed. The number of singular values above $10^{-6}$ was 4 for MNIST and 3 for CIFAR-10. The bottom row of Figure 3 uses ReLU activations in the fully connected head. Again, DNC is beaten by our solution, which is low rank, as is clear from the plot of its singular values. Finally, we consider a network trained with standard regularization, where weight decay is applied to every parameter in the network. Figure 4 shows that the resulting structure is low rank, and the mean logits do not form a simplex. This demonstrates that low-rank bias has implications even outside the modeling context.

# 5 Conclusion

In this work, we investigate low-rank bias in deep UFMs, providing general results on how the loss varies with the rank of different structures and offering insights into globally optimal solutions. This constitutes the first rigorous account of how low-rank bias influences the solutions reached by local search algorithms in these models. We also extend the findings of Sukenik et al. [42] on the suboptimality of DNC to both linear and ReLU-based CE models. Furthermore, we examine how the loss landscapes of these models promote DNC solutions despite their suboptimality, suggesting that the size of the solution space plays a significant role. This is supported by experiments within the model as well as in deep neural networks trained on real data using standard procedures. Together, these results establish the first theoretical foundation for the empirical persistence of DNC.

**Future Work**: In this paper, we examined how variations in the parameters $d$ and $\lambda$ reshape the loss surface and influence the likelihood of DNC solutions. However, we did not explore how the choice of local optimizer or the initialization scheme may bias convergence toward DNC. These factors are clearly relevant. For example, Sukenik et al. [42] note that the learning rate can affect the probability of reaching a DNC solution. Regarding initialization, prior work has shown that small initializations can significantly influence the rank structure of the converged solution [19, 31]. Other hyperparameters, such as batch size, may also play a role. Investigating how these factors contribute to the persistence of DNC would be a valuable direction for future research.

We also only provide a preliminary investigation into how the regularization parameter affects the frequency of DNC in our model. While the implicit bias of linear networks without explicit regularization has been widely studied [46], most of this work focuses either on binary classification—where the logit matrix reduces to a vector and the Schatten quasi-norm collapses to the Frobenius norm—or on two-layer networks, where the form of low-rank bias we describe does not arise. A more comprehensive understanding of how implicit bias manifests in deeper networks without explicit regularization would likely clarify why the limit $\lambda \to 0$ increases the frequency of DNC.

**Limitations**: Our work focuses on a theoretical model of training dynamics and does not address downstream properties such as generalization or robustness. Additionally, while our analysis of the deep UFM captures useful structural biases, it omits important practical factors like batch size, initialization scale, and optimizer choice. These elements may interact with low-rank bias in nontrivial ways. Additionally, the UFM assumes overparameterization—and is well supported empirically for this setting—but its applicability may diminish in underparameterized or highly constrained regimes.

## Acknowledgments and Disclosure of Funding

We are grateful to Christos Thrampoulidis for an extremely helpful discussion that stimulated our consideration of the imbalanced dataset case. CG also thanks Peter Sukenik for discussions regarding the minimal rank of a diagonally superior matrix. CG is supported by the Charles Coulson Scholarship. The authors also acknowledge support from His Majesty's Government in the development of this research. For the purpose of Open Access, the authors have applied a CC BY public copyright license to any Author Accepted Manuscript (AAM) version arising from this submission.

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

# A   Imbalanced classes in the deep linear UFM

Our results on low-rank bias in the deep linear UFM extend naturally to the case of imbalanced classes. This setting has been explored in several recent works [51, 7, 44, 14], though our focus will be on the studies by Thrampoulidis et al. [44] and Hong et al. [14]. We begin by establishing general properties of the low-rank bias in the imbalanced setting, before looking at what can be said specifically about DNC.

## A.1   General results in the imbalanced case

Assume we have $K$ classes, where the number of data points in class $c$ is $n_c$, and the total number of data points is $N = \sum_{c=1}^{K} n_c$. The loss function for the deep linear UFM in this imbalanced setting is given by:

$$\mathcal{L} = g(Z) + \frac{1}{2}\lambda \sum_{l=1}^{L} \|W_l\|_F^2 + \frac{1}{2}\lambda\|H_1\|_F^2.$$

In this expression, the feature vectors are represented by the matrix $H_1 \in \mathbb{R}^{d \times N}$, which we assume is ordered so that the first $n_1$ columns correspond to class 1, the next $n_2$ columns to class 2, and so on. The function $g$ computes the sum of cross-entropy losses between the column vectors of $Z = W_L \cdots W_1 H_1$ and the corresponding labels $y_c$, that is:

$$g(Z) = \frac{1}{N} \sum_{c=1}^{K} \sum_{i=1}^{n_c} -\log\left(\frac{\exp((z_{ic})_c)}{\sum_{c'=1}^{K} \exp((z_{ic})_{c'})}\right),$$

where $(z_{ic})$ denotes the network output of the $i$-th sample of the corresponding class $c$.

Importantly, the result in Lemma 1 holds independently of the class structure; that is, it does not depend on the specific values of $n_c$. While the dimensions of $H_1$ are adjusted accordingly, the structural result remains unchanged. Thus, at any optimal point of the loss, the same structural constraints from Lemma 1 apply, allowing us to reduce the problem to specifying only the network output $Z \in \mathbb{R}^{K \times N}$. The loss can then be expressed as:

$$\mathcal{L}(Z) = g(Z) + \frac{1}{2}\lambda(L+1)\|Z\|_{S_{\frac{2}{L+1}}}^{\frac{2}{L+1}}.$$

We now generalize the notion of a diagonally superior matrix to the imbalanced class setting:

**Definition:** $(K; n_1, ..., n_K)$ **Diagonally superior matrix:** *A matrix $\tilde{M} \in \mathbb{R}^{K \times N}$, is said to be $(K; n_1, ..., n_K)$ diagonally superior, where $N = \sum_c n_c$, if the first $n_1$ columns have their largest element in the first entry, the next $n_2$ columns have their largest element in the second entry, and so on.*

With this definition in place, we can now state a generalization of Theorem 2 from the main text:

**Theorem A1:** *Let $X \in \mathbb{R}^{K \times N}$ be a matrix of rank $p$, and let $\tilde{M} \in \mathbb{R}^{K \times N}$ be a $(K; n_1, ..., n_c)$ diagonally superior matrix of rank $q < p$. Consider the deep linear UFM with $K$ classes of sizes $n_1, ..., n_K$, and suppose $d \geq K$. Construct solutions using the structure induced by an optimal point so that the loss can be written entirely in terms of the output matrix $Z$. Let $\mathcal{L}_L(Z)$ denote this loss, with its dependence on $L$ made explicit. Then there exists $L_0 \in \mathbb{N}$ such that for all $L > L_0$, we have:*

$$\min_{\alpha \geq 0}\{\mathcal{L}_L(\alpha X)\} \geq \min_{\beta \geq 0}\{\mathcal{L}_L(\beta \tilde{M})\},$$

*with equality only if the minimum is attained at $\alpha = \beta = 0$.*

The proof closely mirrors that of Theorem 2 in the main text. Specifically, we identify a scale at which $\tilde{M}$ achieves a lower loss than $X$ on the fit term, and when enough layers are separated, the lower rank of $\tilde{M}$ ensures it also outperforms $X$ on the regularization term. The full proof is provided in Appendix C.1. This establishes that the same low-rank bias persists in the imbalanced setting, with analogous implications to those in the balanced case.

We also present the following generalization of Theorem 3 from the main text:

**Theorem A2:** *Consider a sequence of deep linear UFM optimization problems indexed by the number of layers $L = 1, 2, 3, \ldots$. Assume the regularization parameter obeys $\lambda_L > 0$ for all $L$, that there are $K$ classes of sizes $n_1, \ldots, n_K$ respectively , and the network width obeys $d \geq K$. Let $\mathcal{L}_L$ denote the corresponding loss functions, making the dependence on $L$ explicit. Let $Z_L^*$ represent the network output at a global optimum of $\mathcal{L}_L$. Define the minimal rank of a diagonally superior matrix in $\mathbb{R}^{K \times K}$ to be $q_K$. Then:*

*(i) If $\lambda_L$ is scaled such that $\lambda_L^{-1} = o(L)$, there exists $L_0$ such that for all $L > L_0$, each singular value of $Z_L^*$ is zero, or converges to zero at a faster than exponential rate in $L$.*

*(ii) If $\lambda_L$ is scaled such that $\lambda_L = o(L^{-1})$, there exists an $L_0$ such that for all $L > L_0$ the matrix $Z_L^*$ is $(K; n_1, \ldots, n_c)$ diagonally superior, and consequently has rank at least $q_K$. In addition, when $L > L_0$ at most $q_K$ singular values of the normalized matrix $\hat{Z}_L^*$ are neither zero, nor converge to zero at a rate at least exponential in $L$.*

The proof follows the same structure as that of Theorem 3 in the main text and is given in Appendix C.2. As in the balanced case, these results illustrate how as the number of separated layers $L$ increases, the global optima become more biased towards low rank diagonally superior matrices.

## A.2 Deep neural collapse results in the imbalanced case

We now turn our attention to DNC in the imbalanced setting. Ideally, we would like a generalization of Theorem 1 from the main text. However, this is challenging because full characterizations of optimal structures under general class imbalance are not currently available in the literature. While Hong et al. [14] provide valuable insights for general imbalance, their results involve unevaluated constants, which prevent direct recovery of rank information. Moreover, the phenomenon of minority collapse—where minority classes receive vanishing representation—can result in solutions with lower rank than those observed in the balanced case.

Thrampoulidis et al. [44], on the other hand, focus on the no-regularization setting, but they do provide a concrete result in the $\lambda \to 0$ limit for a specific class of imbalances. Their analysis enables us to draw conclusions about DNC behavior when regularization is sufficiently small. Before stating our result, we recall two useful definitions adapted from their work:

**Definition:** $(R, \rho)$**-Step imbalance:** *In a $(R, \rho)$-STEP imbalance setting with label-imbalance ratio $R \geq 1$ and minority fraction $\rho \in (0, 1)$ the following hold: all minority (resp. majority) classes have the same size $n_{min}$ (resp. $Rn_{min}$). There are $(1 - \rho)K$ majority and $\rho K$ minority classes. Without loss of generality, assume classes $\{1, \ldots, (1 - \rho)K\}$ are majorities.*

**Definition: SEL matrix:** *The simplex-encoded label (SEL) matrix $\hat{S} \in \mathbb{R}^{K \times N}$ is a $(K; n_1, \ldots, n_c)$ diagonally superior matrix, in which all columns corresponding to the same class are identical. The distinct columns of $\hat{S}$ are given by the columns of the standard simplex ETF $S = I_K - \frac{1}{K}1_K 1_K^T \in \mathbb{R}^{K \times K}$.*

In Theorem 2 of their work, Thrampoulidis et al. show that, in the single layer UFM, with a $(R, \rho)$-STEP imbalance, there exists a unique global minimizer $Z_\lambda^*$ for each $\lambda > 0$. However, unlike in the balanced case, they demonstrate in Proposition 1 of their main text that the frame of this matrix depends non-trivially on $\lambda$, not just its scale. Moreover, they show (as a consequence of their Proposition 2) that in the $(R, \rho)$-STEP imbalance setting, the global minimizer of the single-layer UFM aligns with the SEL matrix in the $\lambda \to 0$ limit.

While the structure of the optimal solution at non-zero regularization in the UFM case is not fully known, we treat the frame $Z_\lambda^*$ as the natural generalization of DNC in the deep UFM to the imbalanced case when $\lambda > 0$. This is justified by the fact that, in the balanced setting, the DNC structure is recovered as the global minimizer of the single-layer UFM. Therefore, in the presence of STEP imbalance, we define the DNC solution to be any matrix of the form $Z \propto Z_\lambda^*$, equipped with parameter matrices satisfying the SVD properties established in Lemma 1. Here, "proportional" refers to allowing differences in the overall scale of $Z$, which may vary depending on the depth of the network (i.e., number of separated layers), but we assume the frame structure of the DNC solution is determined entirely by $Z_\lambda^*$.

We have the following theorem about this generalized DNC structure:

**Theorem A3:** *Consider the deep linear UFM with $K$ classes of sizes $n_1, ..., n_K$ respectively, and with network width $d \geq K$. Also assume we are in a $(R, \rho)$-STEP imbalance setting. Then there exists $\lambda_0, L_0 > 0$ such that for all $\lambda < \lambda_0$, $L > L_0$, and $K \geq 4$, there is no generalized DNC structure that is a global minimum of the loss.*

The proof proceeds by noting that in order to align with the SEL geometry for small $\lambda$, the generalized DNC structure must have rank $K - 1$, since this is the rank of the SEL geometry. As a consequence we can apply Theorem A1, where we use a generalization of the same structure from the proof of Theorem 1 in the main text as the explicit lower rank generalized diagonally superior matrix. Full details are provided in Appendix C.3.

We find that notions of DNC in the imbalanced setting are also no longer globally optimal in general when we have more than one layer separated in our model, reinforcing the fact that the suboptimality of DNC is not confined to balanced class settings. However, unlike in the balanced case, we do not obtain an explicit bound on $L_0$, and so it could, in principle, be large.

While we are currently limited to the $(R, \rho)$-STEP setting, we expect this result to extend to more general imbalance structures when $\lambda$ is sufficiently small and minority collapse does not occur.

# B  Deep UFM proofs

We begin by stating some useful definitions for the deep linear UFM model. Recall our loss function from Equation (1), which when using linear layers becomes:

$$\mathcal{L} = g(W_L...W_1 H_1) + \frac{1}{2}\lambda \sum_{l=1}^{L} \|W_l\|_F^2 + \frac{1}{2}\lambda \|H_1\|_F^2,$$

where the function $g$ is defined as:

$$g(Z) = \frac{1}{Kn} \sum_{c=1}^{K} \sum_{i=1}^{n} -\log\left(\frac{\exp((z_{ic})_c)}{\sum_{c'=1}^{K} \exp((z_{ic})_{c'})}\right).$$

Here, the matrix $Z \in \mathbb{R}^{K \times Kn}$ is written in the form $Z = [z_{11}, z_{21}, ..., z_{n1}, z_{12}, ..., z_{Kn}]$, where $z_{ic}$ denotes the column corresponding to the $i$-th sample of class $c$.

We also define the following quantities:

$$\text{for } 0 \leq l \leq L - 1: A_{l+1} = W_L...W_{l+1} \in \mathbb{R}^{K \times d},$$
$$\text{for } 1 \leq l \leq L: H_l = W_{l-1}...W_1 H_1 \in \mathbb{R}^{d \times Kn},$$
$$Z = W_L...W_1 H_1 \in \mathbb{R}^{K \times Kn},$$
$$P = [p_{11}, p_{21}, ..., p_{n1}, p_{12}, ..., p_{Kn}] \in \mathbb{R}^{K \times Kn}, \quad \text{where } p_{ic} = \frac{\exp((z_{ic})_c)}{\sum_{c'=1}^{K} \exp((z_{ic})_{c'})},$$
$$Y = I_K \otimes 1_n^T \in \mathbb{R}^{K \times Kn},$$

We also define for later notational convenience $A_{L+1} = I_K, H_0 = I_{Kn}$.

Lastly, we will use the notation that $D \in \mathbb{R}^{d \times d}, \tilde{D} \in \mathbb{R}^{K \times d}, \bar{D} \in \mathbb{R}^{d \times Kn}, D' \in \mathbb{R}^{K \times K}$ all represent matrices where the top $K \times K$ block is given by $\text{diag}(1, 1, ..., 1, 0)$, with all other entries being 0.

## B.1 Proof of Lemma 1

For this section, it is convenient to denote $H_1 = W_0$ to make the equations more compact. Using the quantities defined at the start of this appendix, we have the following elementary calculations:

$$\frac{\partial Z_{xy}}{\partial (W_l)_{ab}} = (A_{l+1})_{xa}(H_l)_{by}, \tag{6}$$

$$\frac{\partial g(Z)}{\partial Z} = \frac{1}{Kn}(P - Y).$$

Hence, the set of first order differential equations becomes

$$\frac{\partial \mathcal{L}}{\partial W_l} = \frac{1}{Kn} A_{l+1}^T (P - Y) H_l^T + \lambda W_l. \tag{7}$$

At any optimal point, we clearly have

$$W_l^T \frac{\partial \mathcal{L}}{\partial W_l} - \frac{\partial \mathcal{L}}{\partial W_{l-1}} W_{l-1}^T = 0,$$

and substituting the explicit forms of these derivatives yields

$$W_l^T W_l = W_{l-1} W_{l-1}^T.$$

This implies that at a local optimum, all weight matrices have equal rank. Consequently, the rank is at most $K$, since $W_L$ has at rank at most $K$.

We now express each weight matrix using its singular value decomposition (SVD) as follows:

for $l = 1, ..., L - 1$: $W_l = U_l \Sigma_l V_l^T$, with $U_l, \Sigma_l, V_l \in \mathbb{R}^{d \times d}$,

$W_0 = U_0 \Sigma_0 V_0^T$, with $U_0 \in \mathbb{R}^{d \times d}, \Sigma_0 \in \mathbb{R}^{d \times Kn}, V_0 \in \mathbb{R}^{Kn \times Kn}$,

$W_L = U_L \Sigma_L V_L^T$, with $U_L \in \mathbb{R}^{K \times K}, \Sigma_L \in \mathbb{R}^{K \times d}, V_L \in \mathbb{R}^{d \times d}$,

where each $U_l$ and $V_l$ is orthogonal, and $\Sigma_l$ has non-zero entries only on its diagonal. Using the expressions derived from the gradients, we obtain for $2 \leq l \leq L - 1$:

$$(V_l \Sigma_l^T U_l^T)(U_l \Sigma_l V_l^T) = (U_{l-1} \Sigma_{l-1} V_{l-1}^T)(V_{l-1} \Sigma_{l-1} U_{l-1}^T)$$

Which implies

$$V_l \Sigma_l^2 V_l^T = U_{l-1} \Sigma_{l-1}^2 U_{l-1}^T, \tag{8}$$

From this, we conclude that for $2 \leq l \leq L - 1$

$$\Sigma_l = \Sigma_{l-1} = \Sigma,$$

where $\Sigma = \text{diag}(s_1, s_2, ..., s_K, 0, ..., 0)$, and $s_1, ..., s_K$ denote the top $K$ singular values.

We can extend this to the cases $l = L$ and $l = 1$, giving:

$$V_L \Sigma_L^T \Sigma_L V_L^T = U_{L-1} \Sigma^2 U_{L-1}^T, \tag{9}$$

$$V_1 \Sigma^2 V_1^T = U_0 \Sigma_0^T \Sigma_0 U_0^T. \tag{10}$$

This implies that $\Sigma_L = \tilde{\Sigma} = [\mathrm{diag}(s_1, ..., s_K), 0_{K \times (d-K)}]$, and $\Sigma_0 = \bar{\Sigma}$, where the top $K \times K$ block of $\bar{\Sigma}$ is $\mathrm{diag}(s_1, ..., s_K)$, and all other entries are 0.

Finally, we aim to show that there exists a choice of SVD for the weight matrices where the left and right singular vectors align. There is a subtlety here, since Equations (8), (9) and (10) together imply that for any $l$ such a decomposition exists, but does not guarantee a decomposition exists where it holds for all $l$ at once. Such a decomposition can be achieved by working iteratively. choose $V_1 = U_0$ using Equation (10), this places some constraint on the choice of $U_1$. Next pick $V_2$ so it matches $U_1$, which places some constraint on $U_2$, and so on. The outcome is an SVD decomposition where $V_l = U_{l-1}$ for $l = 1, ..., L$, which completes the proof.

## B.2   Proof of Proposition 1

We consider a set of parameter matrices $W_L, ..., W_1, H_1$ for which their SVDs can be expressed in the form described in Lemma 1. We also assume that the network output has the form $Z = \alpha S \otimes 1_n^T$, where $S = I_K - \frac{1}{K} 1_K 1_K^T$. Our aim is to show this implies the parameter matrices obey the DNC properties of Definition 1.

First observe that the matrix $S$ admits an SVD of the form $S = QD'Q^T$, where $D' = \mathrm{diag}(1, 1, ..., 1, 0) \in \mathbb{R}^{K \times K}$ and $Q$ is orthogonal. The SVD of the vector $1_n^T$ is given by $1_n^T = \sqrt{n}(e_1^{(n)})^T R^T$, where $R \in \mathbb{R}^{n \times n}$ is orthogonal, and $e_1^{(n)}$ is the first standard basis vector in $\mathbb{R}^n$. Hence the SVD of the Kronecker product $S \otimes 1_n^T$ can be written as

$$S \otimes 1_n^T = \sqrt{n}Q \left( D' \otimes (e_1^{(n)})^T \right) (Q \otimes R)^T.$$

Using the SVD forms of the parameter matrices from Lemma 1, we have

$$W_L ... W_1 H_1 = U_L [\mathrm{diag}(s_1^{L+1}, ..., s_K^{L+1}), 0_{K \times (n-1)K}] V_0^T. \tag{11}$$

Comparing this with the SVD form of $Z$, we find that $s_i = (\alpha\sqrt{n})^{\frac{1}{L+1}}$ for $i = 1, ..., K-1$, and $s_K = 0$. Hence Equation (11) can be written as

$$(\alpha\sqrt{n})U_L[D', 0_{K \times (n-1)K}]V_0^T = (\alpha\sqrt{n})U_L \left( D' \otimes (e_1^{(n)})^T \right) V_0^T.$$

Using the remaining freedom in specifying the SVDs (as discussed in the proof of Lemma 1), we can choose $V_0 = Q \otimes R$.

Now consider the intermediate feature matrices $H_l = W_{l-1}...W_1 H_1$. Using the SVD structure from Lemma 1, we compute:

$$H_l = U_{l-1} \Sigma U_{l-2}^T ... U_1 \Sigma U_0^T U_0 \bar{\Sigma} V_0^T = U_{l-1} \Sigma^{l-1} \bar{\Sigma} V_0^T.$$

Notice that $\Sigma^{l-1}\bar{\Sigma} = (\alpha\sqrt{n})^{\frac{l}{L+1}} \tilde{D}^T \otimes (e_1^{(n)})^T$, where $\tilde{D} = [\mathrm{diag}(1, 1, ..., 1, 0), 0_{K \times (d-K)}] \in \mathbb{R}^{K \times d}$. Therefore

$$H_l = (\alpha\sqrt{n})^{\frac{l}{L+1}} U_{l-1} \left( \tilde{D}^T \otimes (e_1^{(n)})^T \right) V_0^T = (\alpha\sqrt{n})^{\frac{l}{L+1}} U_{l-1} \left( \tilde{D}^T Q^T \otimes (e_1^{(n)})^T R^T \right)$$

$$= \frac{1}{\sqrt{n}}(\alpha\sqrt{n})^{\frac{l}{L+1}} U_{l-1} \left( \tilde{D}^T Q^T \otimes 1_n^T \right)$$

$$= \frac{1}{\sqrt{n}}(\alpha\sqrt{n})^{\frac{l}{L+1}} \left( U_{l-1} \tilde{D}^T Q^T \otimes 1_n^T \right),$$

and hence we can write $H_l = M_l \otimes 1_n^T$ as claimed, where $M_l = \frac{1}{\sqrt{n}}(\alpha\sqrt{n})^{\frac{l}{L+1}}(U_{l-1}\tilde{D}^T Q^T)$.

Next note that

$$M_l^T M_l \propto Q\tilde{D}U_{l-1}^T U_{l-1}\tilde{D}^T Q^T = QD'Q^T = S,$$

and hence this matrix does align with a simplex ETF.

Finally, note that up to scale the weight matrix is given by

$$W_l^T \propto U_{l-1}DU_l^T \propto [M_l, 0]O$$

Where $O$ is the orthogonal matrix

$$O = \begin{bmatrix} Q & 0 \\ 0 & I \end{bmatrix} U_l^T$$

and hence if we denote the rows of $W_l$ by $w_j^{(l)}$, they satisfy

$$w_j^{(l)} \propto \sum_{c=1}^K O_{cj}\mu_c^{(l)},$$

meaning they can be written as a linear combination of the columns of $M_l$. Hence, all three properties hold.

We also note the property typically used in the linear setting as DNC3 holds as a consequence of our structure, though this does not generalize to the non-linear case and so was not incorporated as part of the definition:

Using the alternative formulation where $U_L = Q$, we find that

$$W_L...W_l = U_L\tilde{\Sigma}U_{L-1}^T...U_l\Sigma U_{l-1}^T = U_L\tilde{\Sigma}\Sigma^l U_{l-1}^T$$

$$= (\alpha\sqrt{n})^{\frac{L-l+1}{L+1}}U_L\tilde{D}U_{l-1}^T.$$

From which it follows

$$(W_L...W_l)(W_L...W_l)^T \propto U_L\tilde{D}U_{l-1}^T U_{l-1}\tilde{D}^T U_L^T = QD'Q^T = S$$

### B.3 Proof of Theorem 1

We begin by observing that, using the form of any critical point from Lemma 1, the loss at a critical point can be written entirely in terms of the output matrix

$$Z = W_L...W_1 H_1 = U_L \begin{bmatrix} \text{diag}(s_1^{L+1}, ..., s_K^{L+1}) & 0_{K\times(n-1)K} \end{bmatrix} V_0^T.$$

The loss function then becomes

$$\mathcal{L} = g(Z) + \frac{1}{2}(L+1)\lambda \sum_{i=1}^K \tilde{s}_i^{\frac{2}{L+1}},$$

where $\tilde{s}_1 = s_1^{L+1}, ..., \tilde{s}_K = s_K^{L+1}$ are the singular values of $Z$. We will denote this second term by $R(Z)$.

Since our DNC solution already conforms to the structure described in Lemma 1, and we will construct a low-rank solution to obey the same properties, it suffices to compare the loss via this

reduced formulation in terms of $Z$ alone. Note this is the exact same expression as one would have using the Schatten quasi-norm result from [40], but it is for general critical points, not just global minima.

In this formulation the only relevant free parameter in the DNC solution is the scale $\alpha$. To make this explicit we denote a DNC solution by $Z_{\mathrm{DNC}}(\alpha)$. If the optimal scale occurs at $\alpha = 0$, then DNC is not optimal by definition. Thus, we assume the best performing scale satisfies $\alpha > 0$. We will similarly fix the structure in the lower-rank solution so its only free parameter will be its scale $\beta$, and denote this solution by $Z_{\mathrm{LR}}(\beta)$. The method of proof is then to show that the low-rank solution outperforms the DNC solution whenever their scales are set equal (and non-zero by definition), and hence this is true even when the DNC scale is optimized, and so no DNC solution can be optimal.

By definition, the DNC solution has $Z_{\mathrm{DNC}}(\alpha) = \alpha(I_K - 1_K 1_K^T) \otimes 1_n^T$. This matrix has repeated columns, with the matrix formed of the distinct columns having $\frac{K-1}{K}\alpha$ on the diagonal, and $-\frac{1}{K}\alpha$ on the off-diagonal. Hence we can compute the fit-term for this solution, which gives:

$$g(Z_{\mathrm{DNC}}(\alpha)) = \log(1 + (K-1)e^{-\alpha}).$$

In addition the singular values of $Z_{\mathrm{DNC}}(\alpha)$ are $\tilde{s}_1 = ... = \tilde{s}_{K-1} = \alpha\sqrt{n}$, $\tilde{s}_K = 0$. So the regularization loss in this case is given by

$$R(Z_{\mathrm{DNC}}(\alpha)) = \frac{1}{2}(L+1)(K-1)\lambda(\alpha\sqrt{n})^{\frac{2}{L+1}}.$$

We can construct a low-rank solution like so: If $K = 2m$, then we write the matrix $Z_{\mathrm{LR}}(\beta)$ in terms of $2 \times 2$ blocks

$$Z_{\mathrm{LR}}(\beta) = \beta \begin{bmatrix} X & 0 & ... & 0 \\ 0 & X & ... & 0 \\ ... & ... & ... & ... \\ 0 & 0 & ... & X \end{bmatrix} \otimes 1_n^T \ , \ \text{where } X = \begin{bmatrix} 1 & -1 \\ -1 & 1 \end{bmatrix}.$$

If $K = 2m + 1$ we simple use the same matrix as $K = 2m$ but with the extra row and column in the class mean matrix having 1 on its diagonal and 0 otherwise (up to the scaling by $\beta$).

In the even case this matrix has singular values $\tilde{s}_1 = ... = \tilde{s}_m = 2\beta\sqrt{n}$, $\tilde{s}_{m+1} = ... = \tilde{s}_{2m} = 0$, whilst for the odd case it has singular values $\tilde{s}_1 = ... = \tilde{s}_m = 2\beta\sqrt{n}$, $\tilde{s}_{m+1} = \beta\sqrt{n}$, $\tilde{s}_{m+2} = ... = \tilde{s}_{2m+1} = 0$. The fit-term of the loss can be computed in both cases, giving

For $K = 2m$:
$$g(Z_{\mathrm{LR}}(\beta)) = \log\left(1 + (K-2)e^{-\beta} + e^{-2\beta}\right).$$

For $K = 2m + 1$:
$$g(Z_{\mathrm{LR}}(\beta)) = \frac{1}{K}\left[2m\log\left(1 + (K-2)e^{-\beta} + e^{-2\beta}\right) + \log\left(1 + (K-1)e^{-\beta}\right)\right].$$

From this expression we have that if we set the scales equal we arrive at $g(Z_{\mathrm{LR}}(\alpha)) < g(Z_{\mathrm{DNC}}(\alpha))$ for both even and odd $K$. For example, in the even case this follows directly since $e^{-2\alpha} < e^{-\alpha}$ and the fact that $\log$ is an increasing function. The odd case is then a weighted average of the even case and a term that exactly equals the DNC term, and so is still strictly smaller.

We can also compute the regularization loss for our low-rank solutions, giving

$$\text{for } K = 2m: \ R(Z_{\mathrm{LR}}(\beta)) = \frac{1}{2}(L+1)m\lambda(2\beta\sqrt{n})^{\frac{2}{L+1}},$$

$$\text{for } K = 2m + 1: \ R(Z_{\mathrm{LR}}(\beta)) = \frac{1}{2}(L+1)\lambda\left[m(2\beta\sqrt{n})^{\frac{2}{L+1}} + (\beta\sqrt{n})^{\frac{2}{L+1}}\right].$$

Again, if we set the scales equal, then the inequality $R(Z_{\mathrm{LR}}(\alpha)) < R(Z_{\mathrm{DNC}}(\alpha))$ reduces in both cases to:

$$m(2\alpha\sqrt{n})^{\frac{2}{L+1}} < (2m-1)(\alpha\sqrt{n})^{\frac{2}{L+1}},$$

which is equivalent to:

$$2^{\frac{2}{L+1}} < \frac{2m-1}{m}.$$

This equation holds when $m \geq 2, L \geq 3$ or $m \geq 3, L = 2$. equivalently stating this in terms of $K$, this holds when $K \geq 4, L \geq 3$ or $K \geq 6, L = 2$.

We can now put this all together, we have that for the given values of $K$ and $L$

$$\mathcal{L}(Z_{\text{LR}}(\alpha)) < \mathcal{L}(Z_{\text{DNC}}(\alpha)).$$

If we define $\alpha_{\min} = \text{argmin}_\alpha\{\mathcal{L}(Z_{\text{DNC}}(\alpha))\}$, then this implies

$$\min_\beta\{\mathcal{L}(Z_{\text{LR}}(\beta))\} \leq \mathcal{L}(Z_{\text{LR}}(\alpha_{\min})) < \min_\alpha\{\mathcal{L}(Z_{\text{DNC}}(\alpha))\},$$

and so the low-rank solution outperforms the DNC solution, so long as we have $K \geq 4, L \geq 3$ or $K \geq 6, L = 2$.

### Result by Dang et al.

A similar loss function was considered by Dang et al. [6] in Appendix A of their work. The model they study is essentially the same as ours, with the exception that they include a bias vector in the final layer of the network. It is stated, in Theorem A.1 of their work, that at a global optima both $W_L...W_1$ and $H_1$ will align with a simplex ETF shape, $S = I_K - \frac{1}{K}1_K1_K^T$. Whilst it isn't explicitly stated in the theorem, since the product of two simplex ETF's returns a simplex ETF, this implies that $Z$ also forms a simplex ETF at global optima, which appears to contradict with our Theorem 1. Whilst their model is slightly different, they do state that if the regularization term of the bias is non-zero, then the bias itself is zero at optima and so the loss function reduces to the same as ours at global optima.

The proof by Dang et al. proceeds similarly to the proof by Zhu et al. [58] for the $L = 1$ case. Specifically, they produce a series of lower bounds for the loss arriving at an expression like

$$\mathcal{L} \geq \xi(s_1, ..., s_K, \lambda; c_1),$$

where $s_1, ..., s_K$ are the singular values of $H_1$, and $c_1$ is an arbitrary constant. They also show this bound is only attained when $c_1$ takes a specific value in terms of $s_1, ..., s_K$ and constants of the problem, and the matrices $W_L, ..., W_1, H_1$ obey a series of properties that characterize DNC. They then show that the minimum of $\xi$ occurs at finite values of these $s_i$, and so the bound can be attained, and is only done so for a DNC solution.

One issue with this specific bound is that for a simplex ETF the $s_i$ are not freely specified. For the simplex ETF we have $s_K = 0$, and $s_i = \rho$ for $i \neq K$, for some constant $\rho > 0$. If $\xi$ has its minimum at a point where this property does not hold, then this line of argument will not work.

To make this clear, let us assume $c_1$ is written in terms of the $s_1, ..., s_K$ as they detail, and we suppress the $\lambda$ dependence in our notation, so $\xi = \xi(s_1, ..., s_K)$. Let the optimal DNC structure be $\{s_i^{\text{DNC}}\}$, and $\xi_{\text{DNC}} = \xi(s_1^{\text{DNC}}, ..., s_K^{\text{DNC}})$, with the loss of the optimal DNC solution being $\mathcal{L}_{\text{DNC}}$. Denote the global minimal structure for $\xi$ by LR in the same way as we have for DNC. Whilst they show that $\mathcal{L}_{\text{DNC}} = \xi_{\text{DNC}}$, it is entirely possible that $\mathcal{L}_{\text{DNC}} > \mathcal{L}_{\text{LR}}$ if $\xi_{\text{DNC}} > \xi_{\text{LR}}$. In fact, if one considers the structure based on our low-rank solution from Theorem 1, then it can be shown that the minimal low-rank structure has a lower value of $\xi$ than the minimal DNC structure, so this concern actually holds.

In the $L = 1$ case Zhu et al. reduce to a similar expression, but in terms of $\|W\|_F^2$, rather than the $s_i$. Since the Frobenius norm is a freely specified quantity for the DNC solution, the above concern does not arise, and so the $L = 1$ proof avoids this issue.

## B.4  Proof of Theorem 2

Suppose the non-zero singular values of $X$ and $M$ are $\mu_1, ..., \mu_p$ and $\nu_1, ..., \nu_q$ respectively. We choose to write $\beta = \gamma\alpha$. We also write $\mathcal{L}_L$ for our loss, to make the dependence on the hyper-parameter $L$ explicitly clear in this section. The losses of solutions built from these matrices are given by:

$$\mathcal{L}_L(\alpha X) = g(\alpha X) + \frac{1}{2}\lambda(L+1)\alpha^{\frac{2}{L+1}}\sum_{i=1}^{p}\mu_i^{\frac{2}{L+1}},$$

$$\mathcal{L}_L(\gamma\alpha M) = g(\gamma\alpha M) + \frac{1}{2}\lambda(L+1)\alpha^{\frac{2}{L+1}}\sum_{i=1}^{q}(\gamma\nu_i)^{\frac{2}{L+1}}.$$

First We compare the fit term of each solution. We denote the columns of $X$ as $x_{ic}$ ordered by class as previously, and similar for $M$ by $m_{ic}$. The fit terms are given by

$$g(\gamma\alpha M) = \frac{1}{Kn}\sum_{i=1}^{n}\sum_{c=1}^{K}\log\left(1 + \sum_{c'\neq c}^{K}\exp\left(\gamma\alpha((m_{ic})_{c'} - (m_{ic})_c)\right)\right),$$

$$g(\alpha X) = \frac{1}{Kn}\sum_{i=1}^{n}\sum_{c=1}^{K}\log\left(1 + \sum_{c'\neq c}^{K}\exp(\alpha((x_{ic})_{c'} - (x_{ic})_c))\right).$$

Since $M$ is diagonally superior, we know $(m_{ic})_{c'} - (m_{ic})_c < 0$ for all $c, c' \neq c$ and for all $i$. Choose a fixed $\gamma > 0$ so that

$$\gamma((m_{ic})_{c'} - (m_{ic})_c) < (x_{ic})_{c'} - (x_{ic})_c, \quad \text{for all } c, c' \neq c, \text{ and for all } i \in \{1, \ldots, n\}.$$

Hence, due to the monotonicity of $\exp$ and $\log$, for this fixed $\gamma$ we have

$$g(\gamma\alpha M) \leq g(\alpha X),$$

where equality only occurs when $\alpha = 0$.

Next we consider the regularization term. First note we can Taylor expand the exponential, since we are only looking at non-zero singular values:

$$x^{\frac{2}{L+1}} = \sum_{n=0}^{\infty}\frac{1}{n!}\left[\frac{2\log(x)}{L+1}\right]^n = 1 + o(1),$$

where we consider this as an expansion in $L^{-1}$ as $L \to \infty$. Hence, in the large $L$ limit we have

$$\sum_{i=1}^{p}\mu_i^{\frac{2}{L+1}} = p + o(1),$$

$$\sum_{i=1}^{q}(\gamma\nu_i)^{\frac{2}{L+1}} = q + o(1).$$

Since $q < p$ we have that there exists some $L_0 \in \mathbb{N}$, such that for $L > L_0$

$$\sum_{i=1}^{q}(\gamma\nu_i)^{\frac{2}{L+1}} < \sum_{i=1}^{p}\mu_i^{\frac{2}{L+1}},$$

and hence

$$\frac{1}{2}\lambda(L+1)\alpha^{\frac{2}{L+1}}\sum_{i=1}^{q}(\gamma\nu_i)^{\frac{2}{L+1}} \leq \frac{1}{2}\lambda(L+1)\alpha^{\frac{2}{L+1}}\sum_{i=1}^{p}\mu_i^{\frac{2}{L+1}},$$

where again equality only occurs when $\alpha = 0$.

Putting this all together we have that for $\alpha \geq 0$

$$\mathcal{L}_L(\gamma\alpha M) \leq \mathcal{L}_L(\alpha X),$$

with equality only at $\alpha = 0$. Define $\alpha_{\min}^{(L)} = \text{argmin}_{\alpha \geq 0}\{\mathcal{L}_L(\alpha X)\}$. Then we know

$$\min_{\beta \geq 0}\{\mathcal{L}_L(\beta M)\} \leq \mathcal{L}_L(\gamma\alpha_{\min}^{(L)}M) \leq \mathcal{L}_L(\alpha_{\min}^{(L)}X) = \min_{\alpha \geq 0}\{\mathcal{L}_L(\alpha X)\},$$

and equality can only occur if the minimum is at $\alpha = \beta = 0$.

## B.5   Proof of Theorem 3

In this proof we will relabel $L+1 \to L$ and $\frac{1}{2}\lambda \to \lambda$. This change is purely for notational convenience and does not affect the underlying results.

Note, the deep linear UFM must attain its global minimum at a finite-valued optimum, since the regularization term diverges as any parameter tends to infinity. We can define our optimization objective at a critical point for a given depth $L$ as

$$\mathcal{L}_L(Z) = g(Z) + L\lambda_L\sum_{i=1}^{K}\tilde{s}_i^{\frac{2}{L}},$$

where $\{\tilde{s}_i\}_{i=1}^{K}$ are the singular values of $Z$. We begin by proving claim (i).

### Proof of (i)

We consider the case $\lambda_L^{-1} = o(L)$. Since the minimizer of $\mathcal{L}_L$ is unchanged if we re-scale the loss by a constant, we can consider the equivalent objective:

$$\mathcal{L}_L = \frac{1}{L\lambda_L}g(Z_L) + \sum_{i=1}^{L}\tilde{s}_{L,i}^{\frac{2}{L}}.$$

Consider the solution $Z_L = 0$, for which $g(0) = \log(K)$, and all singular values are zero. This solution has loss

$$\mathcal{L}_L(0) = \frac{1}{L\lambda_L}\log(K) = o(1).$$

Denote by $Z_L^*$ an optimal matrix at layer number $L$, and suppose it has $r_L$ many non-zero singular values. This solution must perform at least as well as the zero solution, hence

$$\mathcal{L}_L(Z_L^*) \leq o(1) \implies \sum_{i=1}^{r_L}\tilde{s}_{L,i}^{\frac{2}{L}} \leq o(1) \implies \tilde{s}_{L,i}^{\frac{2}{L}} = o(1).$$

Writing $\tilde{s}_{L,i} = \exp(\frac{1}{2}\gamma_i(L)L)$, for some value $\gamma_i(L)$, we obtain:

$$\exp(\gamma_i(L)) = o(1),$$

and this only occurs if $\gamma_i(L) \to -\infty$. This tells us that each singular value of $Z_L^*$ is either 0, or converges to 0 at a rate faster than $\exp(-\beta L)$ for any $\beta > 0$, meaning faster than exponentially.

**Proof of (ii)**

Now consider the case that $L\lambda_L = o(1)$. We first show that, for sufficiently large $L$, any global minimizer $Z_L^*$ must be diagonally superior.

Let $Z \in \mathbb{R}^{K \times Kn}$ be any matrix that is not diagonally superior. We consider the fit term for this matrix

$$g(Z) = \frac{1}{Kn} \sum_{i=1}^{n} \sum_{c=1}^{K} \log \left( 1 + \sum_{c' \neq c}^{K} \exp((z_{ic})_{c'} - (z_{ic})_c) \right).$$

Since $Z$ is not diagonally superior, there exists a value of $i, c, c' \neq c$ such that $(z_{ic})_{c'} - (z_{ic})_c \geq 0$. In this case we see from the above that we can bound the fit term below

$$g(Z) \geq \frac{1}{Kn} \log(2),$$

and the whole loss is then also bounded

$$\mathcal{L}_L(Z) \geq \frac{1}{Kn} \log(2).$$

Now let $Z' \in \mathbb{R}^{K \times Kn}$ be any diagonally superior matrix, and denote its singular values $\{s_i'\}_{i=1}^{K}$. Consider the sequence of solutions for each $L$

$$\mathcal{L}_L(LZ') = g(LZ') + L\lambda_L L^{\frac{2}{L}} \sum_{i=1}^{K} (s_i')^{\frac{2}{L}}.$$

First looking at the fit term we have $(z_{ic}')_{c'} - (z_{ic}')_c < 0$ for all $i, c, c' \neq c$. Hence

$$g(LZ') = \frac{1}{Kn} \sum_{i=1}^{n} \sum_{c=1}^{K} \log \left( 1 + \sum_{c' \neq c}^{K} \exp(L((z_{ic}')_{c'} - (z_{ic}')_c)) \right) = \frac{1}{K} \sum_{c=1}^{K} \log(1 + o(1)) = o(1).$$

In addition, $L^{\frac{2}{L}} = 1 + o(1)$ and $\sum_{i=1}^{K} (s_i')^{\frac{2}{L}} = \text{rank}(Z') + o(1)$. Hence

$$\mathcal{L}_L(LZ') = o(1) + L\lambda_L(1 + o(1))(\text{rank}(Z') + o(1)) = o(1).$$

This sequence attains arbitrarily small loss. In particular, when $L$ is large enough that this loss is less than $\frac{1}{Kn} \log(2)$, it outperforms any matrix that is not diagonally superior. Hence for $L$ larger than this the optimal $Z$ must be diagonally superior.

It remains to show the condition on the singular values of a normalized optimal solution $\hat{Z}_L^*$.

Let $X \in \mathbb{R}^{K \times Kn}$ be a lowest rank diagonally superior matrix, and label its rank by $q_K$. Choose its scale so that we have for all $i, c, c' \neq c$ that $(x_{ic})_{c'} - (x_{ic})_c < -2$. Suppose the optimal solution at each $L$ is $Z_L^*$. We can seperate the scale of $Z_L^*$ from its frame by writing

$$Z_L^* = \alpha_L \hat{Z}_L^*,$$

where $\alpha_L = \|Z_L^*\|_F$, $\hat{Z}_L^* = \frac{Z_L^*}{\|Z_L^*\|_F}$. Note in particular since the zero matrix is not diagonally superior we have that the scale is non-zero and this decomposition is well defined when $L$ large enough.

Note the elements of $\hat{Z}_L^*$ are constrained such that $|(\hat{Z}_L^*)_{ij}| \le 1$, and so for all $i, c, c' \ne c$ we have

$$(\hat{z}_{L,ic}^*)_{c'} - (\hat{z}_{L,ic}^*)_c \ge -2 > (x_{ic})_{c'} - (x_{ic})_c.$$

This means that for $L$ large enough

$$g(\alpha_L \hat{Z}_L^*) > g(\alpha_L X).$$

Now compare the losses of the sequence $Z_L^*$ with the sequence $\alpha_L X$. Denoting the rank of $\hat{Z}_L^*$ by $r_L$, its non-zero singular values by $\hat{s}_{L,i}$, and the non-zero singular values of $X$ by $\mu_i$, we have

$$\mathcal{L}_L(\alpha_L \hat{Z}_L^*) = g(\alpha_L \hat{Z}_L^*) + L\lambda \alpha_L^{\frac{2}{L}} \sum_{i=1}^{r_L} \hat{s}_{L,i}^{\frac{2}{L}},$$

$$\mathcal{L}_L(\alpha_L X) = g(\alpha_L X) + L\lambda \alpha_L^{\frac{2}{L}} \sum_{i=1}^{q_K} \mu_i^{\frac{2}{L}}.$$

Using our inequality for the fit term, and the fact that $Z_L^*$ is the minimum, we have that

$$L\lambda \alpha_L^{\frac{2}{L}} \sum_{i=1}^{r_L} \hat{s}_{L,i}^{\frac{2}{L}} < L\lambda \alpha_L^{\frac{2}{L}} \sum_{i=1}^{q_K} \mu_i^{\frac{2}{L}},$$

and hence

$$\sum_{i=1}^{r_L} \hat{s}_{L,i}^{\frac{2}{L}} < \sum_{i=1}^{q_K} \mu_i^{\frac{2}{L}} = q_K + o(1).$$

Again write $\hat{s}_{L,i} = \exp(\frac{1}{2}\gamma_i(L)L)$, for some function $\gamma_i(L)$, we have

$$\sum_{i=1}^{r_L} \exp(\gamma_i(L)) < q_K + o(1).$$

Now define the quantity $\delta > 0$ to be such that $e^{-\delta} = \frac{q_K+1}{q_K+2}$.

Suppose the number of $\gamma_i(L)$ that are greater than $-\delta$ is $p_L$, then

$$\sum_{i=1}^{r_L} \exp(\gamma_i(L)) \ge p_L e^{-\delta},$$

and hence

$$p_L e^{-\delta} < q_K + o(1) \implies p_L(q_K + 1) < q_K(q_K + 2) + o(1).$$

When the $o(1)$ term is less than $\frac{1}{2}$, this equation is only satisfied if $p_L \le q_K$. This is since $p_L, q_K$ are both integers, $p_L(q_K + 1)$ is an increasing function in $p_L$, and $(q_K + 1)^2 > q_K(q_K + 2) + \frac{1}{2}$. Hence, when $L$ is large enough, all but at most $q_K$ singular value of $\tilde{Z}_L^*$ satisfy

$$0 \le \hat{s}_{L,i} \le \exp\left(-\frac{1}{2}\delta L\right),$$

where $\delta > 0$ is independent of $L$. This means that these singular values are either 0, or converge to 0 at a rate at least exponential in $L$.

## B.6 Proof of Theorem 4

We begin by showing that DNC solutions at a specific scale satisfy the critical point conditions of the deep linear UFM when the regularization parameter $\lambda > 0$ is sufficiently small. Recall from Equation (7) in Appendix B.1 that the first order derivatives of the loss are given by:

$$\frac{\partial \mathcal{L}}{\partial W_l} = \frac{1}{Kn} A_{l+1}^T (P - Y) H_l^T + \lambda W_l,$$

where the matrices $A_{l+1}, P, Y$ and $H_l$ are defined at the start of Appendix B. Hence, the critical point condition is for $0 \leq l \leq L$:

$$W_l = \frac{1}{Kn\lambda} A_{l+1}^T (Y - P) H_l^T. \tag{12}$$

Recall our DNC solution satisfies, for some $\alpha > 0$:

$$Z = \alpha S \otimes 1_n^T, \quad \text{where } S := I_K - \frac{1}{K} 1_k 1_K^T.$$

As a consequence, the softmax probability matrix $P$ is given by

$$P = \frac{1}{(K-1) + e^\alpha} \begin{bmatrix} e^\alpha & 1 & ... & 1 \\ 1 & e^\alpha & ... & 1 \\ ... & ... & ... & ... \\ 1 & 1 & ... & e^\alpha \end{bmatrix} \otimes 1_n^T. \tag{13}$$

We can then compute the error matrix $Y - P$, which gives

$$Y - P = \frac{K}{(K-1) + e^\alpha} (S \otimes 1_n^T), \tag{14}$$

and so we see the error matrix also aligns with a simplex ETF.

We will now start to simplify our critical point conditions in Equation (12), starting with $1 \leq l \leq L - 1$, by writing each matrix in terms of its SVD. Recall that we define the following matrices $D \in \mathbb{R}^{d \times d}, \tilde{D} \in \mathbb{R}^{K \times d}, \bar{D} \in \mathbb{R}^{d \times Kn}, D' \in \mathbb{R}^{K \times K}$ where each has its top $K \times K$ block given by $\text{diag}(1, 1, ..., 1, 0)$, with all other entries being 0.

First write the singular value decomposition of the simplex matrix:

$$S \otimes 1_n^T = \sqrt{n} Q \tilde{D} \bar{D} R^T,$$

where $\tilde{D} \bar{D} \in \mathbb{R}^{K \times Kn}$, and $Q, R$ are orthogonal matrices. We also have for the DNC solution that $U_L = Q, V_0 = R$.

Using the definition of $A_{l+1}$ and $H_l$, as well as Lemma 1, we have for $1 \leq l \leq L - 1$:

$$W_l = (\alpha\sqrt{n})^{\frac{1}{L+1}} U_l D U_{l-1}^T, \tag{15}$$

$$A_{l+1} = U_L \tilde{\Sigma} \Sigma^{L-l-1} U_l^T = (\alpha\sqrt{n})^{\frac{L-l}{L+1}} U_L \tilde{D} U_l^T, \tag{16}$$

$$H_l = U_{l-1} \Sigma^{l-1} \bar{\Sigma} V_0^T = (\alpha\sqrt{n})^{\frac{l}{L+1}} U_{l-1} \bar{D} V_0^T, \tag{17}$$

where we used that $\Sigma, \tilde{\Sigma}$ and $\bar{\Sigma}$ are written in terms of the singular values of the parameter matrices, and for DNC the singular values are $s_1 = ... = s_{K-1} = (\alpha\sqrt{n})^{\frac{1}{L+1}}, s_K = 0$.

Hence our critical point condition in Equation (12) for $1 \leq l \leq L - 1$ becomes

$$(\alpha\sqrt{n})^{\frac{1}{L+1}}U_lDU_{l-1}^T = (\alpha\sqrt{n})^{\frac{L}{L+1}}\frac{1}{n\lambda}\frac{\sqrt{n}}{(K-1)+e^\alpha}(U_l\tilde{D}^TU_L^TQ\tilde{D}\bar{D}R^TV_0\bar{D}^TU_{l-1}^T).$$

We can pre and post multiply by $U_l^T$, $U_{l-1}$ respectively, and use that $U_L^TQ = R^TV_0 = I$ to give

$$D = (\alpha\sqrt{n})^{\frac{L-1}{L+1}}\frac{1}{n\lambda}\frac{\sqrt{n}}{(K-1)+e^\alpha}(\tilde{D}^T\tilde{D}\bar{D}\bar{D}^T).$$

Then using that $\tilde{D}^T\tilde{D}\bar{D}\bar{D}^T = D$, this reduces to

$$D = (\alpha\sqrt{n})^{\frac{L-1}{L+1}}\frac{1}{n\lambda}\frac{\sqrt{n}}{(K-1)+e^\alpha}D, \tag{18}$$

and this holds when

$$\lambda n^{\frac{1}{L+1}} = \frac{1}{(K-1)+e^\alpha}\alpha^{\frac{L-1}{L+1}}. \tag{19}$$

If we write $f(\alpha)$ for the RHS of Equation (19), this function satisfies: $f(0) = 0$, $\lim_{\alpha\to\infty}\{f(\alpha)\} = 0$, and $f(\alpha) > 0$ for $\alpha > 0$. In addition, $f$ has a unique maximum in $\alpha > 0$, since

$$f'(\alpha) = 0 \quad \Leftrightarrow \quad \left(\frac{L-1}{L+1}\right)(K-1) = \left[\alpha - (\frac{L-1}{L+1})\right]e^\alpha,$$

which has exactly one solution in $\alpha > 0$.

Therefore Equation (19) has two positive solutions when $\lambda n^{\frac{1}{L+1}}$ is smaller than the value of the maximum. One solution satisfies $\alpha \to 0$ as $\lambda \to 0$. This solution has a high loss in this limit and lies very close to the zero solution, so we won't consider it further. The other solution has $\alpha \to \infty$ as $\lambda \to 0$, which attains arbitrarily small loss in this limit.

This provides a solution that satisfies the critical point conditions for $1 \le l \le L-1$. The $l = L, l = 0$ cases produce the same equation as Equation (18), but with $\tilde{D}$ or $\bar{D}$ in place of $D$, which reduces to Equation (19) also.

Hence we have shown that there are DNC solutions that are critical points of the model, so long as $\lambda$ is suitably small, with a scale that diverge as $\lambda \to 0$.

It remains to show the positive semi-definiteness of the Hessian of this solution. We start by computing the second-order derivatives of the loss. We then specialize to the DNC case. We will begin with the block diagonal terms. Recalling the form of the first-order derivative from Equation (7), we find

$$\frac{\partial^2\mathcal{L}}{\partial(W_l)_{ab}\partial(W_l)_{cd}} = \lambda\delta_{ac}\delta_{bd} + \frac{1}{Kn}\sum_{x,y}(A_{l+1}^T)_{ax}\frac{\partial P_{xy}}{\partial(W_l)_{cd}}(H_l^T)_{yb}.$$

By the chain rule, we have:

$$\frac{\partial P_{xy}}{\partial(W_l)_{cd}} = \sum_{uv}\frac{\partial P_{xy}}{\partial Z_{uv}}\frac{\partial Z_{uv}}{\partial(W_l)_{cd}}.$$

The partial derivative of the logit matrix $Z$ was calculated in Equation (6). It remains to compute the $P$ partial derivative.

$$\frac{\partial P_{xy}}{\partial Z_{uv}} = \frac{\exp(Z_{xy})}{\sum_{c'}\exp(Z_{c'y})}\delta_{xu}\delta_{yv} - \frac{\exp(Z_{xy})}{[\sum_{c'}\exp(Z_{c'y})]^2}\sum_{c''}\exp(Z_{c''y})\delta_{uc''}\delta_{vy}$$

$$= P_{xy}\delta_{vy}[\delta_{xu} - P_{uy}].$$

If we choose to write $P_{xy} = (p_y)_x$, so that $p_y$ is the $y^{\text{th}}$ column of $P$, then

$$\delta_{xu}(p_y)_x = \text{diag}(p_y)_{xu}, \quad (p_y)_x(p_y)_u = (p_y p_y^T)_{xu},$$

giving

$$\frac{\partial P_{xy}}{\partial Z_{uv}} = \delta_{vy}[\text{diag}(p_y) - p_y p_y^T]_{xu}.$$

Hence

$$\frac{\partial^2 \mathcal{L}}{\partial(W_l)_{ab}\partial(W_l)_{cd}} = \lambda\delta_{ac}\delta_{bd} + \frac{1}{Kn}\sum_{x,y,u,v}(A_{l+1}^T)_{ax}(H_l^T)_{yb}(A_{l+1})_{uc}(H_l)_{dv}\delta_{vy}[\text{diag}(p_y)-p_y p_y^T]_{xu}$$

$$= \lambda\delta_{ac}\delta_{bd} + \frac{1}{Kn}\sum_y(A_{l+1}^T[\text{diag}(p_y) - p_y p_y^T]A_{l+1})_{ac}(H_l)_{by}(H_l)_{dy}.$$

We also choose to write $(H_l)_{by} = (h_y^{(l)})_b$, so that $h_y^{(l)}$ is column $y$ of $H_l$. This gives

$$\text{Hess}_{abcd}^{(l,l)} = \frac{\partial^2 \mathcal{L}}{\partial(W_l)_{ab}\partial(W_l)_{cd}} = \lambda\delta_{ac}\delta_{bd} + \frac{1}{Kn}\sum_y(A_{l+1}^T[\text{diag}(p_y) - p_y p_y^T]A_{l+1})_{ac}(h_y^{(l)}h_y^{(l)T})_{bd}.$$

(20)

Next we compute the off-diagonal blocks. Taking wlog $r < l$, we have

$$\frac{\partial^2 \mathcal{L}}{\partial(W_l)_{ab}\partial(W_r)_{cd}} = \frac{1}{Kn}\sum_{xy}\left\{(A_{l+1}^T)_{ax}(P-Y)_{xy}\frac{\partial(H_l^T)_{yb}}{\partial(W_r)_{cd}} + (A_{l+1}^T)_{ax}\frac{\partial P_{xy}}{\partial(W_r)_{cd}}(H_l^T)_{yb}\right\}.$$

The second of these two terms we can compute in exactly the same way we did for the second term of the diagonal blocks, giving

$$\frac{1}{Kn}\sum_{xy}(A_{l+1}^T)_{ax}\frac{\partial P_{xy}}{\partial(W_r)_{cd}}(H_l^T)_{yb} = \frac{1}{Kn}\sum_y[A_{l+1}^T(\text{diag}(p_y) - p_y p_y^T)A_{r+1}]_{ac}[h_y^{(l)}h_y^{(r)T}]_{bd}.$$

We now compute the first term. Note that

$$\frac{\partial(H_l^T)_{yb}}{\partial(W_r)_{cd}} = \sum_{u,v}[W_{r-1}...W_1 H_1]_{yu}^T\frac{\partial(W_r^T)_{uv}}{\partial(W_r)_{cd}}[W_{l-1}...W_{r+1}]_{vb}^T$$

$$= (H_r^T)_{yd}[(W_{l-1}...W_{r+1})^T]_{cb},$$

and so the first term is

$$\frac{1}{K}\sum_{x,y}(A_{l+1}^T)_{ax}(P-Y)_{xy}\frac{\partial(H_l^T)_{yb}}{\partial(W_r)_{cd}} = \frac{1}{Kn}(A_{l+1}^T(P-Y)H_r^T)_{ad}[(W_{l-1}...W_{r+1})^T]_{cb}.$$

Combining both terms, we arrive at for $r < l$:

$$\text{Hess}^{(l,r)}_{abcd} = \frac{1}{Kn}(A^T_{l+1}(P-Y)H^T_r)_{ad}[(W_{l-1}...W_{r+1})^T]_{cb}+$$

$$\frac{1}{Kn}\sum_y[A^T_{l+1}(\text{diag}(p_y)-p_yp^T_y)A_{r+1}]_{ac}[h^{(l)}_y h^{(r)T}_y]_{bd}. \tag{21}$$

This gives the general forms of the blocks of the Hessian matrix. We now show that the Hessian of the scale divergent DNC solution has no negative eigenvalues when $\lambda$ is small enough. To show this we can demonstrate that

$$\forall v \in \mathbb{R}^p \quad v^T\text{Hess}\, v \geq 0,$$

where Hess is the full Hessian after the blocks have been flattened into matrices, and $p$ is the total number of parameters. Writing $v^T = (v_0, ..., v_L)^T$, where each component is of appropriate dimensions, this is equivalent to

$$\sum_{l,r} v^T_l\text{Hess}^{(l,r)}v^T_r \geq 0.$$

Denoting the matrix $X_l$ to be the pre-flattened version of $v_l$, this can be written as

$$\sum_{l,r}\sum_{abcd}(X_l)_{ab}\text{Hess}^{(l,r)}_{abcd}(X_r)_{cd} \geq 0, \tag{22}$$

or equivalently

$$\sum_{l}\sum_{abcd}(X_l)_{ab}\text{Hess}^{(l,l)}_{abcd}(X_l)_{cd} + 2\sum_{r<l}\sum_{abcd}(X_l)_{ab}\text{Hess}^{(l,r)}_{abcd}(X_r)_{cd} \geq 0.$$

Both summations have two terms due to the form of the on and off diagonal layer-wise Hessians described in Equations (20) and (21). We will make a series of transformations of the matrices $X_l$ that leave the range of these matrices as $X_l$ varies unchanged, and show how this impacts each term individually.

Starting with the off-diagonal terms, and working with the second term arising from Equation (21), this gives a contribution

$$2\sum_{r<l}\sum_y \frac{1}{Kn}\text{Tr}(A^T_{l+1}\rho_y A_{r+1}X_r h^{(r)}_y h^{(l)T}_y X^T_l), \tag{23}$$

where we now choose to denote $\rho_y = \text{diag}(p_y) - p_yp^T_y$.

The first transformation is $X_l \rightarrow U_lX_lU^T_{l-1}$. Since the matrices $U_l$ are invertible this does not affect the validity of the inequality. Doing this, and using the expressions for $A_{l+1}, H_l$ in Equations (16) and (17), transforms Equation (23) into:

$$2\sum_{r<l}\sum_y \frac{1}{Kn}(\alpha\sqrt{n})^{\frac{2L}{L+1}}\text{Tr}(\tilde{D}^TU^T_L\rho_yU_L\tilde{D}X_r\bar{D}V^T_0 e_y e^T_y V_0\bar{D}^T X^T_l),$$

where $e_y$ are the standard basis vectors in $\mathbb{R}^{Kn}$. Now we perform the transformation $X_l \rightarrow O^TX_lO$, where

$$O = \begin{bmatrix} U_L & 0 \\ 0 & I \end{bmatrix}.$$

In this expression the dimensions of the $0$ and $I$ blocks are designed to match the dimensions of $X_l$. Again this is invertible. Using that

$$OD\tilde{D}^T U_L^T = \begin{bmatrix} S \\ 0 \end{bmatrix}, \quad OD\bar{D}V_0^T = \begin{bmatrix} S \otimes 1_n^T \\ 0 \end{bmatrix},$$

our contributing term becomes:

$$2\sum_{r<l}\sum_{y}\frac{1}{Kn}(\alpha\sqrt{n})^{\frac{2L}{L+1}}\mathrm{Tr}\left(\begin{bmatrix} S \\ 0 \end{bmatrix}\rho_y\begin{bmatrix} S & 0 \end{bmatrix}X_r\begin{bmatrix} S \otimes 1_n^T \\ 0 \end{bmatrix}e_y e_y^T\begin{bmatrix} S \otimes 1_n^T & 0 \end{bmatrix}X_l^T\right)$$

$$= 2\sum_{r<l}\sum_{y}\frac{1}{Kn}(\alpha\sqrt{n})^{\frac{2L}{L+1}}\mathrm{Tr}\left(\begin{bmatrix} S\rho_y S & 0 \\ 0 & 0 \end{bmatrix}X_r\begin{bmatrix} (S \otimes 1_n^T)e_y e_y^T(S \otimes 1_n) & 0 \\ 0 & 0 \end{bmatrix}X_l^T\right)$$

$$= 2\sum_{r<l}\sum_{y}\frac{1}{Kn}(\alpha\sqrt{n})^{\frac{2L}{L+1}}\mathrm{Tr}(S\rho_y SX_r'(S \otimes 1_n^T)e_y e_y^T(S \otimes 1_n)(X_l')^T),$$

where we define the top $K \times K$ block of $X_l$ to be $X_l'$. Hence only the top $K \times K$ block of the transformed matrices emerges in the expression. These steps are exactly the same for the the second term in the on-diagonal block, just with $l = r$.

The only remaining difficult term is the first term in the off diagonal Hessian. This is given, before transformations, by

$$2\sum_{r<l}\frac{1}{Kn}\mathrm{Tr}(A_{l+1}^T(P - Y)H_r^T X_r^T(W_{l-1}...W_{r+1})^T X_l^T).$$

Performing both transformations, and using that the matrix $P - Y$ is proportional to $S \otimes 1_n^T = \sqrt{n}U_L\tilde{D}\bar{D}V_0^T$ as stated in Equation (14), and $W_l$ has SVD given by Equation (15), a similar calculation to previous gives that this term is

$$-2\sum_{r<l}\frac{1}{\sqrt{n}}\frac{1}{K - 1 + e^\alpha}(\alpha\sqrt{n})^{\frac{L-1}{L+1}}\mathrm{Tr}(S(X_r')^T S(X_l')^T),$$

and so this term also only features the top $K \times K$ block. We also identify the coefficient in the sum as $\lambda$ via Equation (19), reducing this to:

$$-2\lambda\sum_{r<l}\mathrm{Tr}(S(X_r')^T S(X_l')^T).$$

This gives us finally the expression for the positive semi-definite condition of the Hessian, stated in Equation (22), is equivalent to:

$$\sum_l \lambda\mathrm{Tr}(X_l^T X_l) + \frac{1}{Kn}(\alpha\sqrt{n})^{\frac{2L}{L+1}}\sum_l\sum_y\mathrm{Tr}(S\rho_y SX_l'(S \otimes 1_n^T)e_y e_y^T(S \otimes 1_n)(X_l')^T)$$

$$-2\lambda\sum_{r<l}\mathrm{Tr}(S(X_r')^T S(X_l')^T) + \frac{2}{Kn}(\alpha\sqrt{n})^{\frac{2L}{L+1}}\sum_{r<l}\sum_y\mathrm{Tr}(S\rho_y SX_r'(S\otimes 1_n^T)e_y e_y^T(S\otimes 1_n)(X_l')^T) \geq 0.$$

The only place the entries not belonging to the top $K \times K$ block of $X_l$ contribute is in the first term, and they clearly contribute a quantity that is non-negative, hence we can drop them and work with the reduced inequality. We do this, and now drop the primes from the matrices $X_l'$, using $X_l$ to denote just the top $K \times K$ block. In addition, notice the second term in both lines are the same and so can be combined into a sum over all $r, l$. This reduces our positive semi-definiteness condition to

$$\sum_l \lambda\mathrm{Tr}(X_l^T X_l) + \frac{1}{Kn}(\alpha\sqrt{n})^{\frac{2L}{L+1}}\sum_{l,r}\sum_y\mathrm{Tr}(S\rho_y SX_r(S\otimes 1_n^T)e_y e_y^T(S\otimes 1_n)X_l^T) - 2\lambda\sum_{r<l}\mathrm{Tr}(SX_r^T SX_l^T) \geq 0.$$

Now note that the sum over $y = 1, ..., Kn$ can be replaced by a sum over $i = 1, ..., n$ and $c = 1, ...K$. Additionally, we have

$$\rho_{ic} = \rho_c, \quad (S \otimes 1_n^T)e_{ic} = s_c,$$

where here we use $\rho_c = \text{diag}(p_c) - p_c p_c^T$, where $p_c$ are the softmax probabilities of the network output means, and $s_c$ are the columns of the matrix $S$ (this is a departure from the rest of the document where we use $s_i$ to denote singular values). Using this our condition simplifies to:

$$\sum_l \lambda \text{Tr}(X_l^T X_l) + \frac{1}{K}(\alpha\sqrt{n})^{\frac{2L}{L+1}} \sum_{l,r} \sum_{c=1}^K \text{Tr}(S\rho_c SX_r s_c s_c^T X_l^T) - 2\lambda \sum_{r<l} \text{Tr}(SX_r^T SX_l^T) \geq 0.$$

Defining the matrix $X = \sum_l X_l$, this then reduces further to

$$\sum_l \lambda \text{Tr}(X_l^T X_l) + \frac{1}{K}(\alpha\sqrt{n})^{\frac{2L}{L+1}} \sum_c \text{Tr}(S\rho_c SX s_c s_c^T X^T) - 2\lambda \sum_{r<l} \text{Tr}(SX_r^T SX_l^T) \geq 0.$$

We can perform a similar trick with the final term, but now we must add the missing diagonal terms, giving us that our positive semi-definite condition is:

$$\sum_l \lambda \text{Tr}(X_l^T X_l) + \lambda \sum_l \text{Tr}(SX_l^T SX_l^T) + \frac{1}{K}(\alpha\sqrt{n})^{\frac{2L}{L+1}} \sum_c \text{Tr}(S\rho_c SX s_c s_c^T X^T) - \lambda \text{Tr}(SX^T SX^T) \geq 0. \tag{24}$$

We first show that the first two terms together are non-negative. Note we can use $I = S + \frac{1}{K}1_K 1_K^T$ to write the first term as:

$$\lambda \sum_l \text{Tr}(SX_l SX_l^T) + \text{Tr}(1_K 1_K^T X_l 1_K 1_K^T X_l^T) + \text{Tr}(SX_l 1_K 1_K^T X_l^T) + \text{Tr}(1_K 1_K^T X_l SX_l^T).$$

The last three terms are all non-negative, since each can be written as the Frobenius norm of a matrix. Hence to show the sum of the first two terms in Equation (24) is non-negative it is sufficient to show the following is non-negative:

$$\sum_l \lambda \text{Tr}(SX_l^T SX_l) + \lambda \sum_l \text{Tr}(SX_l^T SX_l^T)$$

$$= \lambda \sum_l \text{Tr}\left(SX_l^T S(SX_l S + SX_l^T S)\right),$$

where we used that $S^2 = S$. Note the right matrix in the trace is the symmetrization of the transpose of the left matrix (up to a factor of two). Since the trace of a symmetric matrix times an antisymmetric matrix is zero, this just becomes equal to the sum over $l$ of the Frobenius norms squared of the symmetric part of $2\lambda SX_l S$, and hence is clearly non-negative.

It only remains to show that the last two terms in Equation (24) are non-negative. For this we need to compute $\rho_c$. We detail the $c = 1$ case, the general case can be inferred from this. Using Equation (13), we have

$$
\rho_1 = \frac{1}{(K-1+e^\alpha)^2} \left( e^\alpha \underbrace{\begin{bmatrix} K-1 & -1 & -1 & \dots & -1 \\ -1 & 1 & 0 & \dots & 0 \\ -1 & 0 & 1 & \dots & 0 \\ \dots & \dots & \dots & \dots & \dots \\ -1 & 0 & 0 & \dots & 1 \end{bmatrix}}_{\rho_1'} + \underbrace{\begin{bmatrix} 0 & 0 & 0 & \dots & 0 \\ 0 & K-2 & -1 & \dots & -1 \\ 0 & -1 & K-2 & \dots & -1 \\ \dots & \dots & \dots & \dots & \dots \\ 0 & -1 & -1 & \dots & K-2 \end{bmatrix}}_{\rho_1''} \right).
$$

$$(25)$$

Label the first matrix by $\rho_1'$ and the second by $\rho_1''$. For general $y$ we get a matrix similar in both cases, but crucially $\rho_c''$ is always positive semi-definite. We hence have that the remaining terms in our positive semi-definite condition are:

$$
-\lambda \mathrm{Tr}(SX^TSX^T) + \frac{1}{K}(\alpha\sqrt{n})^{\frac{2L}{L+1}} e^\alpha \frac{1}{(K-1+e^\alpha)^2} \sum_c \mathrm{Tr}(S\rho_c' SX s_c s_c^T X^T)
$$

$$
+ \frac{1}{K}(\alpha\sqrt{n})^{\frac{2L}{L+1}} \frac{1}{(K-1+e^\alpha)^2} \sum_c \mathrm{Tr}(S\rho_c'' SX s_c s_c^T X^T) \geq 0.
$$

Since $\rho_c''$ is positive semi-definite, this last term is non-negative and can be dropped from the expression. We also can use the expression for $\lambda$ in Equation (19) to simplify the coefficient, leaving us with the following condition

$$
-\lambda \mathrm{Tr}(SX^TSX^T) + \frac{1}{K}\alpha n\lambda \frac{e^\alpha}{K-1+e^\alpha} \sum_c \mathrm{Tr}(S\rho_c' SX s_c s_c^T X^T) \geq 0.
$$

We now use Lemma 2, detailed at the end of the section, which gives us that $S\rho_c'S = Ks_c s_c^T + S$. Also using that $\sum_c s_c s_c^T = S$, the above condition becomes:

$$
-\lambda \mathrm{Tr}(SX^TSX^T) + \frac{1}{K}\alpha n\lambda \frac{e^\alpha}{K-1+e^\alpha} \mathrm{Tr}(SXSX^T) + \alpha n\lambda \frac{e^\alpha}{K-1+e^\alpha} \sum_c \mathrm{Tr}(s_c s_c^T X s_c s_c^T X^T) \geq 0.
$$

Since $s_c s_c^T$ is positive semi-definite, the last term is again non-negative and can be dropped. The remaining two terms can be written as:

$$
\lambda \left( \frac{\alpha n}{K} \frac{e^\alpha}{K-1+e^\alpha} - 1 \right) \mathrm{Tr}(SXSX^T) + \lambda \mathrm{Tr}(SX^TS(SXS - SX^TS)) \geq 0.
$$

A similar argument by considering the symmetric and antisymmetric parts of the matrix $SXS$ shows that the last term is up to a positive scale equal to the Frobenius norm squared of the antisymmetric part of $SXS$, which is always non-negative. The first term is non-negative, so long as we have:

$$
\frac{\alpha n}{K} \frac{e^\alpha}{K-1+e^\alpha} \geq 1.
$$

Since $\alpha$ diverges as $\lambda \to 0$, this holds when $\lambda$ is small enough.

Hence there exists a $\lambda_0$ such that when $\lambda < \lambda_0$ there is a DNC solution that is an critical point of the model, with scale that diverges as $\lambda \to 0$, and with Hessian that is positive semi-definite.

It only remains to prove the following Lemma:

**Lemma 2:** *The matrix $\rho_c'$, as defined in Equation* (25)*, satisfies*

$$
S\rho_c'S = Ks_c s_c^T + S,
$$

where $S = I_K - \frac{1}{K}1_K1_K^T$, and $s_c$ is column $c$ of the matrix $S$.

**Proof:** Note that the entries of $\rho_c'$ are given by

$$(\rho_c')_{ij} = \begin{cases} K - 1, & \text{if } i = j = c \\ 1, & \text{if } i = j \neq c \\ -1, & \text{if } i = c, \text{ or } j = c, \text{ but } i \neq j \\ 0, & \text{otherwise} \end{cases}.$$

Similarly the entries of $K s_c s_c^T$ and $S$ are given by

$$(K s_c s_c^T)_{ij} = \begin{cases} \frac{(K-1)^2}{K}, & \text{if } i = j = c \\ -\frac{K-1}{K}, & \text{if } i = c, \text{ or } j = c, \text{ but } i \neq j \\ \frac{1}{K}, & \text{otherwise} \end{cases},$$

$$S_{ij} = \begin{cases} \frac{K-1}{K}, & \text{if } i = j \\ -\frac{1}{K}, & \text{otherwise} \end{cases}.$$

Looking at this case by case, it is clear that $\rho_c' = K s_c s_c^T + S$. Since the columns of $S$ are invariant under transformation by $S$, we have $S\rho_c'S = \rho_c'$.

### B.7   Proof of Theorem 5

Recall, from Lemma 1, we have the following SVDs for our parameter matrices at a critical point:

$W_l = U_l \Sigma U_{l-1}^T$, for $l = 1, ..., L - 1$,

$W_L = U_L \tilde{\Sigma} U_{L-1}^T$,

$H_1 = U_0 \bar{\Sigma} V_0^T$,

where $U_L \in \mathbb{R}^{K \times K}$, $V_0 \in \mathbb{R}^{Kn \times Kn}$, $U_{L-1}, ..., U_0 \in \mathbb{R}^{d \times d}$ are all orthogonal matrices, $\Sigma \in \mathbb{R}^{d \times d}$, $\tilde{\Sigma} \in \mathbb{R}^{K \times d}$ and $\bar{\Sigma} \in \mathbb{R}^{d \times Kn}$ all have their top $K \times K$ block given by $\text{diag}(s_1, ..., s_K)$, with all other entries being zero. Also, as a consequence, $Z = W_L ... W_1 H_1$ has the form:

$$Z = W_L ... W_1 H_1 = U_L \begin{bmatrix} \text{diag}(s_1^{L+1}, ..., s_K^{L+1}) & 0_{K \times (n-1)K} \end{bmatrix} V_0^T.$$

Suppose we have some structure $Z^*$ which is a critical point for some specific choice of scale, and we want to assess the dimension of the space of solutions that correspond to this structure. Let us denote the rank of this critical point by $\text{rank}(Z^*) = r$. First note that specifying $Z^*$ fully determines $s_1, ..., s_K$, and as a consequence $\Sigma, \tilde{\Sigma}$ and $\bar{\Sigma}$. It also specifies $U_L, V_0$, up to some potential reparametrization of the singular spaces that we will account for later. Also note, for intuition purposes, that none of these depend on $d$ when $d \geq K$, the impact of this parameter ultimately is not in the loss of any given structure.

The only remaining quantities that are not determined are $U_{L-1}, ..., U_0 \in \mathbb{R}^{d \times d}$. The only constraint is that they must be orthogonal matrices. On the surface it appears that each choice of $Z^*$ has the same degeneracy, since each just requires orthogonal matrices for the $U_l$. However, there are two forms of degeneracy. The first leads to a different point in the parameter space, whilst the second leads to a different expression of the same point in the parameter space. We only aim to count the first. To demonstrate this let us look at a single $l \in \{1, ..., L - 1\}$. We have:

$$W_l = U_l \Sigma U_{l-1}^T = \sum_{i=1}^{r} s_i u_i^{(l)} u_i^{(l-1)T},$$

where $u_i^{(l)}$ are the columns of $U_l$. We see that only the singular vectors corresponding to non-zero singular values contribute to the expression of $W_l$. So instead of specifying a full orthogonal matrix,

we must specify $r$ orthonormal vectors, since changing the zero singular-vectors does not change the parameter matrix.

Whilst we must specify $r$ orthonormal vectors, we must account for potential degenerate singular values, since this means there are transformations of the degenerate singular vectors that leave the matrix unchanged. Indeed this is precisely the case for the DNC solution. We consider the two extreme ends:

**All singular values are different:** When all non-zero singular values are different, we are picking an orthogonal $r$-frame in $\mathbb{R}^d$. The space of such solutions is known as the Stiefel manifold $\mathrm{St}(r, d)$, and it is a standard result that this space has dimension given by:

$$\dim(\mathrm{St}(r, d)) = rd - \frac{1}{2}r(r+1).$$

This is the largest the matrix $U_l$ can have its degeneracy as a function of $r$.

**All singular values are equal:** When all singular values are equal, we are picking a $r$-dimensional vector subspace. The space of such solutions is known as the Grassmannian manifold $\mathrm{Gr}(r, d)$. It is a standard result that this space has dimension given by:

$$\dim(\mathrm{Gr}(r, d)) = rd - r^2.$$

This is the smallest the matrix $U_l$ can have its degeneracy as a function of $r$.

This gives the degeneracy for a single $U_l$, we have the same degeneracy for each of $U_{L-1}, ..., U_0$. Denoting the dimension of the space of solutions corresponding to the structure $Z^*$ as $D_{Z^*}(d)$, we have

$$L(rd - r^2) \leq D_{Z^*}(d) \leq L\left(rd - \frac{1}{2}r(r+1)\right).$$

Where exactly it falls in this range depends on the number of degeneracies of the singular values, but in all cases we get a value of the form $rd - C(r)$, for some function $C(r)$ obeying $\frac{1}{2}r(r+1) \leq C(r) \leq r^2$.

Also note that the reparametrizations of $U_L, V_0$ are now accounted for since they correspond to reparametrizations of the other matrices as are accounted for in the above.

In the case of DNC, the dimension is exactly

$$D_{\mathrm{DNC}}(d) = L((K-1)d - (K-1)^2),$$

and so we can consider the ratio of this dimension to the dimension of our other critical point $Z^*$:

$$\frac{(K-1)d - (K-1)^2}{rd - \frac{1}{2}r(r+1)} \leq \frac{D_{\mathrm{DNC}}(d)}{D_{Z^*}(d)} \leq \frac{(K-1)d - (K-1)^2}{rd - r^2},$$

from which it immediately follows that as $d \to \infty$ we have

$$\frac{D_{\mathrm{DNC}}(d)}{D_{Z^*}(d)} \to \frac{(K-1)}{r}.$$

It remains to show that the ratio is monotonic increasing and starts below 1 initially. First note that our upper bound on the ratio is less than 1 when:

$$\frac{(K-1)d - (K-1)^2}{rd - r^2} < 1 \iff d < K + r - 1,$$

and since $K + r - 1 > K$, this gives that the ratio starts below 1. For monotonicity, it is sufficient to note that the function

$$f(d) = \frac{(K-1)d - (K-1)^2}{rd - C(r)}$$

is monotonic on $d \geq K$ if $(K-1)r > C(r)$, which is the case since we know $C(r) \leq r^2 < r(K-1)$.

## B.8 Proof of Theorem 6

We begin by showing that the low-rank solutions described previously are indeed critical points of the model when the level of regularization $\lambda$ is sufficiently small. The logit matrix of our low-rank solution is

$$Z = \beta \tilde{X} \otimes 1_n^T, \quad \text{where} \quad \tilde{X} = \begin{bmatrix} X & 0 & ... & 0 \\ 0 & X & ... & 0 \\ ... & ... & ... & ... \\ 0 & 0 & ... & X \end{bmatrix}, \quad \text{with } X = \begin{bmatrix} 1 & -1 \\ -1 & 1 \end{bmatrix},$$

and we assume the parameter matrices satisfy the structural properties of Lemma 1. Also, note the matrix $Z$ has $K/2$ non-zero singular values, each equal to $2\beta\sqrt{n}$.

Recall from Equation (12) in Appendix B.6 that, at any critical point, we require for $l = 0, ..., L$:

$$W_l = \frac{1}{Kn\lambda} A_{l+1}^T (Y - P) H_l^T. \tag{26}$$

We focus on the case $1 \leq l \leq L - 1$, noting the boundary cases can be handled similarly.

From Lemma 1, and the definitions of $A_{l+1}$ and $H_l$, we have:

$$W_l = U_l \Sigma U_{l-1}^T, \quad A_{l+1} = U_L \tilde{\Sigma} \Sigma^{L-l-1} U_l^T, \quad H_l = U_{l-1} \Sigma^{l-1} \bar{\Sigma} V_0^T.$$

Substituting into the criticality condition of Equation (26) yields:

$$Kn\lambda U_l \Sigma U_{l-1}^T = U_l \Sigma^{L-l-1} \tilde{\Sigma}^T U_L^T (Y - P) V_0 \bar{\Sigma}^T \Sigma^{l-1} U_{l-1}^T.$$

Pre and post multiplying by $U_l^T, U_{l-1}$, respectively, reduces this to:

$$Kn\lambda\Sigma = \Sigma^{L-l-1} \tilde{\Sigma}^T U_L^T (Y - P) V_0 \bar{\Sigma}^T \Sigma^{l-1}.$$

We now, for the purpose of this section only, define $D \in \mathbb{R}^{d \times d}, \tilde{D} \in \mathbb{R}^{K \times d}, \bar{D} \in \mathbb{R}^{d \times Kn}, D' \in \mathbb{R}^{K \times K}$ to be matrices where the top $K/2 \times K/2$ block is the identity matrix, with all other entries being zero. We can now pull out the scales of the matrices in our previous equation, giving:

$$Kn\lambda D = (2\beta\sqrt{n})^{\frac{L-1}{L+1}} \tilde{D}^T U_L^T (Y - P) V_0 \bar{D}^T.$$

We now perform a conjugate transform by the orthogonal matrix:

$$O = \begin{bmatrix} U_L & 0 \\ 0 & I_{d-K} \end{bmatrix}.$$

This reduces both sides of our criticality condition to only having their top $K \times K$ block being non-zero. The reduced expression is then:

$$Kn\lambda U_L D' U_L^T = (2\beta\sqrt{n})^{\frac{L-1}{L+1}} U_L D' U_L^T (Y - P) V_0 [D', 0_{K \times K(n-1)}]^T U_L^T.$$

We then note that

$$U_L D' U_L^T = \frac{1}{2}\tilde{X}, \quad U_L[D', 0_{K \times K(n-1)}]V_0^T = \frac{1}{2\sqrt{n}}\tilde{X} \otimes 1_n^T,$$

and so this reduces further to:

$$Kn\lambda\tilde{X} = \frac{(2\beta\sqrt{n})^{\frac{L-1}{L+1}}}{2\sqrt{n}}\tilde{X}(Y-P)(\tilde{X}\otimes 1_n).$$

Calculating the probability matrix for this specific logit matrix, we get, written in block form,

$$P = \frac{1}{K-2+e^\beta+e^{-\beta}}\begin{bmatrix} P' & 1 & ... & 1 \\ 1 & P' & ... & 1 \\ ... & ... & ... & ... \\ 1 & 1 & ... & P' \end{bmatrix} \otimes 1_n^T, \quad \text{where} \quad P' = \begin{bmatrix} e^\beta & e^{-\beta} \\ e^{-\beta} & e^\beta \end{bmatrix},$$

Then:

$$Y-P = \frac{1}{K-2+e^\beta+e^{-\beta}}\left(\begin{bmatrix} E & 1 & ... & 1 \\ 1 & E & ... & 1 \\ ... & ... & ... & ... \\ 1 & 1 & ... & E \end{bmatrix} \otimes 1_n^T + e^{-\beta}\tilde{X}\otimes 1_n^T\right),$$

where

$$E = \begin{bmatrix} K-2 & 0 \\ 0 & K-2 \end{bmatrix}.$$

Explicit computation then gives:

$$\tilde{X}(Y-P)(\tilde{X}\otimes 1_n^T) = \frac{n\left(2(K-2)+4e^{-\beta}\right)}{K-2+e^\beta+e^{-\beta}}\tilde{X},$$

and so our criticality condition reduces to

$$\lambda n^{\frac{1}{L+1}}\tilde{X} = \frac{1}{K}(2\beta)^{\frac{L-1}{L+1}}\frac{1}{K-2+e^\beta+e^{-\beta}}\left(K-2+2e^{-\beta}\right)\tilde{X},$$

which holds when:

$$\lambda n^{\frac{1}{L+1}} = \frac{1}{K}(2\beta)^{\frac{L-1}{L+1}}\frac{1}{K-2+e^\beta+e^{-\beta}}\left(K-2+2e^{-\beta}\right). \tag{27}$$

Plots of the RHS function in $\beta$ for characteristic $L, K$ reveal there is a single solution $\beta$ such that $\beta \to \infty$ as $\lambda \to 0$. To see the existence of such a solution theoretically, note that if $\beta$ is large this equation reduces to a similar one to what we saw for the DNC case in the proof of Theorem 4, as described in Equation (19). A similar argument then shows that there is a solution for which $\beta$ diverges as $\lambda \to 0$. Thus, this structure is a critical point of the model for small enough $\lambda$, with a scale that diverges as $\lambda \to 0$.

We now turn to the comparison of the Hessian of this low-rank solution versus that of a DNC solution, and analyze the asymptotic scaling behavior in the limit $\lambda \to 0$.

Recall from the proof of Theorem 4 (Equation (20)) that the on-diagonal Hessian blocks (pre-flattening) are given by:

$$\frac{\partial^2 \mathcal{L}}{\partial(W_l)_{ab}\partial(W_l)_{cd}} = \lambda\delta_{ac}\delta_{bd} + \frac{1}{K}\sum_y (A_{l+1}^T\rho_y A_{l+1}^T)_{ac}(h_y^{(l)}h_y^{(l)T})_{bd},$$

while the off-diagonal blocks for $r < l$ are given by (Equation (21)):

$$\frac{\partial^2 \mathcal{L}}{\partial(W_l)_{ab}\partial(W_r)_{cd}} = \frac{1}{K}(A_{l+1}^T(P-Y)H_r^T)_{ad}[(W_{l-1}...W_{r+1})^T]_{cb} + \frac{1}{K}\sum_y (A_{l+1}^T \rho_y A_{r+1}^T)_{ac}(h_y^{(l)}h_y^{(r)T})_{bd},$$

where $\rho_y = \mathrm{diag}(p_y) - p_y p_y^T$, with $p_y$ the $y^{\text{th}}$ column of the softmax probability matrix $P$.

We argue the first term in each block is $O(\lambda)$, whilst the second is $O(\beta\lambda)$, and so in the $\lambda \to 0$ limit the first term becomes small compared to the second and does not contribute to any flatness metric. Clearly for the on-diagonal Hessian blocks, the first term is $O(\lambda)$. For the off-diagonal, note, all dependence on $\lambda$ to leading order is given by the scaling constant, with that scaling being

$$\beta^{\frac{L-l}{L+1}}\beta^{\frac{l-r}{L+1}}\beta^{l-r-1}\frac{1}{K-2+e^\beta+e^{-\beta}} = \frac{1}{K-2+e^\beta+e^{-\beta}}\beta^{\frac{L-1}{L+1}} \sim \lambda,$$

where we used Equation (27) to relate $\beta$ to $\lambda$. Hence, this is also $O(\lambda)$ as claimed.

To evaluate the scale of the second terms of the block Hessians, we need to know how $\rho_y$ depends on $\lambda$. Doing the computation explicitly for $\rho_1$ gives:

$$\rho_1 = \frac{e^\beta}{(K-2+e^\beta+e^{-\beta})^2}\begin{bmatrix} K-2 & 0 & -1 & ... & -1 \\ 0 & 0 & 0 & ... & 0 \\ -1 & 0 & 1 & ... & 0 \\ ... & ... & ... & ... & ... \\ -1 & 0 & 0 & ... & 1 \end{bmatrix} + O(e^{-2\beta}).$$

This scale is representative for all $y$. The scale of the second block Hessian term for any $l, r$ is then:

$$\frac{e^\beta}{(K-2+e^\beta+e^{-\beta})^2}\beta^{\frac{L-l}{L+1}}\beta^{\frac{L-r}{L+1}}\beta^{\frac{l}{L+1}}\beta^{\frac{r}{L+1}} \sim \frac{1}{K-2+e^\beta+e^{-\beta}}\beta^{\frac{2L}{L+1}} \sim \beta\lambda.$$

Since $\beta$ diverges, this term dominates, and we can safely ignore the first terms of the block Hessians in the $\lambda \to 0$ limit.

We have found that each block of the Hessian is, to leading order, a fixed object (i.e. no $\lambda$ dependence), multiplied by $\beta\lambda$. Since each of our norms allow a scaling constant to be pulled out, we arrive at the scale of the Hessian being asymptotically proportional to $\beta\lambda$ in the limit $\lambda \to 0$.

The exact same steps for DNC lead to the same implications, with each leading term in the block Hessian being proportional to $\alpha\lambda$, and so the scale of the DNC Hessian is also asymptotically proportional to $\alpha\lambda$ to leading order.

We now need only understand how the two scales $\alpha$ and $\beta$ depend on the parameter $\lambda$. This is described in Equations (19), (27), which when written as asymptotic relations give:

$$\lambda n^{\frac{1}{L+1}} \sim \frac{K-2}{K}(2\beta)^{\frac{L-1}{L+1}}e^{-\beta}, \quad \lambda n^{\frac{1}{L+1}} \sim \alpha^{\frac{L-1}{L+1}}e^{-\alpha}.$$

Starting with the $\beta$ equation, taking logarithms gives:

$$\frac{-\log(\lambda)}{\beta} \sim 1 - \frac{L-1}{L+1}\frac{\log(\beta)}{\beta} - \frac{1}{\beta}\left(\frac{L-1}{L+1}\log(2) - \log\left(\frac{K}{K-2}\right) - \frac{1}{L+1}\log(n)\right),$$

and since $\beta \to \infty$, we arrive at $\beta \sim -\log(\lambda)$.

Similarly for the $\alpha$ equation:

$$\frac{-\log(\lambda)}{\alpha} \sim 1 - \frac{L-1}{L+1}\frac{\log(\alpha)}{\alpha} + \frac{1}{\alpha}\left[\frac{1}{L+1}\log(n)\right],$$

and since $\alpha$ also diverges, we find $\alpha \sim -\log(\lambda)$ also. Hence the ratio of the scales is:

$$\frac{\alpha\lambda}{\beta\lambda} \sim 1.$$

Since the two deterministic matrices are different, the constant of proportionality is not 1, but we conclude that the ratio of any norm of the Hessian must tend to a constant as $\lambda \to 0$. This constant is clearly dependent on the choice of norm. We also note that since $\lambda \log(\lambda) \to 0$ as $\lambda \to 0$, both these scales do tend to zero in the limit. This completes the proof.

## B.9 Proof of Theorem 7

We aim to show that no solution with DNC structure in the deep ReLU UFM can be globally optimal for the values of $K$ and $L$ stated in Theorem 7, under the given technical assumption. We will first demonstrate that the loss attained by a DNC solution in the deep linear UFM serves as a lower bound for the loss of a DNC solution in the ReLU case.

Recall under the DNC structure, defined in Section 3.4, the matrices $H_l$ and $\tilde{H}_l$, (as defined in Section 2) take the form:

$$H_l = M_l \otimes 1_n^T, \quad \tilde{H}_l = \tilde{M}_l \otimes 1_n^T,$$

and the matrices $M_l - \mu_G^{(l)}, \tilde{M}_l - \tilde{\mu}_G^{(l)}$ align with the simplex ETF. We will denote these globally centered class mean matrices as $M_l^{(G)} = M_l - \mu_G^{(l)}$ and $\tilde{M}_l^{(G)} = \tilde{M}_l - \tilde{\mu}_G^{(l)}$ respectively.

We write $S = I_K - \frac{1}{K}1_K 1_K^T = QD'Q^T$, Where $Q$ is an orthogonal matrix that diagonalizes $S$, and $D' = \mathrm{diag}(1,1,...,1,0) \in \mathbb{R}^{K \times K}$.

The alignment of $M_l^{(G)}, \tilde{M}_l^{(G)}$ with the simplex ETF implies that we can write their SVDs as

$$M_l^{(G)} = \alpha_l R_l \tilde{D}^T Q^T, \quad \tilde{M}_l^{(G)} = \beta_l \tilde{R}_l \tilde{D}^T Q^T,$$

where $R_l, \tilde{R}_l \in \mathbb{R}^{d \times d}$ are orthogonal matrices, $\tilde{D} = [D, 0_{K \times (d-K)}] \in \mathbb{R}^{K \times d}$ and $\alpha_l, \beta_l > 0$ are scales that give the non-zero singular values of each matrix respectively.

Recall, also by definition of a DNC solution, we can write the logit matrix $Z = W_L \sigma(...W_2\sigma(W_1 H_1)...)$, as $Z = \alpha_{L+1} S \otimes 1_n^T$. Inputting this into our loss function, we find the loss of a DNC solution is given by

$$\mathcal{L}_{\mathrm{DNC}} = \log(1 + (K-1)e^{-\alpha_{L+1}}) + \frac{1}{2}\lambda \|H_1\|_F^2 + \frac{1}{2}\lambda \sum_{l=1}^{L} \|W_l\|_F^2. \tag{28}$$

We now seek to express $\alpha_{L+1}$ in terms of the singular values of the parameter matrices.

First note that for each $l$ we have the equation $H_l = W_{l-1}\tilde{H}_{l-1}$. By using repeated columns and global centering both sides, we deduce:

$$M_l^{(G)} = W_{l-1}\tilde{M}_{l-1}^{(G)}. \tag{29}$$

Let the first $K$ singular values of $W_{l-1}$ be denoted by $\omega_i^{(l-1)}$ for $i = 1, ..., K$, where as usual these are in decreasing order. Note all other singular values of $W_{l-1}$ must be zero by the third DNC property.

We quote the following lemma from Horn & Johnson [16] about the singular values of a matrix product in terms of the singular values of the components.

**Lemma:** *Given two matrices $A \in \mathbb{R}^{m \times k}$ and $B \in \mathbb{R}^{k \times n}$, and denoting the $i^{th}$ singular value of a matrix by $s_i(\cdot)$ in descending order, we have*

$$s_i(AB) \le s_i(A)s_1(B).$$

Recall that the singular values of $M_l^{(G)}$ are $\alpha_l$ with multiplicity $K-1$, and 0 with multiplicity 1, and the singular values of $\tilde{M}_{l-1}^{(G)}$ are $\beta_{l-1}$ with multiplicity $K-1$, and 0 with multiplicity 1. Therefore, applying the Lemma to Equation (29), this gives us the following inequality for $2 \le l \le L+1$ and $1 \le i \le K-1$:

$$\alpha_l \le \omega_i^{(l-1)}\beta_{l-1}.$$

In particular since $\omega_i^{(l-1)} \ge \omega_{K-1}^{(l-1)}$ for $1 \le i \le K-1$, the strongest of these inequalities is

$$\alpha_l \le \omega_{K-1}^{(l-1)}\beta_{l-1}.$$

This gives us an inequality relating $\alpha_l$ to $\beta_{l-1}$, we will now get an inequality that relates $\beta_{l-1}$ to $\alpha_{l-1}$ using the technical assumption.

From the definition of $\tilde{H}_l$ we have

$$\tilde{M}_l^{(G)} + \tilde{\mu}_G^{(l)}1_K^T = \sigma(M_l^{(G)} + \mu_G^{(l)}1_K^T).$$

Taking Frobenius norms on both sides and using the fact that the ReLU activation satisfies $\|\sigma(M)\|_F \le \|M\|_F$ for any matrix $M$, we get:

$$\|\tilde{M}_l^{(G)} + \tilde{\mu}_G^{(l)}1_K^T\|_F^2 \le \|M_l^{(G)} + \mu_G^{(l)}1_K^T\|_F^2.$$

Now using the fact that $\tilde{M}_l^{(G)}1_K = M_l^{(G)}1_K = 0$, the above becomes

$$\|\tilde{M}_l^{(G)}\|_F^2 + K\|\tilde{\mu}_G^{(l)}\|_2^2 \le \|M_l^{(G)}\|_F^2 + K\|\mu_G^{(l)}\|_2^2.$$

Now define the ratios

$$r_l = \frac{\|\mu_G^{(l)}\|_2}{\|M_l^{(G)}\|_F}, \quad \tilde{r}_l = \frac{\|\tilde{\mu}_G^{(l)}\|_2}{\|\tilde{M}_l^{(G)}\|_F}.$$

Using these ratios, and the expressions for the Frobenius norms in terms of the singular values, we can write:

$$\beta_l^2 \le \alpha_l^2 \left[\frac{1 + Kr_l^2}{1 + K\tilde{r}_l^2}\right],$$

and using the technical assumption this gives

$$\beta_l \le \alpha_l, \quad \text{for } 2 \le l \le L.$$

If we define $\tilde{H}_1 = H_1$, then this inequality extends to $l = 1$.

Using our two inequalities involving $\alpha_l, \beta_l$ we can recursively derive a bound on the output scale:

$$\alpha_l \le \omega_{K-1}^{(l-1)}\beta_{l-1} \le \omega_{K-1}^{(l-1)}\alpha_{l-1}.$$

Iterating this repeatedly yields:

$$\alpha_{L+1} \leq \alpha_1 \omega_{K-1}^{(L)} ... \omega_{K-1}^{(1)}.$$

We can now return to the DNC loss given by Equation (28). Using the fact that the function $\tilde{g}(x) = \log(1 + (K-1)e^{-x})$ is a strictly decreasing function, we have that:

$$\mathcal{L}_{\text{DNC}} \geq \tilde{g}\left(\alpha_1 \prod_{l=1}^{L} \omega_{K-1}^{(l)}\right) + \frac{1}{2}\lambda\|H_1\|_F^2 + \frac{1}{2}\lambda\sum_{l=1}^{L}\|W_l\|_F^2.$$

Next, we construct a lower bound on the Frobenius norm terms.

From the fact that $M_1^{(G)}1_K = 0$, we have:

$$\|M_1\|_F^2 = \|M_1^{(G)}\|_F^2 + K\|\mu_G^{(1)}\|_2^2 \geq \|M_1^{(G)}\|_F^2 = (K-1)\alpha_1^2,$$

and hence:

$$\|H_1\|_F^2 = n\|M_1\|_F^2 \geq (K-1)n\alpha_1^2.$$

For the weight matrices, recall that $\omega_i^{(l)} \geq \omega_{K-1}^{(l)}$ for $i = 1, ..., K-1$, giving that

$$\|W_l\|_F^2 = \sum_{l=1}^{K}\left(\omega_i^{(l)}\right)^2 \geq (K-1)\left(\omega_{K-1}^{(l)}\right)^2.$$

Combining everything, we conclude that the loss of a DNC solution satisfies:

$$\mathcal{L}_{\text{DNC}} \geq \tilde{g}\left(\alpha_1 \prod_{l=1}^{L} \omega_{K-1}^{(l)}\right) + \frac{1}{2}\lambda(K-1)n\alpha_1^2 + \frac{1}{2}\lambda(K-1)\sum_{l=1}^{L}\left(\omega_{K-1}^{(l)}\right)^2.$$

We will now show that this lower bound is attainable by a linear model DNC solution, and thus, the loss attained at a DNC solution in the non-linear model cannot outperform the global minimum of a DNC solution in the linear case.

First consider the function on the right hand side of the previous inequality and for notational convenience, relabel

$$\alpha_1\sqrt{n} \to x_0, \quad \omega_{K-1}^{(l)} \to x_l.$$

Then the DNC loss lower bound becomes:

$$\text{RHS} = \tilde{g}\left(\frac{1}{\sqrt{n}}\prod_{l=0}^{L} x_l\right) + \frac{1}{2}\lambda(K-1)\sum_{l=0}^{L} x_l^2.$$

Note any global-minimum of this structure occurs for finite $x_l$, since the regularization term diverges otherwise. Hence the sum $\sum_{l=0}^{L} x_l$ is finite at all global minima. Now note, since $\tilde{g}$ is a decreasing function, it is minimized when all $x_l$ are equal given that their sum is finite, this follows from the AM-GM inequality. In addition since the quadratic function is convex, Jensen's inequality says that the regularization term is minimized when all the $x_l$ are equal. Hence the total loss is minimized when all $x_l$ take the same value. Setting $x_l = (\alpha\sqrt{n})^{\frac{1}{L+1}}$, we find the minimum DNC loss is lower bounded by the following

$$\mathcal{L}_{\text{DNC}} \geq \min_{\alpha \geq 0}\left\{\tilde{g}(\alpha) + \frac{1}{2}\lambda(K-1)(L+1)(\alpha\sqrt{n})^{\frac{2}{L+1}}\right\}.$$

Note if the minimum occurs at $\alpha = 0$, then this is attainable by setting all the parameter matrices to zero, which is not a DNC solution, and so DNC is not optimal in this case. Hence we assume minimum occurs at $\alpha \neq 0$.

This is precisely the loss attained by the deep linear UFM at the best performing DNC solution, as derived in the proof of Theorem 1 in Appendix B.3. Thus, we can compare to the following loss parameterized by a scale $\alpha \geq 0$

$$\mathcal{L}_{\text{DNC}}^{\text{linear}}(\alpha) = \log(1 + (K-1)e^{-\alpha}) + \frac{1}{2}\lambda(L+1)(K-1)(\alpha\sqrt{n})^{\frac{2}{L+1}}.$$

We will now construct a low-rank solution that can beat this loss, for any $\alpha$, even for ReLU non-linearity included.

Initially let us assume $K$ is even. In constructing our low-rank solution, we can no longer use the same structure as we used in the linear model. This is since for any choice of the orthogonal matrices $U_{L-1}, ..., U_0$ there will be negative entries that are zeroed out by ReLU. To address this, we modify the construction by adding a rank-1 positive matrix to ensure all entries remain non-negative throughout the network.

Define the matrices $\bar{X}, \bar{Y} \in \mathbb{R}^{K \times K}$ as follows:

$$\bar{X} = \begin{bmatrix} X & 0 & \dots & 0 \\ 0 & X & \dots & 0 \\ \dots & \dots & \dots & \dots \\ 0 & 0 & \dots & X \end{bmatrix} \text{ where } X = \begin{bmatrix} 1 & -1 \\ -1 & 1 \end{bmatrix},$$

$$\bar{Y} = 1_K 1_K^T.$$

We observe the following key properties

$$\bar{X}\bar{Y} = \bar{Y}\bar{X} = 0,$$
$$\bar{X}^2 = 2\bar{X}, \quad \bar{Y}^2 = K\bar{Y},$$
$$\|\bar{X}\|_F^2 = 2K, \quad \|\bar{Y}\|_F^2 = K^2.$$

these algebraic identities will be useful in analyzing the propagation of signals throughout the layers.

We construct our solution so that for $2 \leq l \leq L$ the intermediate features are of the form:

$$H_l = \begin{bmatrix} \phi_l(\bar{X} + \bar{Y}) \\ 0_{(d-K) \times K} \end{bmatrix} \otimes 1_n^T \in \mathbb{R}^{d \times Kn}, \tag{30}$$

for some scalar $\phi_l \geq 0$.

We'll also construct our weight matrices so that for $2 \leq l \leq L-1$ they are of the form:

$$W_l = \begin{bmatrix} \psi_l \bar{X} + \chi_l \bar{Y} & 0_{K \times (d-K)} \\ 0_{(d-K) \times K} & 0_{(d-K) \times (d-K)} \end{bmatrix} \in \mathbb{R}^{d \times d},$$

for some scalars $\psi_l, \chi_l \geq 0$.

Since all entries of $H_l$ are non-negative by construction, the ReLU activation has no effect, meaning $H_{l+1} = W_l \sigma(H_l) = W_l H_l$. Using this, and the properties of $\bar{X}$ and $\bar{Y}$ stated above, we have for $2 \leq l \leq L-1$

$$H_{l+1} = \begin{bmatrix} 2\psi_l\phi_l\bar{X} + K\chi_l\phi_l\bar{Y} \\ 0_{(d-K) \times K} \end{bmatrix} \otimes 1_n^T.$$

To preserve the structural form, we require that both components scale equally. This is achieved by setting $\chi_l = \frac{2}{K}\psi_l$. Hence we arrive at the following recurrence relation for $2 \leq l \leq L-1$

$$\phi_{l+1} = 2\psi_l\phi_l.$$

In addition, for the last layer we set $W_L = [\psi_L \bar{X}, 0_{K \times (d-K)}] \in \mathbb{R}^{K \times d}$.

We want to set the scales of each of our parameter matrices to be equal, and so we need to also calculate the appropriate choices of $W_1, H_1$. We have our given form for $H_2$ stated in Equation (30), suppose its SVD is $H_2 = U_2 \Sigma V_2^T$. Then we set

$$W_1 = U_2 \Sigma^{\frac{1}{2}} \tilde{U}^T, \quad H_1 = \tilde{U} \Sigma^{\frac{1}{2}} V_2^T,$$

where $\tilde{U}$ is any orthogonal matrix of appropriate dimensions. We then find

$$\|H_1\|_F^2 = \|W_1\|_F^2 = \sum_i s_i(H_2),$$

where $s_i(H_2)$ are the singular values of $H_2$. using the forms of $\bar{X}, \bar{Y}$, the non-zero singular values of $H_2$ are $K\sqrt{n}\phi_2$ with multiplicity 1, and $2\sqrt{n}\phi_2$ with multiplicity $\frac{K}{2}$. Hence we find that

$$\|H_1\|_F^2 = \|W_1\|_F^2 = 2K\sqrt{n}\phi_2.$$

For $W_l$ with $2 \le l \le L - 1$ we can also simply calculate the norm

$$\|W_l\|_F^2 = \psi_l^2 \|\bar{X}\|_F^2 + \frac{4}{K^2}\psi_l^2 \|\bar{Y}\|_F^2 = 2(K+2)\psi_l^2,$$

and also clearly for the final layer we have

$$\|W_L\|_F^2 = 2K\psi_L^2.$$

We now enforce equal Frobenius norms for each parameter matrix. We see for $2 \le l \le L - 1$ that if the scales are equal then $\psi_l = \psi$. In addition setting the scale of $W_1$ and $W_L$ equal to the scale of $W_l$ for $2 \le l \le L - 1$ gives

$$\phi_2 = \frac{K+2}{K\sqrt{n}}\psi^2, \quad \psi_L = \sqrt{\frac{K+2}{K}}\psi.$$

This determines all the scales of our parameter matrices up to a single scale $\psi$. We now want to compute how this scale arises in the fit term through the output of the network.

Using the previously established recurrence relation: for $2 \le l \le L - 1$ we have $\phi_{l+1} = 2\psi_l\phi_l$, this gives that for $2 \le l \le L - 1$

$$\phi_{l+1} = (2\psi)^{l-1}\phi_2,$$

and hence

$$\phi_L = (2\psi)^{L-2}\phi_2 = (2\psi)^{L-2}\psi^2 \frac{K+2}{K\sqrt{n}}.$$

Since $Z = W_L \sigma(H_L) = \psi_L \phi_L (\bar{X}^2 \otimes 1_n^T)$, we then have

$$Z = \left(\frac{1}{\sqrt{n}}\right) 2^{L-1}\psi^{L+1}\left(\frac{K+2}{K}\right)^{\frac{3}{2}} \bar{X} \otimes 1_n^T.$$

We now compare this low rank solution to the DNC solution, beginning with the fit term characterized by the function $g : \mathbb{R}^{K \times Kn} \to \mathbb{R}$ that was definied at the start of Appendix B. In the proof of Theorem 1 in Appendix B.3, we saw that for any scale $\gamma > 0$:

$$g(\gamma \bar{X} \otimes 1_n^T) < g\left(\gamma \left(I_K - \frac{1}{K} 1_K 1_K^T\right) \otimes 1_n^T\right).$$

Hence if we match the scales of the output matrices, the low rank solution will strictly outperform the DNC solution on the fit term. To match the scales, set $\psi$ to have the following relationship with the scale $\alpha$ of the DNC solution:

$$\alpha = \left(\frac{1}{\sqrt{n}}\right) 2^{L-1} \psi^{L+1} \left(\frac{K+2}{K}\right)^{\frac{3}{2}}.$$

This guarantees that our low-rank solution performs better on the fit term. We now assess when it will do better on the regularization term. Computing the regularization term for the low rank solution, we require that:

$$\frac{1}{2}\lambda(L+1)(K-1)(\alpha\sqrt{n})^{\frac{2}{L+1}} \geq \frac{1}{2}\lambda(L+1)[2(K+2)\psi^2].$$

Removing the common factors, since they are positive, and using the form of $\alpha$ in terms of $\psi$, this is equivalent to

$$(K-1)\left(\frac{K+2}{K}\right)^{\frac{3}{L+1}} 2^{\frac{2(L-1)}{(L+1)}} \psi^2 \geq 2(K+2)\psi^2.$$

Eliminating $\psi^2$, since we assume it is non-zero, and cleaning the expression slightly gives:

$$\left(\frac{K+2}{K}\right)^{\frac{3}{L+1}} 2^{\frac{L-3}{L+1}}(K-1) \geq K+2,$$

and this holds when $L = 4, K \geq 14$, or $L \geq 5, K \geq 10$. Hence we find that in these cases our low-rank solution outperforms the DNC solution when the scales of the output matrices are set equal. Hence this is true at the optimal scale of the DNC solution, and DNC cannot be optimal. In this proof, we assumed when minimizing the scale that $\alpha \neq 0$ at the minima, but if the optimal $\alpha$ occurs at $\alpha = 0$ then DNC is not optimal, since we required $\alpha > 0$ in its definition.

Hence we find that the DNC solution cannot be optimal in the deep ReLU UFM when $K$ is even, subject to these conditions on $L$ and $K$.

It remains to cover the odd $K$ cases. Let $K = 2m + 1$. We similarly define our matrices $\bar{X}, \bar{Y}$ to before, with a slight change to account for the odd dimension.

$$\bar{X} = \begin{bmatrix} X & 0_{2\times 2} & \ldots & 0_{2\times 2} & 0 \\ 0_{2\times 2} & X & \ldots & 0_{2\times 2} & 0 \\ \ldots & \ldots & \ldots & \ldots & \ldots \\ 0_{2\times 2} & 0_{2\times 2} & \ldots & X & 0 \\ 0 & 0 & \ldots & 0 & 2 \end{bmatrix} \text{ where } X = \begin{bmatrix} 1 & -1 \\ -1 & 1 \end{bmatrix},$$

$$\bar{Y} = \begin{bmatrix} 1_{2m} 1_{2m}^T & 0_{2m\times 1} \\ 0_{1\times 2m} & 0 \end{bmatrix}.$$

These matrices now satisfy

$$\bar{X}\bar{Y} = \bar{Y}\bar{X} = 0,$$
$$\bar{X}^2 = 2\bar{X}, \quad \bar{Y}^2 = (K-1)\bar{Y},$$
$$\|\bar{X}\|_F^2 = 2(K+1), \quad \|\bar{Y}\|_F^2 = (K-1)^2.$$

As before, define the intermediate features and weights to have the following form:

$$\text{for } 2 \leq l \leq L: \quad H_l = \begin{bmatrix} \phi_l(\bar{X} + \bar{Y}) \\ 0_{(d-K) \times K} \end{bmatrix} \otimes 1_n^T \in \mathbb{R}^{d \times Kn},$$

$$\text{for } 2 \leq l \leq L-1: \quad W_l = \begin{bmatrix} \psi_l \bar{X} + \chi_l \bar{Y} & 0_{K \times (d-K)} \\ 0_{(d-K) \times K} & 0_{(d-K) \times (d-K)} \end{bmatrix} \in \mathbb{R}^{d \times d}.$$

To maintain the desired form of the $H_l$ matrices we now require $\chi_l = \frac{2}{K-1}\psi_l$, and we find $\phi_{l+1} = 2\psi_l\phi_l$ as before. For the final layer, define:

$$W_L = [\psi_L \bar{X}, 0_{K \times (d-K)}].$$

As before, define the $W_1, H_1$ matrices in terms of the components of the SVD of $H_2$. This yields

$$\|H_1\|_F^2 = \|W_1\|_F^2 = 2K\sqrt{n}\phi_2.$$

Computing the norms of the other parameter matrices gives

$$2 \leq l \leq L-1: \quad \|W_l\|_F^2 = 2(K+3)\psi_l^2,$$

$$\|W_L\|_F^2 = 2(K+1)\psi_L^2.$$

Again setting each parameter matrix norm equal we find that for

$$2 \leq l \leq L-1: \quad \psi_l = \psi, \quad \psi_L = \sqrt{\frac{K+3}{K+1}}\psi, \quad \phi_2 = \frac{K+3}{K\sqrt{n}}\psi^2.$$

We then use our recurrence relation for $\phi_l$ to get $\phi_L = (2\psi)^{L-2}\phi_2$. We hence find that the output matrix $Z$ is

$$Z = \left(\frac{1}{\sqrt{n}}\right) 2^{L-1} \sqrt{\left(\frac{K+3}{K+1}\right)} \left(\frac{K+3}{K}\right) \psi^{L+1} \bar{X}.$$

Again setting the scale of the DNC solution equal to the scale of this $Z$ guarantees it outperforms on the fit term, whilst the regularization term reduces to

$$K + 3 \leq (K-1) 2^{\frac{L-3}{L+1}} \left(\frac{K+3}{K+1}\right)^{\frac{1}{L+1}} \left(\frac{K+3}{K}\right)^{\frac{2}{L+1}},$$

and this holds for $L = 4, K \geq 17$, or $L \geq 5, K \geq 11$. Hence by the same arguments the DNC solution cannot be optimal for odd $K$ in these cases.

Putting these two together, we find that the DNC solution is not optimal when $K, L$ satisfy $L = 4, K \geq 16$, or $L \geq 5, K \geq 10$.

## C  Proofs in imbalanced deep linear UFM

The proofs in this section closely follow those of the corresponding theorems in the main text. Here, we highlight the key differences that arise due to class imbalance, while referring the reader to the previous appendix for all other details that remain unchanged.

## C.1 Proof of Theorem A1

We follow the proof of Theorem 2, and begin by showing that there exists a constant $\gamma > 0$ such that $\alpha\gamma\tilde{M}$ achieves a strictly lower fit loss than $\alpha X$ for any $\alpha > 0$. Let the column of $X$ corresponding to the $i^{\text{th}}$ sample of class $c$ be denoted by $x_{ic}$, and let the corresponding column for $\tilde{M}$ be $m_{ic}$. Then the respective fit losses are :

$$g(\gamma\alpha\tilde{M}, Y) = \frac{1}{N}\sum_{c=1}^{K}\sum_{i=1}^{n_c}\log\left(1 + \sum_{c'\neq c}^{K}\exp\left(\gamma\alpha[(m_{ic})_{c'} - (m_{ic})_c]\right)\right),$$

$$g(\alpha X, Y) = \frac{1}{N}\sum_{c=1}^{K}\sum_{i=1}^{n_c}\log\left(1 + \sum_{c'\neq c}^{K}\exp\left(\alpha[(x_{ic})_{c'} - (x_{ic})_c]\right)\right).$$

To ensure $\gamma\alpha\tilde{M}$ yields a lower fit loss than $\alpha X$, it suffices that:

$$\forall c, c' \neq c \in \{1, ..., K\}, \ \forall i \in \{1, ..., n_c\} \quad \gamma[(m_{ic})_{c'} - (m_{ic})_c] < [(x_{ic})_{c'} - (x_{ic})_c].$$

Since all $x_{ic}$ are finite, and the differences $(m_{ic})_{c'} - (m_{ic})_c < 0$ for all $c' \neq c$ due to the generalized diagonal superiority of $\tilde{M}$, we can choose $\gamma$ large enough that this inequality holds for all $c, c', i$. Therefore, for this fixed $\gamma$, we have:

$$g(\gamma\alpha\tilde{M}) \leq g(\alpha X) \quad \text{for all } \alpha > 0,$$

where equality only occurs when $\alpha = 0$. The remainder of the proof is identical to that of Theorem 2 found in Appendix B.4.

## C.2 Proof of Theorem A2

As in the proof of Theorem 3, we relabel $L + 1 \to L$ and $\frac{1}{2}\lambda \to \lambda$, and similarly apply Lemma 1 to reduce the loss function to the following form:

$$\mathcal{L}_L(Z) = g(Z, Y) + L\lambda_L\sum_{i=1}^{K}\tilde{s}_i^{\frac{2}{L}},$$

where $\{\tilde{s}_i\}_{i=1}^{K}$ are the singular values of $Z$.

The proof of claim (i) follows through exactly as in the proof of Theorem 3 found in Appendix B.5, and we refer the reader there for details.

**Proof of (ii)**

Assume $L\lambda = o(1)$. We show that, for large enough $L$, any optimal solution $Z_L^*$ must be $(K; n_1, ..., n_K)$ diagonally superior. Let $Z \in \mathbb{R}^{K \times N}$ be any matrix that is not $(K; n_1, ..., n_K)$ diagonally superior. Denote the columns of $Z$ corresponding to the $i^{\text{th}}$ sample of class $c$ by $z_{ic}$. Then the fit term is given by

$$g(Z, Y) = \frac{1}{N}\sum_{c=1}^{K}\sum_{i=1}^{n_c}\log\left(1 + \sum_{c'\neq c}^{K}\exp((z_{ic})_{c'} - (z_{ic})_c)\right).$$

Since $Z$ is not $(K; n_1, ..., n_K)$ diagonally superior, there exists values $c' \neq c \in \{1, ..., K\}$, $i \in \{1, ..., n_c\}$ such that $(z_{ic})_{c'} - (z_{ic})_c \geq 0$. This implies that the fit term is bounded below

$$g(Z, Y) \geq \frac{1}{N}\log(2),$$

and hence the whole loss is also bounded

$$\mathcal{L}_L(Z) \geq \frac{1}{N}\log(2).$$

Now let $Z'$ be any $(K; n_1, ..., n_K)$ diagonally superior matrix, with singular values $\{s_i'\}_{i=1}^K$. Let $z_{ic}'$ denote the column of $Z'$ corresponding to the $i^{\text{th}}$ sample of class $c$. Consider the sequence of solutions $LZ'$ for each $L$. The loss becomes:

$$\mathcal{L}_L(LZ') = g(LZ', Y) + L\lambda_L L^{\frac{2}{L}} \sum_{i=1}^{K}(s_i')^{\frac{2}{L}}.$$

For the fit term, note by definition of diagonally superiority, we have $(z_{ic}')_{c'} - (z_{ic}')_c < 0$ for all $c' \neq c \in \{1, ..., K\}, i \in \{1, ..., n_c\}$. Thus,

$$g(LZ', Y) = \frac{1}{N}\sum_{c=1}^{K}\sum_{i=1}^{n_c}\log\left(1 + \sum_{c' \neq c}^{K}\exp\left(L((z_{ic}')_{c'} - (z_{ic}')_c)\right)\right) = \frac{1}{N}\sum_{c=1}^{K}\sum_{i=1}^{n_c}\log(1 + o(1)) = o(1).$$

Moreover, since $L^{\frac{2}{L}} = 1 + o(1)$ and $\sum_{i=1}^{K}(s_i')^{\frac{2}{L}} = \text{rank}(Z') + o(1)$, we have

$$\mathcal{L}_L(LZ') = o(1) + L\lambda_L(1 + o(1))(\text{rank}(Z') + o(1)) = o(1).$$

Thus, for sufficiently large $L$, the loss of $LZ'$ is strictly less than $\frac{1}{N}\log(2)$, which is a lower bound on the loss of any matrix that is not $(K; n_1, ..., n_K)$ diagonally superior. Therefore, any optimal $Z_L^*$ must be $(K; n_1, ..., n_K)$ diagonally superior.

The remainder of the proof of claim (ii) proceeds exactly as in the proof of Theorem 3 found in Appendix B.5, once it is established that the minimal rank $q$ of a diagonally superior matrix in $\mathbb{R}^{K \times K}$ is the same as the minimal rank $\tilde{q}$ of a $(K; n_1, ..., n_K)$ diagonally superior matrix in $\mathbb{R}^{K \times N}$. First note clearly $\tilde{q} \leq q$ since we could just column repeats of the solution in the diagonally superior case. Also $q \leq \tilde{q}$, since we can just select individual columns from the $(K; n_1, ..., n_K)$ diagonally superior case, and the resulting matrix will have at most the same rank and will be diagonally superior in $\mathbb{R}^{K \times K}$.

### C.3 Proof of Theorem A3

Suppose we have some fixed $(R, \rho)$-STEP imbalanced dataset, and denote the unique global minimizer of the corresponding single layer UFM problem by $Z_\lambda^* \in \mathbb{R}^{K \times N}$. By Proposition 2 in Thrampoulidis et al. [44], we know that the normalized matrix

$$\hat{Z}_\lambda^* = Z_\lambda^*/\|Z_\lambda^*\|_F$$

converges to the corresponding normalized SEL matrix as $\lambda \to 0$.

Note that any SEL matrix has rank $K - 1$, since its column span is identical to the standard simplex ETF. This implies our corresponding normalized SEL matrix, has $K - 1$ non-zero singular values, suppose the smallest is $\mu_{K-1}$. Then, by convergence, there exists a constant $\lambda_0 > 0$ such that when $\lambda < \lambda_0$, the matrix $\hat{Z}_\lambda^*$ has $K - 1$ non-zero singular values, all greater than $\frac{1}{2}\mu_{K-1}$.

Now let $M \in \mathbb{R}^{K \times K}$ be a diagonally superior matrix of rank $r < K - 1$. Using the structure described in the proof of Theorem 1, we know this is achievable whenever $K \geq 4$. Additionally, set the scale of $M$ so that for all $c, c' \neq c$

$$M_{c'c} - M_{cc} < -2.$$

Construct the matrix $\tilde{M} \in \mathbb{R}^{K \times N}$ by simply repeating the columns of $M$ so that the first $n_1$ columns of $\tilde{M}$ are the first column of $M$, and so on. We now compare the loss of $\alpha \hat{Z}_\lambda^*$ to the solution $\alpha \tilde{M}$, with the goal of showing the latter outperforms the former. Again work with the loss written in terms of only the logit matrix:

$$\mathcal{L}(\alpha \hat{Z}_\lambda^*) = g(\alpha \hat{Z}_\lambda^*) + \frac{1}{2}\lambda(L+1)\alpha^{\frac{2}{L+1}}\|\hat{Z}_\lambda^*\|_{S_{\frac{2}{L+1}}}^{\frac{2}{L+1}},$$

$$\mathcal{L}(\alpha \tilde{M}) = g(\alpha \tilde{M}) + \frac{1}{2}\lambda(L+1)\alpha^{\frac{2}{L+1}}\|\tilde{M}\|_{S_{\frac{2}{L+1}}}^{\frac{2}{L+1}}.$$

Since $\hat{Z}_\lambda^*$ is a normalized matrix, its margins cannot be greater than 2, and so we must have $g(\alpha \hat{Z}_\lambda^*) > g(\alpha \tilde{M})$, and so our low rank solution does perform better on the fit term. It is then sufficient that we also outperform on the regularization term, meaning

$$\frac{1}{2}\lambda(L+1)\alpha^{\frac{2}{L+1}}\|\tilde{M}\|_{S_{\frac{2}{L+1}}}^{\frac{2}{L+1}} < \frac{1}{2}\lambda(L+1)\alpha^{\frac{2}{L+1}}\|\hat{Z}_\lambda^*\|_{S_{\frac{2}{L+1}}}^{\frac{2}{L+1}},$$

after dropping positive constants, this is

$$\|\tilde{M}\|_{S_{\frac{2}{L+1}}}^{\frac{2}{L+1}} < \|\hat{Z}_\lambda^*\|_{S_{\frac{2}{L+1}}}^{\frac{2}{L+1}}.$$

For this to hold when $\lambda < \lambda_0$, it is sufficient that

$$\|\tilde{M}\|_{S_{\frac{2}{L+1}}}^{\frac{2}{L+1}} < (K-1)\left(\frac{1}{2}\mu_{K-1}\right)^{\frac{2}{L+1}}.$$

Denoting the singular values of $\tilde{M}$ by $\nu_i$, $i = 1, ..., r$, this becomes

$$\sum_{i=1}^{r} \nu_i^{\frac{2}{L+1}} < (K-1)\left(\frac{1}{2}\mu_{K-1}\right)^{\frac{2}{L+1}}.$$

The LHS of this inequality tends to $r$ as $L \to \infty$, whereas the RHS tends to $K - 1 > r$, hence there exists an $L_0$ such that when $L > L_0$, this inequality is satisfied, and hence $\mathcal{L}(\alpha \tilde{M}) < \mathcal{L}(\alpha \hat{Z}_\lambda^*)$.

Hence no generalized DNC structure in the imbalanced setting is globally optimal for $K \geq 4$, when $\lambda < \lambda_0$ and $L > L_0$, as claimed.

## D   Further experiments

**Further numerical details**: In most of our experiments, we present a single indicative run of the network or model. Since our aim is to demonstrate the suboptimality of DNC, we consider a single run sufficient. The behavior we describe is representative when using high levels of regularization, as we did in our experiments and as is detailed in the figure captions. For the standard regularization experiments, we apply weight decay equally at each layer in both the ResNet-20 backbone and the fully connected head. For experiments on MNIST, we subsample 5,000 examples per class to match the class balance of the CIFAR-10 dataset. Input data is preprocessed by subtracting the mean and dividing by the standard deviation. We use batch gradient descent with batch size 10,000 so as to approximate gradient descent, which is used in the model. However experiments with other batch sizes still lead to low rank solutions when regularization was high.

We also provide here additional experiments that support our theoretical findings:

**Deep UFM experiments:** The left panel of Figure 5 illustrates how the rank of the solution found by gradient descent depends on the width of the network in the deep linear UFM. As predicted by our theory, the rank increases with width. Solutions with ranks between 5 and 9 represent a midpoint

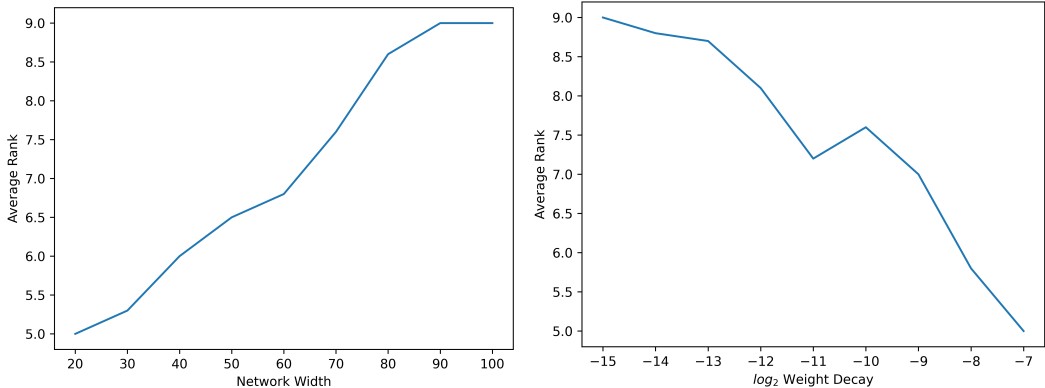

Figure 5: Experiments in the deep linear UFM: **Left**: Rank of converged solution versus width $d$. **Right**: Rank of converged solution versus regularization $\lambda$. Averaged over 10 runs; same hyperparameters as Figure 1.

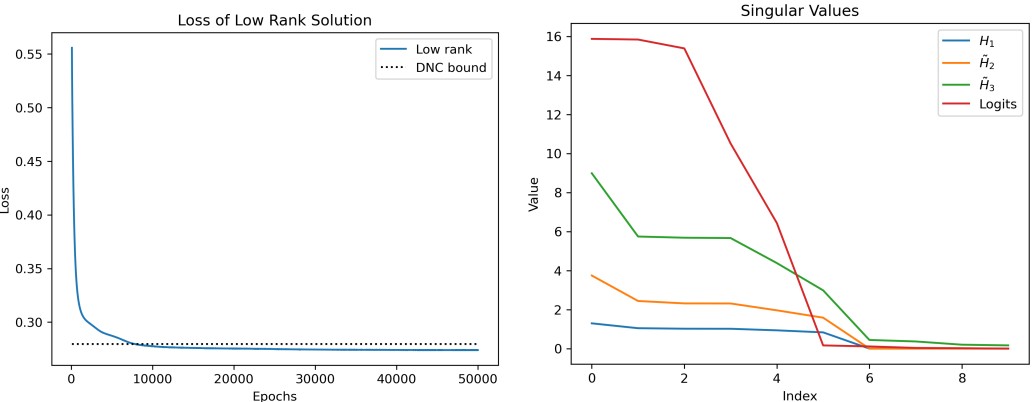

Figure 6: Experiments in the deep ReLU UFM: **Left**: Loss curve for a low-rank solution against the DNC lower bound. **Right**: Singular values of each feature matrix in the fully connected head. Hyperparameters: $L = 3$, $d = 70$, $\lambda = 2^{-9}$, $K = 10$, $n = 5$, learning rate $= 0.5$.

between DNC and our low-rank solution, where a subset of the classes forms a restricted simplex ETF and the remaining classes pair up in a manner similar to the low-rank structure. The right panel of Figure 5 shows that lower levels of regularization are also associated with higher ranks.

Figure 6 presents the experiment in the deep UFM with ReLU activations, again demonstrating that low-rank solutions outperform DNC in this setting.

**Full network experiments:** The corresponding experiment using UFM-style regularization with ReLU activations on the MNIST dataset is shown in Figure 7, with similar implications to those observed for CIFAR-10 in the bottom row of Figure 3. Figure 8 displays the analogous experiment on MNIST using standard regularization, again with implications consistent with those of Figure 4 in the main text.

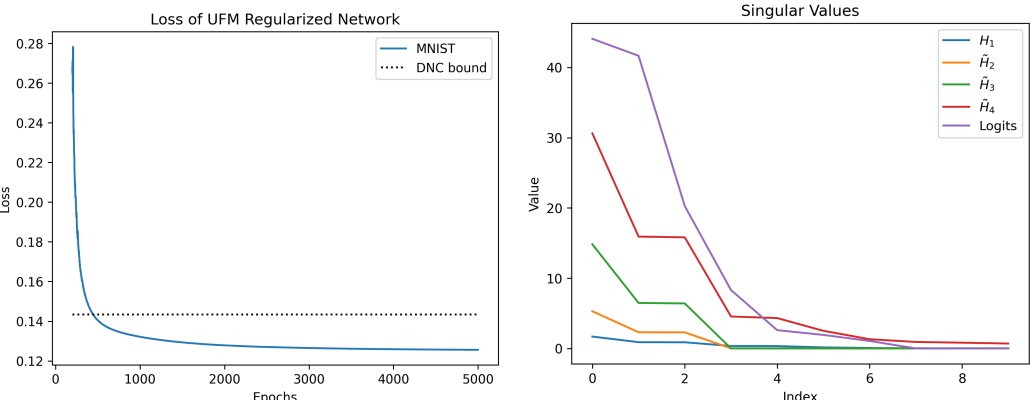

Figure 7: Experiments using UFM-style regularization on MNIST: **Left**: Loss curve for a low-rank solution against the DNC lower bound. **Right**: Singular values of each feature matrix in the fully connected head. Hyperparameters: $L = 4$, $d = 64$, $\lambda_H = 1 \times 10^{-6}$, $\lambda_W = 5 \times 10^{-3}$, learning rate $= 10^{-3}$.

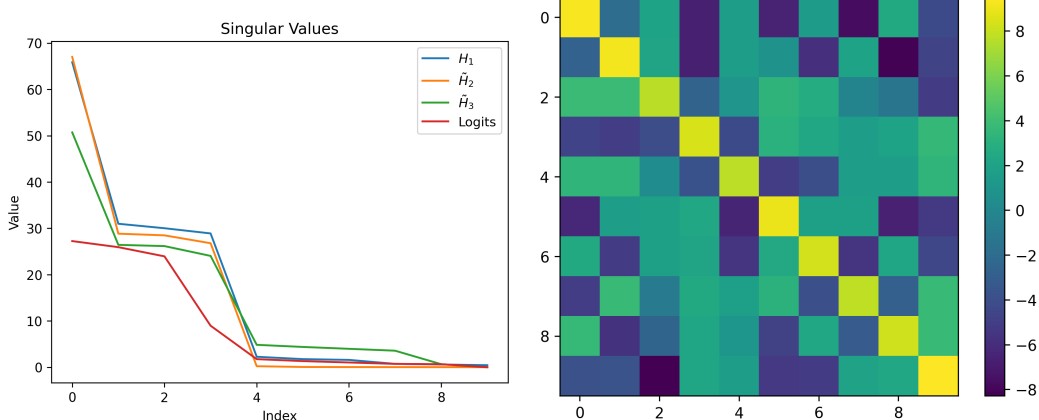

Figure 8: Experiments with standard regularization on MNIST: **Left**: Singular values of each feature matrix in the fully connected head. **Right**: Mean logit matrix. Hyperparameters: $L = 3$, $d = 64$, $\lambda = 10^{-2}$, learning rate $= 10^{-3}$.

