# OpenReview forum: "The Persistence of Neural Collapse Despite Low-Rank Bias"
_NeurIPS.cc/2025/Conference — NeurIPS 2025 poster_

### Official Review · Reviewer_34uB · 2025-06-29

**Clarity:** 3
**Significance:** 3
**Originality:** 3
**Rating:** 4
**Confidence:** 3

**Summary:**

This work makes use of a theoretical tool called Unconstrained Features Model (UFM) to studying the phenomenon of neural collapse (NC). Deep versions exist. Namely deep UFM (DUFM) to study deep neural collapse (DNC).

DNC is not globally optimal once a network trained with weight decay under cross-entropy (CE) loss is deeper than two layers (unless the number of classes is very small, see Theorem 1). Instead, lower rank solutions can be found that achieve lower loss than DNC and Theorem 2 shows how there is a low rank bias for DUFM. The authors then explain why sub-optimal DNC, which is a local minimum or degenerate saddle point, are quite "prevalent" (this is theorems 4 and 5): DNC remains a benign local minimum whose attraction basin grows rapidly with width, whereas the low-rank minima are fewer but achieve smaller objective value.

The finding gives a theoretical explanation for the prevalence of DNC despite the fact that they are not optimal.

**Questions:**

1. UFM has a dizzying number of qualifiers. When these different names refer to the same thing, I cannot be entirely sure. I counted at least six variants:
* Deep MSE UFM
* Deep Linear UFM
* Deep UFM
* Deep CE UFM
* Deep Linear CE UFM
* Deep ReLU UFM
2. The title does not quite reflect the actual message conveyed. It's more that neural collapse persists in spite of a bias for lower-rank solutions, which neural collapse solutions are not.
3. I'm never quite sure what people mean when they say overparametrization as this has become an overloaded term. Do you there are more parameters than sample size? It doesn't sound like it as you don't speak much of sample size. Or do you mean, as I've sometimes seen, that there is a sub-network that is functionally identical? Where exactly do you use the mathematical notion of overparametrization?

**Ethical Concerns:**

["NO or VERY MINOR ethics concerns only"]

**Final Justification:**

I maintain my positive score.

**Limitations:**

Yes

**Quality:**

3

**Strengths And Weaknesses:**

**Strengths**

* The paper tackles a timely, well-posed extension of the neural-collapse literature: it asks whether DNC remains optimal under cross-entropy training with weight decay in deeper architectures—an incremental but natural progression beyond existing one- and two-layer analyses.

**Weaknesses**

* The claim that optimisation *prefers* the sub-optimal DNC over the lower-rank solution is not convincingly supported. The authors only show that DNC occupies a higher-dimensional region of parameter space (i.e.\ is more “prevalent”), but they do not establish a causal link between that prevalence and the trajectories taken by practical optimisers. In other words, dimensional dominance alone does not explain *why* gradient-based training should consistently land in the DNC basin.

---

Overall, my confidence in this assessment is limited because I am not deeply familiar with the theoretical NC/DNC, UFM/DUFM literature.

---

> ### Author Rebuttal · Authors · 2025-07-29
>
> We thank the reviewer for the thoughtful feedback and constructive suggestions. Below, we address the main concerns and outline the specific changes we plan to make in the revised version of our paper.
>
> **The claim that optimisation prefers the sub-optimal DNC over the lower-rank solution is not convincingly supported**
>
> We appreciate this important point. We agree that our current analysis does not establish a causal link between the prevalence of DNC and the behaviour of particular optimization algorithms such as SGD. Rather our goal is to provide a heuristic description of how first-order properties of the loss landscape encourage the frequent occurrences of DNC during training.
>
> To clarify:
> * Theorem 4 shows that DNC solutions are critical points with positive semi-definite Hessians. While this does not imply convergence, it confirms that such solutions are reachable by local descent methods.
> * Theorem 5 shows that the dimension of the parameter space corresponding to DNC configurations grows faster than that of low-rank alternatives as network width increases.
> * Theorem 6 shows that both DNC and our low-rank minima have comparable local flatness, with their flatness measures converging to a constant ratio as regularization decreases.
>
> Taken together, these results provide a heuristic argument that DNC occupies a larger volume in the loss landscape and is therefore more likely to be encountered by local search methods in overparameterized regimes—assuming no strong inductive biases. Of course, many other factors affect optimization trajectories in practice, including the choice of optimizer, initialization scale and batch size. There are also higher-order landscape properties to account for such as dynamical accessibility. A rigorous dynamical explanation would likely require tools such as gradient flow analysis that can model these influences explicitly.
>
> That said, we believe our landscape-based approach remains impactful: it highlights how one specific property of the optimization problem—geometry— can encourage convergence to DNC, and represents a first step in this direction within the literature. To support this heuristic view, we also include empirical results in Figure 2, which show that the likelihood of DNC increases with width and decreases with regularization, consistent with the predictions of our geometric analysis.
>
> We will clarify this point in the paper and revise the relevant discussion to better communicate the scope and limitations of our conclusions. Specifically, we will emphasize that:
>
> * Our results provide landscape-level insight, not a rigorous theoretical claim.
> * Our empirical findings support the idea that DNC becomes more likely under conditions predicted by the landscape analysis.
>
> We would welcome any suggestions on how to make this distinction more transparent in the revised manuscript.
>
> **UFM has a dizzying number of qualifiers… I counted at least six variants**
>
> Thank you for highlighting this. We agree that the terminology has become confusing due to the number of variants. In the revised manuscript, we will streamline the naming convention and clearly distinguish the two central models analysed: the deep UFM with cross-entropy loss, using either linear or ReLU activations. References to MSE-based variants will be reduced and redundant qualifiers will be removed unless essential.
>
> **The title does not quite reflect the actual message… it's more that NC persists in spite of low-rank bias**
>
> We appreciate this insight. We agree that the central message is that neural collapse persists despite a bias toward lower-rank solutions. The phrase "under low-rank bias" in the title was intended to convey this tension, but we now see that it may be too ambiguous. In response, we are happy to revise the title to more directly reflect the paper’s core message. If the reviewer has a specific change in mind, could they please let us know.
>
> **What exactly do you mean by overparameterization? Is it sample size? Or functional redundancy?**
>
> We appreciate the reviewer raising this point. “Overparameterization” is indeed an overloaded term and we will clarify our usage. In our paper, we adopt the definition used in the original neural collapse work of Papyan et al. [34]: A network is overparameterized if it has enough capacity to fit the training data to near-zero training error.
>
> This aligns with our use in both theory and experiments:
> * In the UFM, overparameterization is built-in, since features are treated as free optimization variables, and perfect fitting is always achievable.
> * In real-world networks, such as our experiments with ResNet-20, overparameterization means the model achieves near-zero training loss on the considered dataset, such as CIFAR-10 and MNIST in our case.
>
> In the revision, we will explicitly define our usage of the term early in the paper.
>
>
> We hope these clarifications help reinforce the contributions of our work and the care we have taken in interpreting and presenting our findings. We thank the reviewer again for their valuable feedback and for helping improve the clarity and impact of our paper.

---

### Official Review · Reviewer_6xQX · 2025-06-30

**Clarity:** 3
**Significance:** 3
**Originality:** 2
**Rating:** 4
**Confidence:** 4

**Summary:**

This paper studies the optimality of DNC in deep UFM, and prevalence of DNC udner cross-entropy loss. The authors first show that DNC solutions are not in general optimal under this setting, and then prove properties of the loss landscape that intuitively indicates the prevalence of DNC.

**Questions:**

1. Please also refer to the Strengths and Weaknesses part.

2. Since the results in section 3.3. requires the regularization factor $\lambda \rightarrow 0,$ I wonder if it is possible to prove implicit bias results for GD/GF as in [4,5], and understand whether when those training dynamics are implicitly biased towards NC/DNC solutions?

3. While the DNC solutions are not optimal as proved in this paper, is it possible to show that DNC are partially optimal,  e.g. DNC1 is still optimal?

*References:*

[4] "Gradient Descent Maximizes the Margin of Homogeneous Neural Networks." Kaifeng Lyu, Jian Li

[5] "An unconstrained layer-peeled perspective on neural collapse." Wenlong Ji, Yiping Lu, Yiliang Zhang, Zhun Deng, Weijie J Su

**Ethical Concerns:**

["NO or VERY MINOR ethics concerns only"]

**Final Justification:**

This paper studies the optimality of deep neural collapse (DNC) solution under cross-entropy loss. It provides interesting results on the loss landscape, showing the non-optimality of the DNC solution, and the intuitive cause of the prevalence of DNC, which I've pointed out in the original *Strengths* part.

The first two points in my origin *Weaknesses* part include the comparison with [1], and the definition of DNC, and these issues are addressed in the authors' rebuttal. In particular, this paper provides a structural characterization of the global optima of the deep UFM under cross-entropy, and technically it could also be interesting to potential future works in this direction.

The final concern is that this paper still does not provide any direct evidence on the prevalence of the DNC through its analysis, but from the authors' rebuttal, we know that all local minimas are DNC1 with some other structural properties, and this compensates some of the weaknesses in my opinion.

Thus, I would like raise my score to 4, and recommend an acceptance.

**Limitations:**

There's no potential negative societal impact of this work in my opinion. The authors adequately addressed the limitations in Section 5.

**Paper Formatting Concerns:**

I haven't noticed any major formatting issues in this paper.

**Quality:**

3

**Strengths And Weaknesses:**

### Strength

1. This paper studies DNC for cross-entropy loss, which I think is a well-motivated problem, and is new in the literature.

2. The paper provide high level intuitions on the non-optimality of the DNC solution, and the cause of prevalence of DNC, which I think is interesting.

3. The paper is well-written, and is easy to follow the main points.

### Weaknesses

1. While I think the study of  DNC for cross-entropy loss is interesting, this paper focus more on studying the optimality of DNC, which I feel is less significance than explaining the prevalence of DNC. In particular, the non-optimality of DNC under MSE loss is already studied in [1], and cause of non-optimality of DNC in this paper is the same as the one in [1], which is the low-rank bias of the problem. In my opinion, this results do not provide much fundamentally new insights compare to [1]. Besides, the study of  the optimality of DNC is not that well-motivated in my opinion, as NC or DNC is just an experimental phenomenon observed empirically, and not much empirical evidence indicates the global optimality of NC or DNC in practice to my best knowledge.

2. I'm not entirely sure the non-optimality of DNC is due to the low-rank bias as claimed in the paper, or is due to the fact that the strong definitions.  (1)In Definition 1, DNC is defined as all the last $L$ linear layers are collapse, this seems a bit strong to me, since it is possible that e.g. the level of collapse is inceasing by concatenating more linear layers, as observed in [2] for MSE loss. (2) Also, in Definition 1, it also requires the features after applying the activation functions also satisfies NC2, which seems a bit strong, since a feature that is NC2 may not be NC2 any more after apply e.g. ReLU, and in that case other structure such as orthogonal frame could be optimal as discussed in [3] for MSE loss.

3. The paper discuss the prevalence of NC1 in section 3.3, however, those results are not direct cause of the occurrence of the DNC solutions. In particular, Theorem 4 only shows the existence of DNC solution as a non-supirious local minima, and Theorem 5,6 only provide high-level intuitions. I wonder it is possible to provide some more direct evidences, such as claims like: all local minima with small loss are DNC solutions, or gradient descent/gradient flow is implicitly biased towards DNC solutions.


*References*:

[1] "Neural Collapse versus Low-rank Bias: Is Deep Neural Collapse Really Optimal?" Peter Súkeník, Marco Mondelli, Christoph Lampert

[2] "Wide Neural Networks Trained with Weight Decay Provably Exhibit Neural Collapse." Arthur Jacot, Peter Súkeník, Zihan Wang, Marco Mondelli

[3] "Extended Unconstrained Features Model for Exploring Deep Neural Collapse." Tom Tirer, Joan Bruna

---

> ### Author Rebuttal · Authors · 2025-07-29
>
> We thank the reviewer for their thoughtful and detailed feedback. Below, we address all raised weaknesses and questions. We hope this resolves many of their concerns, and kindly ask that they consider raising their scores if satisfied.
>
> **the non-optimality of DNC under MSE loss is already studied in [1] … In my opinion, this results do not provide much new insights compare to [1]**
>
> We appreciate the opportunity to clarify the distinction between our work and that of Sukenik et al. [40], which we regard as foundational.
>
> Sukenik et al. introduce the notion of low-rank bias and offer heuristic arguments and empirical observations supporting its role in DNC suboptimality under MSE loss. Their primary theoretical result on this topic is their Theorem 5, which constructs a specific low-rank solution—termed the SRG solution—that achieves lower loss than DNC.
>
> Their result is example-based, relying on a single explicit construction, without analysing global minima or how the loss landscape changes. Crucially, their work does not preclude the possibility that the global minimum is not a low rank solution, making it unclear theoretically whether low rank bias is the cause of suboptimality.
>
> In contrast, we provide a complete theoretical characterization of how low-rank bias manifests across the loss landscape, and why it leads to the systematic suboptimality of DNC. Specifically:
>
> * Theorem 2 shows that for sufficiently large depth L, any lower-rank, diagonally superior matrix will eventually outperform any higher-rank structure—including DNC. This generalizes the insight in [40] from a specific pairwise comparison (DNC vs. SRG) to a global classification of how low-rank bias affects the loss surface.
> * Theorem 3 shows that global optima must exhibit approximate low-rankness: all but a small number of singular values decay exponentially with depth. This demonstrates that low-rank bias governs the structure of global minima, not just the existence of an isolated better solution.
> * Theorem 4 proves that DNC remains a local minimum for small regularization. Thus, the loss surface for L>1 is not a strict saddle function, in contrast to the L=1 case shown in prior works [52,54]. This indicates that the fundamental geometry of the optimization landscape changes with depth, beyond just a shift in which solution is optimal.
>
> None of these deeper structural or global results appear in [40]. While thematically related, our work turns the intuition in [40] into a rigorous theory. We also go beyond their scope by analysing the prevalence of DNC, which we discuss in response to another of your questions.
>
> We thank the reviewer for pointing out that the distinctions between our work and [40] could be more clearly highlighted, and we will revise the manuscript to emphasize these differences more explicitly.
>
> **The study of the optimality of DNC is not that well-motivated in my opinion, as DNC is just an experimental phenomenon … not much empirical evidence indicates global optimality.**
>
> We agree that prevalence is an important direction. We also agree that NC and DNC have not had their optimality explored directly via experimentation. This is likely because they arise in highly overparameterized neural networks, where the loss surface is highly non-convex and empirically identifying optimal solutions is not a realistic goal.
>
> However, we believe exploring optimality remains a valuable research direction, particularly in theoretical studies. The optimality of NC and DNC has been a central topic in numerous theoretical works—just among the references in our paper, this includes [1,3,6,7,14,20,22,28,30,40,41,42,43,48,49,52,53,54]. In many of these works, the optimality of NC/DNC within specific models is used as evidence for, and a theoretical explanation of, its prevalence. Until the work by Sukenik et al. [40] and our own, optimality and prevalence were treated as closely intertwined.
>
> Moreover, the extent to which DNC is suboptimal appears to directly impact its prevalence. This is demonstrated empirically in the right panel of Figure 2: as regularization increases, the gap in performance between DNC and lower-rank solutions increases, and the figure shows the prevalence of DNC correspondingly decreases.
>
> If desired, we can also elaborate on the further research directions made possible as a direct consequence of our observation of suboptimality, though we believe the above arguments sufficiently justify studying optimality.
>
> **In Definition 1, DNC is defined as all the last L linear layers are collapse, this seems a bit strong to me, since it is possible that the level of collapse is increased by concatenating more linear layers**
>
> Thank you for this perceptive comment. Definition 1 is to be interpreted as emerging ‘in the limit of an overparametrized network as training continues’. What we mean by this is it will only approximately hold in real networks, since they are not infinitely expressive or trained for infinite time. This is similar to the description of NC in Papyan et al. [34] which is stated as a limit rather than something that is literally achieved. We agree with your observation that in practical settings, collapse typically intensifies with depth—this is compatible with our interpretation of Definition 1 as an asymptotic phenomenon.
>
> In contrast, Definitions 2 and 5, which formalize DNC within the deep UFM, do require collapse at each layer. This is since the deep UFM, due to the freely optimized features, represents the overparameterization limit, and consideration of critical points represents the training limit.
>
> This is formalized in Lemma 1, which shows that at any critical point the rank of any intermediate layer's features must match that of the logit matrix. Therefore, either all layers will be approximate, or none will be. Empirically, this is precisely what we observe in deep UFMs: when DNC occurs, all layers collapse up to a numerical threshold.
>
> We hope this makes clear that this definition is not restrictive under the modelling assumptions. We will revise Definition 1 to make clear that for real networks this is considered as an approximate relation.
>
> **Definition 1 requires the features after applying the activation functions also satisfies NC2, which seems a bit strong, since a feature that is NC2 may not be NC2 any more after apply ReLU, and in that case other structure such as orthogonal frame could be optimal**
>
> Our Definition 1 is adapted from empirical studies of DNC [35,37], which explicitly examine features after ReLU and report consistent evidence of the DNC2 property. To the best of our knowledge, all numerical experiments studying DNC and NC have considered the output of the activation function. Additionally, in both our experiments and prior work on deep UFMs [40,41], whenever DNC2 emerges, it remains preserved after nonlinearity.
>
> We also note that the only theoretical result in our paper where this distinction is relevant is Theorem 7, as the issue does not arise in the deep linear UFM setting.
>
> Regarding your comment on orthogonal frames: our definition of DNC2 is intentionally broad and is compatible with both simplex ETFs and orthogonal frames. Any orthogonal frame, once globally centred, becomes a simplex ETF. In particular, the work by Tirer et al. [43] shows orthogonal frame structure before centring—these still satisfy our DNC2 condition after centring. This is why we impose no constraints on the centring vectors $μ_G^{(l)}$, $μ ̃_G^{(l)}$ in Definition 1.
>
> We would be happy to revise the manuscript to clarify these points, and we are of course open to further discussion if any part of our definition or response remains unclear.
>
> **The paper discuss the prevalence of NC1 in section 3.3 … those results are not direct cause of the occurrence of the DNC solutions … I wonder it is possible to provide some direct evidences**
>
> We thank the reviewer for this important comment. While our current results do not constitute a direct dynamical explanation for why optimization converges to DNC, they aim to provide landscape-level insight into why DNC frequently emerges during training.
>
> Specifically:
>
> * Theorem 4 shows that DNC is a local minimum, making it reachable by local search methods.
> * Theorem 5 demonstrates that the dimension of the parameter space corresponding to DNC grows faster with width than that of low-rank alternatives, suggesting DNC occupies a larger portion of the landscape.
> * Theorem 6 shows that both DNC and our low-rank minima have comparable local flatness as regularization decreases.
>
> Though not a formal justification, our results reveal structural biases favouring DNC under first-order methods in wide, regularized networks. We support this argument with empirical findings in Figure 2, where DNC becomes more frequent as predicted.
>
> We agree that a full dynamical explanation would be stronger, we see our work as a step toward such rigorous understanding. We are happy to revise the manuscript to clarify the heuristic nature of our argument.
>
> **Is it possible to prove implicit bias results for GD/GF, and understand whether those training dynamics are implicitly biased towards NC/DNC solutions?**
>
> First, we clarify that only Theorem 6 concerns the $λ→0$ limit; Theorems 4 and 5 hold for $λ<λ_0$ for some $λ_0$.
>
> Ji et al. [20] show for L=1 that the UFM will tend to a KKT point of a constrained minimal nuclear norm problem. Appendix G of the work of Lyu and Li implies when using more layers the nuclear norm is replaced with a Schatten quasi-norm, mirroring our analysis. A similar result to [20] likely could be attained in the unregularized setting.
>
> **Is it possible to show that DNC are partially optimal**
>
> Yes, we believe DNC1 remains optimal for CE, and arguments from [40] in the MSE setting likely extend.
>
> We appreciate the reviewer’s detailed feedback and hope our responses have addressed their concerns. If there are further questions, we’d be happy to discuss.

---

> > ### Comment · Reviewer_6xQX · 2025-08-03
> >
> > Thank you for the detailed reponse that address most of my questions and concerns!
> >
> > However, I still have questions on Theorem 4. While I appreciate the authors explanation, my point about Theorem 4 is that it only proves the **existence** of a DNC solution as a local minima, as written in line 216 "There exists solutions with DNC structure that are critical points of the model."
> >
> > Thus I wonder:
> > 1. Is it true that there exist local minimum that are not DNC ? If yes, is it there any structral properties for those solutions that are not DNC?
> > 2. The passage under below Theorem 4 say "This result makes clear that although DNC solutions no longer correspond to global minima, they correspond to local minima or degenerate saddle point." Does this means that all DNC solutions must be local minima  or degenerate saddle point? If yes, how could I see this from Theorem 4?
> >
> > Thank you again for your time!

---

> > > ### Author Response · Authors · 2025-08-04
> > >
> > > Thank you again for your thoughtful and constructive questions. We sincerely appreciate the time and effort you are devoting to evaluating our paper. Below, we aim to clarify the remaining points you raised concerning Theorem 4 and the nature of DNC and non-DNC local minima.
> > >
> > > **Is it true that there exist local minimum that are not DNC?**
> > >
> > > Yes, there definitely exist local minima that are not DNC. Since Theorem 1 shows that DNC solutions are not global minima, it follows that whatever configuration achieves the global minimum must itself be a local minimum that is not DNC.
> > >
> > > If your question instead refers to the existence of other non-global local minima that are not DNC, then we believe so as well. Take the structure described in Equation (5), just below line 163, for example. It does not correspond to a DNC solution, and an application of Theorem 2 shows that it is suboptimal when the number of layers L is sufficiently large. In the proof of Theorem 6, we show that this structure is a critical point, though we do not prove that its Hessian is positive semi-definite. However, the empirical evidence in Figure 1 shows that this solution is reachable from random initialization via gradient descent, which makes it unlikely to be a saddle point.
> > >
> > > More generally, in our experiments with gradient descent, we observed convergence to many different structures that are not DNC.
> > >
> > > **If yes, is it there any structural properties for those solutions that are not DNC?**
> > >
> > > This is a fascinating question. We find that it is DNC2—the property of features forming a simplex equiangular tight frame (ETF)—that fails in these suboptimal solutions. Local minima that are not DNC have feature matrices of rank strictly less than K−1, meaning they cannot form a simplex ETF. Instead, their features form other lower-rank configurations, though these can vary significantly.
> > >
> > > Despite this variability, we do observe some consistent empirical patterns:
> > > * The feature matrix always has rank less than K−1, as predicted by the low-rank bias.
> > > * The features always satisfy DNC1.
> > > * The class-mean matrix is always symmetric.
> > > * The feature means always lie on a common hypersphere.
> > >
> > > These observations align with the hypotheses proposed by Liu et al. in *"Generalizing and Decoupling Neural Collapse via Hyperspherical Uniformity Gap"*, where they show that under small width constraints—where DNC is impossible—the optimal configuration tends to uniformly distribute the feature means on a hypersphere. Investigating the exact geometric characterization of these non-DNC local minima is promising future work.
> > >
> > > **The passage below Theorem 4 say "This result makes clear that although DNC solutions no longer correspond to global minima, they correspond to local minima or degenerate saddle point." Does this means that all DNC solutions must be local minima or degenerate saddle point? If yes, how could I see this from Theorem 4?**
> > >
> > > Thank you for this excellent clarification prompt. We realize now that this point was not explained as clearly as it should have been.
> > >
> > > There are many possible configurations that satisfy the DNC definition, but not all of them are critical points of the loss function. What Theorem 4 guarantees is that there exist solutions satisfying Definition 2 (i.e., DNC structure) that are critical points with positive semi-definite Hessians. These are thus either local minima or degenerate saddles.
> > >
> > > However, Theorem 4 does not claim that every configuration satisfying Definition 2 is a critical point. In fact, this is not the case. Whether a DNC configuration is a critical point depends crucially on the scale of the logit matrix $Z$. The definition of DNC is invariant to rescaling of $Z$, since it only concerns the normalized structure $\hat{Z}=Z/ \| Z \|_F$. But being a critical point (and having a positive semi-definite Hessian) requires that the norm $\|Z\|_F$ also takes on a specific value that balances the loss and regularization.
> > >
> > > So in summary:
> > > * Not all DNC solutions are critical points.
> > > * Theorem 4 shows that some DNC configurations are critical points with positive semi-definite Hessians.
> > > * These correspond to the specific scale $\| Z \|_F$ that yield criticality.
> > >
> > > We shall also note that empirically if you initialize near a DNC configuration but with incorrect scale, gradient descent tends to adjust $\| Z\|_F$ toward the critical scale while preserving the frame $\hat{Z}$, thereby converging to a DNC solution that is a critical point.
> > >
> > > We hope these clarifications fully address your concerns and provide a clearer understanding of the landscape of local minima. Once again, we greatly appreciate the care and time you are investing in your review, and we welcome any further questions or suggestions you may have.

---

> > > > ### Comment · Reviewer_6xQX · 2025-08-04
> > > >
> > > > Thank you for the detailed explanation, now I have a better understanding on the utility and the scope of the results. Thus, I would like to raise my score and recommend acceptance of the paper.

---

### Official Review · Reviewer_esms · 2025-07-01

**Clarity:** 4
**Significance:** 3
**Originality:** 3
**Rating:** 5
**Confidence:** 4

**Summary:**

The paper analyzes the deep neural collapse phenomenon, showing it is not globally optimal under cross-entropy loss. The proof relies on theoretical results on the implicit bias of deep matrix factorization with L2 regularization. The paper then claims that DNC emerges in practice due to large width, as it is more prevalent in the loss surface relative to low-rank solutions. Overall, the paper manages to connect three separate ideas in deep learning theory: (1) neural collapse, (2) implicit bias in matrix factorization, and (3) large-width behavior.

**Questions:**

1. In the definition of DNC, you assume that all layers have fixed equal rank. Isn’t DNC characterised through gradual collapse towards lower rank matrices as we progress through the layers? Isn’t that what we actually see empirically in deep networks?
2. What’s the motivation for studying diagonally superior matrices?
3. In what sense is the relu analysis/results different than the deep linear case?
4. Have you explored whether low-rank solutions generalize better, worse, or similarly compared to DNC solutions in practical networks?
5. How do optimizer choices (say SGD vs. Adam) affect the low-rank bias? Isn't the low-rank bias due to SGD? (see for example "Implicit Regularization in Matrix Factorization")

**Ethical Concerns:**

["NO or VERY MINOR ethics concerns only"]

**Final Justification:**

The authors have addressed all of my concerns and I retain my very positive score.

**Limitations:**

Yes

**Quality:**

4

**Strengths And Weaknesses:**

Strengths
1. The paper is well-written and easy to follow, both the theoretical part and the experiments.
2. The theory is accompanied by experiments both on toy data and real networks.

Weaknesses
1. The last two paragraphs before the related works subsection are a little too dense and hard to digest. Perhaps the authors might want to summarise their contributions in a bullet list.
2. The experiments are somewhat simple.
3. The relation between DNC and low-rank implicit bias was already established for MSE by Sukenik et al.
4. The analysis relies on the Unconstrained Feature Model, which assumes full expressivity (but almost all prior work do that as well).

---

> ### Author Rebuttal · Authors · 2025-07-29
>
> We thank the reviewer for their constructive and thoughtful comments. We appreciate the time and care taken in reviewing our submission and respond to your points in turn below.
>
> **The last two paragraphs before the related works subsection are a little too dense and hard to digest**
>
> We appreciate the suggestion to improve readability at the end of the theory section. We agree that the current presentation would benefit from improved clarity. In the revised manuscript, we will restructure this portion using bullet points to better summarize our key contributions and improve readability.
>
> **The experiments are somewhat simple**
>
> Thank you for this observation. Our primary aim in the experimental section was not to provide a broad empirical study, but rather to validate specific theoretical predictions made by our model—namely, the suboptimality of DNC, its increased prevalence with width, and the role of regularization strength. We intentionally designed the experiments to isolate these effects in controlled settings, highlighting the theoretical phenomena without confounding variables.
> While the experiments are deliberately minimal to remain closely aligned with the theory, we agree that extending the analysis to more complex architectures and training regimes would be a valuable direction for future work.
>
> **The relation between DNC and low-rank implicit bias was already established for MSE by Sukenik et al.**
>
> Thank you for highlighting this point. We fully agree that the connection between low-rank bias and the suboptimality of DNC under MSE loss was first introduced by Sukenik et al. [40], whose work we greatly respect. Their central theoretical contribution—Theorem 5 in their paper—shows that DNC is outperformed by a specific low-rank construction they term the SRG solution. However, their analysis does not characterize how low-rank bias shapes the broader structure of the loss landscape, nor do they make claims about the rank or geometry of global minima.
>
> Our work significantly extends this line of inquiry—not only to CE loss, but also in terms of the generality and theoretical completeness of the results:
>
> * Theorem 2 proves that for sufficiently large depth $L$, any lower-rank, diagonally superior structure eventually outperforms any higher-rank structure (including DNC). This generalizes the single comparison result (SRG vs DNC) of [40] to a full structural characterization.
> * Theorem 3 shows that global minima must be approximately low-rank, with all but a few singular values decaying exponentially in $L$. This demonstrates that low-rank bias governs the structure of global optima, not just isolated examples.
> * Theorem 4 establishes that DNC remains a local minimum or degenerate saddle point when regularization is sufficiently small. As a result, the loss surface for $L>1$ no longer satisfies the strict saddle property, which does hold for both MSE [52] and CE [54] when $L=1$.This reflects a fundamental shift in the geometry of the optimization landscape with increasing depth.
>
> While both works reveal a tension between DNC and low-rank structures, we believe our results offer a more general and theoretically complete picture. We are happy to revise the manuscript to ensure this relationship and distinction with [40] is more clearly explained.
>
> **The analysis relies on the Unconstrained Feature Model, which assumes full expressivity.**
>
> We agree that the assumptions of the unconstrained feature model must hold for our theoretical results to apply directly to real neural networks. To support this connection, we included experiments using highly expressive architectures that approximate the UFM regime. In addition, there is broad empirical evidence in the literature suggesting that the implications of the UFM hold in practice when regularization is matched to the assumptions of the model. Indeed, most of the UFM-based works cited in our related work section numerically validate its predictions on real data using expressive networks.
>
> That said, moving beyond the UFM and relaxing its assumptions would be an important and valuable direction for future work.
>
> **You assume that all layers have fixed equal rank. Isn’t DNC characterised through gradual collapse towards lower rank matrices as we progress through the layers? Isn’t that what we actually see empirically in deep networks?**
>
> Yes, your description is correct. In empirical experiments with overparametrized neural networks, the level of DNC—measured by various metrics—typically increases as one moves deeper into the network (see references [13,35,37]). The feature matrices at each layer tend to exhibit decreasing effective rank, eventually approaching that of a simplex ETF, which is either $K$ or $K-1$ depending on whether global centring occurs.
>
> In the deep UFM, the modelling assumptions treat each layer as effectively ‘infinitely deep’, meaning it can collapse exactly. As a result, there is no need to model the gradualness of collapse across layers; it is implicitly captured by the framework. Some prior works, such as Perturbation Analysis of Neural Collapse by Tirer et al., aim to make this more realistic by introducing additional constraints and observing increasing collapse depth by depth.
>
> We also provided a definition of DNC for real neural networks in Definition 1 of the background section. This definition is intended to apply in the overparameterization and training limits, and is therefore compatible with the gradual emergence of DNC in networks that do not literally attain those limits. We would be happy to revise the manuscript to clarify this point.
>
> **What’s the motivation for studying diagonally superior matrices?**
>
> Diagonally superior matrices arise naturally when considering cross-entropy loss on hard labelled data. In the unregularized case, any diagonally superior logit matrix can achieve arbitrarily small loss by making the matrix scale sufficiently large. This property emerged in our theoretical analysis, motivating us to study the rank structure of these matrices.
>
> **In what sense is the relu analysis/results different than the deep linear case?**
>
> The inclusion of the ReLU activation function makes the analysis significantly more challenging. This is because ReLU can raise and lower the rank of a matrix, and as a result, there is no analogue of Lemma 1 in the ReLU setting. In particular, the rank of the feature matrix can vary as one move through the network, unlike in the linear case where rank is preserved.
>
> In terms of results, the ranges of $K$ and $L$ for which we can guarantee that DNC is suboptimal (as shown in Theorems 1 and 7) are narrower in the ReLU case, reflecting the reduced analytical tractability. However, the essential conclusion—that DNC is not globally optimal—remains the same in both the linear and ReLU settings for those parameter regimes.
>
> The other results (Theorems 2–6) are not directly available in the nonlinear case, again due to the greater analytical difficulty. That said, we anticipate that the insights and structural intuitions from the linear case still carry over and may inform future theoretical developments in this more complex setting.
>
> **Have you explored whether low-rank solutions generalize better, worse, or similarly compared to DNC solutions?**
>
> When training networks on CIFAR10 and MNIST, we found that both DNC solutions and low-rank solutions achieve similar performances on the test set. However, we did not conduct a systematic study of generalization behaviour. Exploring how generalization—and other properties such as robustness—compare would be an interesting direction for future work. This is particularly relevant in light of the original claims by Papyan et al. [34], who suggest that neural collapse improves deployment properties; low-rank solutions provide a natural control group for testing such claims.
>
> **How do optimizer choices affect the low-rank bias? Isn't the low-rank bias due to SGD?**
>
> We conducted experiments using both batch SGD and Adam, and in both cases, we observed that low-rank solutions can emerge. However, we did not perform a systematic analysis of how optimizer choice affects the frequency or precise rank of converged solutions. In the limitations section of our paper, we note that several important practical factors—such as optimizer choice, batch size, and initialization scale—merit further exploration.
>
> Regarding the second part of your question: there is a distinct line of work studying implicit bias, where optimization dynamics alone (typically of SGD or gradient flow) shape the solution in the absence of explicit regularization. The work by Gunasekar et al., which you reference, falls in this category.
> In contrast, our analysis focuses on bias that is explicit in the loss function, arising directly from the L2 regularization term, as made clear in our theorems and proofs. We chose to focus on this regularized regime, but there are interesting follow-ups related to what happens in the absence of explicit regularization.
> For example, Ji et al. [20] study the single-layer CE UFM without regularization and show a bias toward minimizing the nuclear norm of the logit matrix. Similarly, the analysis in Appendix G of *Gradient Descent Maximizes the Margin of Homogeneous Neural Networks* by Lyu and Li suggests that for deeper models, this implicit bias transitions to a Schatten quasi-norm—closely resembling the explicit bias in our formulation.
>
>
> Once again, we sincerely thank the reviewer for the encouraging assessment and valuable suggestions. We will revise the manuscript to clarify our contributions relative to prior work, improve the exposition in dense sections, and more explicitly communicate the theoretical and empirical implications of our results. We appreciate the opportunity to strengthen the paper and are grateful for the reviewer’s time and insight.

---

### Official Review · Reviewer_9eBN · 2025-07-02

**Clarity:** 3
**Significance:** 3
**Originality:** 3
**Rating:** 5
**Confidence:** 4

**Summary:**

This paper investigates why deep neural collapse (DNC)—a symmetric geometric structure often seen in trained deep networks—frequently emerges, despite being theoretically suboptimal. Building on prior work that showed DNC is not globally optimal under MSE loss due to low-rank bias, the authors extend this result to the more realistic setting of cross-entropy loss, including models with ReLU activations.

They prove that as network depth increases, optimal solutions become increasingly low-rank, with most, but not all, singular values of the output matrix suppressed exponentially. Although DNC has full rank (or rank-one deficiency for simplex ETFs) and thus incurs higher regularisation cost, it remains a frequent outcome in training. The authors explain this paradox by showing that DNC corresponds to stable critical points with its solution space grows faster with width than that of low-rank alternatives.

Empirical results on synthetic models (UFMs) and real datasets (MNIST, CIFAR-10) confirm the theory: low-rank solutions achieve lower loss, but DNC still frequently arises, especially with small regularisation and wide layers. The work offers the first rigorous explanation for the persistence of DNC in deep learning, linking it to geometry, low-rank bias, and network overparameterisation.

**Questions:**

1) Theorem 1 addresses also the $L=2$ case. However, the work in [43] proves that DNC is optimal for $L=2$ under MSE loss, and further suggests that the result could extend to CE loss. In contrast, your result appears to contradict this, showing that DNC is suboptimal for CE even when $L=2$. Could you clarify whether this discrepancy is entirely due to the change in loss function (MSE vs CE), or if there are additional structural or technical reasons that break the equivalence? It would be interesting to better understand how the choice of loss affects optimality—potentially highlighting why the generalisation of [43] to CE (and potentially other losses, as suggested in [53] for the L=1) may not hold. A brief discussion on this as the authors do wrt the result from [6] for Theorem 1 would be helpful.

2) The authors mention that the single-layer UFM case ($L=1$) cannot capture the optimal structure of the last-layer for deep networks, since the last-layer weights have an optimal structure different from what NC claims (a simplex ETF). Am I correct in assuming that this is specific to CE? For instance, under MSE, the fit-optimal solution would need to have the same rank as $Y$ (true labels) to fully minimise the loss, implying that a lower-rank solution, specifically in the last-layer, (as allowed under CE) wouldn’t suffice. This suggests that optimality is strongly loss-dependent—particularly given CE’s focus on margin maximisation versus the regression-type MSE. Also, for the CE case, I can see this as an attribute of a deep linear network (from Lemma 1), but not sure how/if this generalises when non-linearity takes effect.

3) The authors present a construction of diagonally superior matrix that achieves lower DUFM objective value than DNC. In the discussion around Theorem 3, do you claim that the optimal structure for $Z^\star$ under DUFM is necessarily a rank-2 diagonally superior matrix? Or is rank-2 just an illustrative example of suboptimality of DNC, rather than a necessary property of all global optima? Some clarification would be helpful. To be clear, I’m focusing here on the rank of $Z^\star$, rather than the specific construction itself. Also, I suspect it’s not hard to show that rank-1 solutions cannot be diagonally superior. Is that correct?

4) Just to confirm: in Figure 1, are the optimisation curves plotted with respect to scaling parameters applied to fixed DNC and low-rank (LR) constructions? That is, do the curves reflect optimisation over the scalar factors (e.g., $\alpha$ and $\beta$) for a fixed geometry of $Z_{DNC}$ and $Z_{LR}$ respectively?

**Ethical Concerns:**

["NO or VERY MINOR ethics concerns only"]

**Final Justification:**

I believe this line of work in deep learning theory is important and advances the knowledge we have in the field. Thus I have increased my score (from 4 to 5) to account for the merit of this paper. It would be good for the authors in their revision to incorporate the comments and suggestions from the reviewer to increase the delivery and clarity of the work.

**Limitations:**

The authors addressed the limitations of their analysis.

**Paper Formatting Concerns:**

No issues

**Quality:**

3

**Strengths And Weaknesses:**

Strengths:

- For deep linear cross-entropy unconstrained feature models (CE DUFMs), the paper identifies configurations of $K$ and $L$ where DNC does not constitute a global optimum.

- The authors establish that diagonal superior matrices are a necessary condition for global optimality, tying it directly to achieving perfect classification, and proceed to investigate how such matrices can also exhibit reduced rank.

- A notable insight is the contrast between the prevalence of DNC (in regimes of small regularisation and large width) and the existence of lower-rank global solutions that achieve lower objective values, showed by a dimension-counting argument. This is not really unexpected, at least in my opinion, since as $\lambda \to 0$, we are moving to the unregularised case of UFM where only the fit loss matter, so any diagonally superior matrix (simplex ETF construction being one) is equally optimal. The more interesting thing here is the large width regime, since it indicates a generalisation in some way (here the proper NTK-regime analysis is missing) the result of [19] in its deep NC counterpart.

- The results are extended to the deep ReLU UFM case, showing that low-rank, diagonally superior solutions also arise in networks with nonlinearity, although the costruction here is handled in a similar way as in [40], meaning making the features non-negative for ReLU to not have any effect. Nonetheless, I believe it's an important negative result against the optimality of DNC in non-linear networks.

- The analysis is further generalised to imbalanced datasets, demonstrating robustness of the framework beyond the balanced case.

- Finally, the experimental findings support the theory, validating that networks trained in relevant regimes indeed with the constructed low-rank, diagonally superior structures indeed achieve lower loss.

Weaknesses:

1) While the paper demonstrates that DNC is suboptimal under CE DUFM and presents low-rank constructions that outperform it, it stops short of fully characterising the global minimisers. It remains unclear whether the proposed constructions are exhaustive or whether other classes of solutions exist. A complete description of the optimal solution set is still missing.

2) In the ReLU setting, the paper establishes that DNC is not globally optimal, but provides no analysis of what the optimal structures might be. This result is strictly negative, and the paper offers no insight into how nonlinearity shapes the optimal logits or feature structure. The nonlinear case remains theoretically underdeveloped.

3) Theorem 5 argues for the prevalence of DNC-type solutions by comparing the manifold dimensions of the DNC set and the set of low-rank diagonally superior solutions. This is done by accounting for certain degeneracies and essentially bounding the dimension of low-rank solutions between the Stiefel and Grassmann manifolds. Then they compare those bounds to the fixed Grassmannian dimensions of DNC. However, the interpretation of these dimension bounds as evidence of prevalence is still a heuristic step. Dimensionality alone does not account for the topology, measure, or dynamical accessibility of these sets under training. More refined tools from differential geometry would be needed to rigorously support claims about which solutions are likely to be attained. Thus, while the dimension argument is a reasonable first-order heuristic, it should not be equated with likelihood or abundance in a training context. This is something that the authors also acknowledge.

4) A minor detail: Line 77: Zangrando et al., I assume the authors refer to the relevant arxiv paper, but the reference is missing.

---

> ### Author Rebuttal · Authors · 2025-07-29
>
> We thank the reviewers for their thoughtful and detailed feedback. The following responses address each of the raised questions and weaknesses.
>
> **The paper stops short of fully characterizing global minimisers**
>
> We appreciate this important observation. While we agree that a complete characterization of global minima would be highly desirable, our paper does not aim to provide such a result. Rather, our goal is to highlight the emergence of low-rank bias in deep networks and analyse its consequences for the geometry of the loss surface and on optimization outcomes.
>
> * Theorem 2 shows that, for sufficiently large depth L, any lower-rank, diagonally superior matrix eventually outperforms any higher-rank structure (including DNC), revealing a systematic low-rank bias in the landscape.
> * Theorem 3 further establishes that all global minimizers must exhibit approximate low-rankness, with all but a few singular values decaying exponentially in depth. This imposes strong constraints on the structure of global optima, even without a closed-form description.
>
> We believe this perspective offers valuable and broadly applicable insight into the geometry of deep learning with regularization.
>
> We also note that fully characterizing the global minimizers in this setting is inherently difficult: the problem becomes similar to minimizing cross-entropy loss under simultaneous hypersphere and rank constraints—a class of non-convex problems known to be highly analytically challenging.
>
> **There is no analysis of what the optimal structure in the ReLU setting might be**
>
> Thank you for raising this point. We agree that the nonlinear setting remains less theoretically developed than the linear case, and that understanding the structure of global optima under nonlinearity is a challenging and important open problem.
> Our main result in the ReLU case—that DNC is not globally optimal in the deep UFM with weight decay—is intentionally scoped as a first step. While this is a “negative” result in form, it is nontrivial: proving suboptimality even under ReLU required overcoming challenges that do not arise in the linear setting, such as the possibility of increasing rank across layers.
>
> That said, we expect several qualitative features of the linear case to persist under ReLU, including a preference for low-rank diagonally superior structures at large depth (the results of Theorems 2 and 3). We see our result as an early indicator of this shift in behaviour and a foundation for future analysis.
>
> **the interpretation of these dimension bounds as evidence of prevalence is still a heuristic step. Dimensionality alone does not account for the topology, measure, or dynamical accessibility of these sets under training. More refined tools from differential geometry would be needed to rigorously support claims about which solutions are likely to be attained.**
>
> Thank you for this insightful comment. We agree that interpreting dimension as evidence for which solutions are actually attained during training is heuristic in nature and does not capture many important factors such as those you mention. We appreciate the opportunity to clarify both our intent and terminology in Section 3.3.
>
> The primary goal of our theoretical analysis was to highlight a geometric asymmetry in the loss landscape—namely, that DNC solutions occupy a higher-dimensional region than lower-rank alternatives, with this gap being influenced by network width and regularization. Based on this structural observation, we propose a heuristic explanation for the frequent appearance of DNC during training.
>
> While a rigorous dynamical explanation would require more advanced tools—such as gradient flow analysis or differential geometric methods—our contribution is to identify a structural bias in the landscape that favours DNC. To support this perspective, we provide empirical results in Figure 2, which show that DNC becomes more common with increasing width and decreasing regularization—consistent with our heuristic interpretation.
>
> After reviewing your comment, we recognize that the manuscript does not always consistently distinguish between two related but distinct notions of prevalence: (1) prevalence in the geometry of the loss surface, and (2) prevalence as an empirical outcome of training. In particular, some of our language may have inadvertently suggested stronger claims about optimization dynamics than we intended.
>
> We thank the reviewer for making us aware of this and will revise the manuscript to enforce the following terminological distinction:
>
> * Prevalence will refer strictly to geometric properties of the loss surface (e.g., the volume or dimension of solution sets).
> * Persistence will refer to empirical outcomes under gradient-based optimization.
>
> **Zangrando et al., I assume the authors refer to the relevant arxiv paper, but the reference is missing**
>
> Thank you for pointing this out, we will correct this in our revision.
>
> **The work in [43] proves that DNC is optimal for L=2 under MSE loss, and further suggests that the result could extend to CE loss… Could you clarify whether this discrepancy is entirely due to the change in loss function (MSE vs CE), or if there are additional structural or technical reasons that break the equivalence?**
>
> Yes, the discrepancy is entirely due to the change in loss function. Your next question includes an accurate description of why this is: under MSE, the network output must have the same rank as the matrix $Y$ (i.e., rank $K$); otherwise, the fit loss is bounded below by a positive constant. The equivalent of Lemma 1 from our paper then forces the network to be rank $K$ throughout, and DNC ends up being the best among solutions at that rank. The CE case is different: to fit $Y$, the network output only needs to be diagonally superior, which imposes a much weaker rank constraint. Other loss functions will impose different structural conditions on the network output to achieve arbitrarily small fit loss. These differences in constraints affect how low-rank bias manifests—and ultimately influence the structure of global minima when $L>1$.
>
> **The authors mention that the single-layer UFM case (L=1) cannot capture the optimal structure of the last-layer for deep networks… Am I correct in assuming that this is specific to CE? … Also, for the CE case, I can see this as an attribute of a deep linear network (from Lemma 1), but not sure how/if this generalises when non-linearity takes effect.**
>
> Yes, you are correct—different losses impose different conditions on the network output to achieve arbitrarily small fit loss, which in turn leads to different outcomes in how low-rank bias manifests. The deep UFM case when using MSE loss and linear layers is covered by Dang et al. [6] who show that the optimal solution still must be DNC, and this is for the reason you point out.
>
> When nonlinearity is introduced, this becomes theoretically much harder to evaluate, as ReLU can both raise and lower the rank of a matrix in nontrivial ways. For example, we can construct rank 3 matrices that become full rank after ReLU application by using a diagonally superior matrix that has its only positive entries on the diagonal. As a consequence, there is no clear analogue of Lemma 1 in this setting.
>
> In experiments, solutions tend to use positive feature matrices, in which case ReLU acts like a linear function and the network effectively behaves as a linear model. However, due to the complexity of the loss surface, we suspect these solutions do not represent global optima—and without further theoretical advances, we do not yet know what the global optimal structures are in the ReLU case.
>
> **do you claim that the optimal structure for Z⋆ under DUFM is necessarily a rank-2 diagonally superior matrix?**
>
> The result of Theorem 3 applies as we increase the number of layers, so we expect that as $L$ becomes large, the optimal structure will have only two singular values above an arbitrarily small threshold. A key reason for this is that the Schatten quasi-norm, which captures the regularization terms, approximates the hard rank of the matrix—and the hard rank does not penalize overall scale.
>
> For small or moderate $L$ the picture is different: the overall scale affects the loss non-trivially, and different diagonally superior matrices can have different fit losses at a fixed scale. Hence, at small $L$ low rank bias is traded off against the fit loss, and we do not expect the optimal solution to necessarily be a rank-minimal diagonally superior matrix.
>
> This trade-off is also evident in Theorem 2: if we consider setting $M$ to be a rank-minimal diagonally superior matrix, it only becomes preferable when the low-rank bias is strong enough to outweigh the loss from reduced fit. Therefore, to be clear, we do not believe that for small or moderate $L$, $Z^*$ must be a rank-2 diagonally superior matrix.
>
> Also, yes—it is straightforward to show that a rank-1 matrix can only be diagonally superior when its size is 1 or 2.
>
> **in Figure 1, are the optimisation curves plotted with respect to scaling parameters applied to fixed DNC and low-rank (LR) constructions?**
>
> The results in Figure 1 show the losses of deep UFMs trained from random initialization. It is only after training that we identify them as $Z_{DNC}$ and $Z_{LR}$, based on the recovered logit matrices, as well as the features and weights matching the respective definitions.
>
> The dotted lines indicate the losses achieved by DNC and LR solutions at the optimal choice of α and β. We apologize that this was not clear and will revise the caption accordingly.
>
>
> We appreciate the reviewer’s insightful questions and suggestions, which have helped us identify several areas to refine and clarify in our manuscript. We will incorporate these revisions to ensure the claims are more precisely framed, and the scope of our results more clearly communicated. Thank you again for your constructive and thoughtful evaluation.

---

> > ### Comment · Reviewer_9eBN · 2025-08-05
> >
> > Thank you for addressing my questions and concerns. I'm happy to increase my score, since I believe this line of work is important.

---

### Decision · Program_Chairs · 2025-09-17

**Decision:**

Accept (poster)

**Comment:**

This paper studies the interaction between low-rank bias and Neural Collapse (NC), building upon previous work that showed that NC is not optimal in large depth networks with weight decay. This paper shows a similar result for linear networks with the cross-entropy loss and nonlinear networks. This paper also provides a few arguments that even though NC is sub-optimal, GD might still converge to such a solution because of there are comparatively many critical points that exhibit NC that GD might converge to. This paints a more subtle picture, where NC could still be proven if one can track the GD dynamics closely enough.

In agreement with the reviewers, we are happy to accept this paper as a poster at NeurIPS.